# ENHANCING LANGUAGE MODEL REASONING WITH STRUCTURED MULTI-LEVEL MODELING

**Siheng Xiong[1], Ali Payani[2], Faramarz Fekri[1]**
[1]Georgia Institute of Technology    [2]Cisco Research
sxiong45@gatech.edu    apayani@cisco.com    fekri@ece.gatech.edu

## ABSTRACT

Inference-time scaling enhances a model's reasoning by extending its chain-of-thought (CoT). However, existing approaches typically rely on a single policy trained with outcome-reward reinforcement learning (RL), which often suffers from long-horizon plan failures where the implicit plan drifts from valid strategies, especially for small LMs with limited capacity. To address this, we propose Multi-Level Reasoning (MLR), which reformulates long-CoT generation as a two-level stochastic process. A high-level planner generates structured step descriptors specifying both the reasoning mode and the semantic subgoal. The low-level executor then produces detailed reasoning conditioned on these descriptors, forming an alternating plan–execute loop. To maintain scalability, we adopt a minimal design where the base model serves as the low-level policy and a lightweight LoRA module implements the high-level policy. For training, we observe that outcome-reward RL provides sparse and delayed feedback for long trajectories (e.g., several thousand tokens), hindering credit assignment. We therefore introduce iterative Step-DPO, a process-level preference optimization scheme that leverages Twisted Sequential Monte Carlo (TSMC) to provide scalable stepwise supervision. This yields more effective training, improved stability, and higher accuracy. Extensive experiments on challenging math, science, and logical reasoning benchmarks show that, under the same reduced data budget (10% SFT and 5% preference relative to the DeepSeek-R1 distillation setup), MLR outperforms both SFT-based distillation and strong RL/preference-optimization baselines across multiple base models and tasks. Moreover, MLR exhibits slower performance degradation on long-horizon reasoning, demonstrating stronger robustness under extended CoT generation[1].

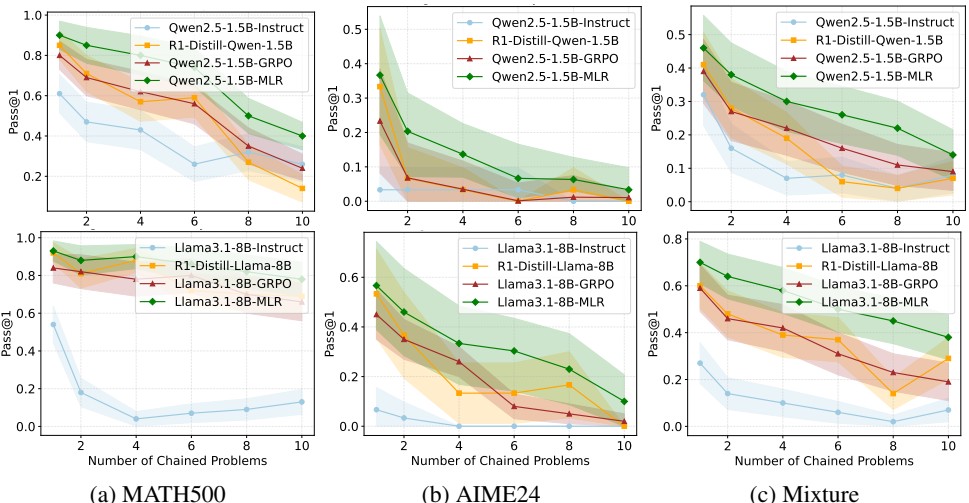

(a) MATH500    (b) AIME24    (c) Mixture

Figure 1: Long-horizon accuracy on MATH500, AIME24, and a mixture benchmark (MATH500 / AIME24 / GPQA / BoardGameQA). Top: Qwen-2.5-1.5B; bottom: Llama-3.1-8B. Problems are concatenated in the prompt to simulate long horizons; MLR degrades more slowly than baselines (see Section C for details).

---

[1]Code and data are available at `https://github.com/xiongsiheng/MLR`.

# 1 INTRODUCTION

OpenAI's o1 series (OpenAI, 2024) introduce inference-time scaling by allocating more computation to the reasoning process, often realized as longer Chain-of-Thought (CoT) (Wei et al., 2022) generation. This design improves performance on complex reasoning tasks compared to non-reasoning models. Building on this direction, DeepSeek (Guo et al., 2025) proposes a large-scale reinforcement learning (RL) pipeline that directly incentivizes the generation of long CoTs through policy optimization. Despite their effectiveness, approaches that rely on single-policy long CoTs for RL face important limitations, including long-horizon plan failures and the inherent challenges of RL with sparse outcome rewards[2]. These issues are especially pronounced for small LMs with long CoTs due to their limited capacity.

In reinforcement learning, long-horizon policy learning remains a fundamental challenge due to difficult credit assignment. To address this, hierarchical reinforcement learning (HRL) introduces temporal abstraction by learning high- and low-level policies that operate over different time scales (Sutton et al., 1999). While HRL has proven effective in domains such as robotics (Nachum et al., 2018; Gupta et al., 2019), its application to LMs[3] presents several challenges: 1) Scalability. Modeling multiple policies, especially when implemented as separate LMs, incurs additional compute. Naïve multi-agent setups introduce substantial coordination and communication overhead (Guo et al., 2024b). 2) Flexibility. The plan-then-execute structure (Wang et al., 2023a; Xu et al., 2023) is brittle in LM-based reasoning tasks where new information or execution failures may require mid-course corrections. Thus, it is desirable to allow the high-level plan to evolve dynamically based on the ongoing progress of the low-level execution process. To address these challenges, we propose a multi-level reasoning (MLR) strategy, where the model alternates between generating a step-level descriptor and its corresponding detailed content, to enable efficient multi-policy modeling and dynamic plan adaptation.

We further introduce an iterative Step-DPO pipeline for long-CoT training to improve stability and performance. Existing RL fine-tuning frameworks struggle to obtain effective process-level supervision (Guo et al., 2025). First, evaluating the correctness of intermediate steps is difficult. Automated annotation using LLMs (Guo et al., 2024a) can be noisy, while manual annotation (Lightman et al., 2023) is prohibitively expensive at scale. Second, introducing a separate process reward model (PRM) adds complexity. It is vulnerable to reward hacking (Gao et al., 2023), requires substantial training data, and complicates the pipeline by necessitating repeated retraining. To overcome these limitations, we repurpose Twisted Sequential Monte Carlo (TSMC) (Doucet et al., 2001; Whiteley & Lee, 2014; Heng et al., 2020) to construct step-level preferences for Step-DPO (Lai et al., 2024). In the LM-based reasoning setting, TSMC assigns importance weights that correct distribution mismatch (and incorporate terminal correctness), thereby upweighting partial trajectories whose continuations are more consistent with reaching a correct outcome. We then define process preferences between two candidate continuations at the same step by comparing their incremental log-weights. This formulation has two advantages: (i) taking log-increments converts multiplicative weight updates into an additive form, improving numerical stability; and (ii) it matches the pairwise preference objective used in Step-DPO. Empirically, this yields stable step-level preferences and improves long-CoT training efficiency and performance.

We summarize our key contributions as follows:

- We propose Multi-Level Reasoning (MLR) to address the limitations of single-policy long-CoT approaches, such as long-horizon plan failures. MLR decomposes reasoning into alternating high-level step descriptors and low-level detailed content, enabling structured abstraction, dynamic plan adjustment, and more reliable long-horizon reasoning.

- We repurpose Twisted Sequential Monte Carlo (TSMC) to provide process-level preferences for Step-DPO training. This eliminates the need for a separate process reward model, while supplying reliable and informative supervision throughout long trajectories.

---

[2]See Section 2 for a detailed discussion.

[3]Note that HRL differs from prompting-based CoT decomposition: it treats high- and low-level actions as separate distributions with distinct objectives and temporal scopes.

- We perform extensive experiments on challenging benchmarks in math, science, and logical reasoning. Results show that our approach consistently outperforms both SFT-based distillation and strong RL/preference-optimization baselines under the same data budget.

## 2 INFERENCE-TIME SCALING VIA LONG CHAIN-OF-THOUGHT

**Formulation.**   Given a query $q$, a reasoning model generates a CoT $c$ before producing the final response $a$, where $q, c, a$ are all sequences of tokens, i.e., $c = (c[1], c[2], \cdots, c[L])$. To improve performance, these models allocate more computation to reasoning, often realized as generating longer $c$ with behaviors such as exploration, self-verification, and reflection. The generation of long CoTs follows the standard autoregressive modeling: the probability of each token $c[l]$ depends on its preceding tokens $(c[1 : l - 1])$, which enables the factorization of the joint likelihood of the entire sequence as:

$$p_\theta(c[1:L]) = \prod_{l=1}^{L} p_\theta(c[l] \mid c[1:l-1]).$$
(1)

Note that, for notational simplicity, we omit the conditioning on $q$ in Eq. 1 and in the following derivations. Training the model $p_\theta$ involves maximizing the likelihood of each token conditioned on its prefix, i.e., optimizing $p_\theta(c[l] \mid c[1:l-1])$ over the training data.

**Post-training.**   Guo et al. (2025) detail how they incentivize the long CoT generation from a base model through large-scale RL without relying on SFT. Specifically, they employ GRPO guided by rule-based outcome reward. For each query $q$, GRPO samples a group of outputs $\{o_1, o_2, \cdots, o_G\}$ from the old policy $\pi_{\theta_{old}}$, where each output is composed of a CoT followed by the final response, i.e., $o_i = [c_i, a_i]$, and then optimizes the policy $\pi_\theta$ by maximizing the corresponding objective.

**Discussion on the weakness of single-policy long CoT.**   The above approach of using single-policy long CoT enables inference-time scaling with LMs, but introduces several issues:

1) **Long-horizon plan failures.** In single-policy long CoT generation, the same policy is responsible for both planning and execution. Without guidance or structure, errors can accumulate and cause the implicit plan to drift away from any valid strategy (see examples in Section C).

2) **Long-horizon RL with sparse outcome reward.** Long CoTs involve thousands of token-level actions before receiving a reward, which hinders credit assignment. As shown in Figure 2, these trajectories can be extremely long, with errors occurring at widely varying positions, which undermines the effectiveness of outcome-based RL. Moreover, Figure 3 shows that latency and memory usage grow rapidly with trajectory length, while outcome-based supervision requires the entire trajectory to finish before feedback is provided. Consequently, optimization can be slow and unstable, especially early in training when correct trajectories are rare.

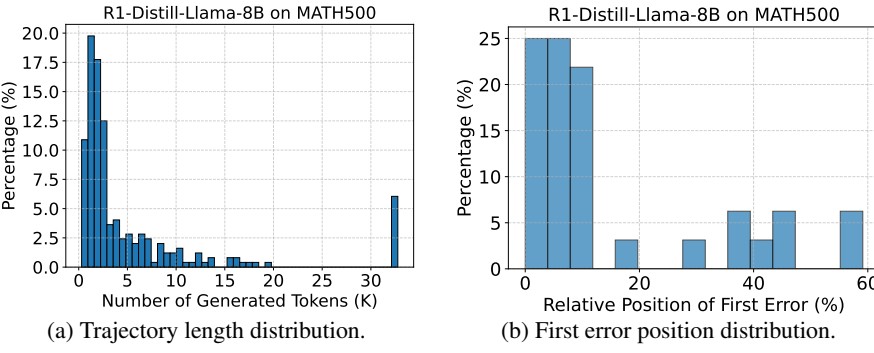

(a) Trajectory length distribution.  (b) First error position distribution.

Figure 2: The CoT trajectories can be long, and the position of the first error varies widely across trajectories, making outcome-only RL fine-tuning less informative. Statistics in (b) are computed from 100 trajectories with incorrect final answers, where the first error was manually identified.

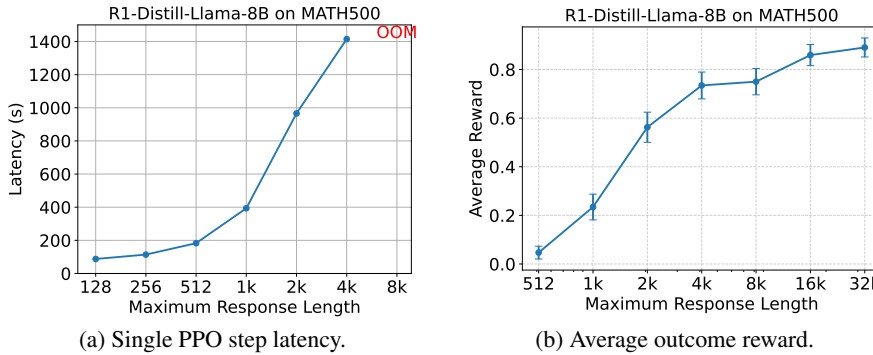

(a) Single PPO step latency.

(b) Average outcome reward.

Figure 3: Outcome-reward RL becomes increasingly costly for long trajectories: (a) PPO step latency grows with the maximum generation length, and (b) feedback is only available after the full trajectory completes. All measurements use Verl (Sheng et al., 2024) with vLLM (Kwon et al., 2023) on a single A100 node (batch size = 32).

# 3 METHODOLOGY

## 3.1 ENHANCING INFERENCE-TIME SCALING WITH MULTI-LEVEL REASONING

**Reformulation as MLR.** We reconceptualize inference-time scaling by shifting from "single-policy long CoT" to "multi-level reasoning" strategy (Figure 4). Here, the reasoning process is organized hierarchically, capturing both high-level abstractions $d$ and low-level details $c$. Formally, the overall reasoning chain is represented at two levels: $d = (d^{(1)}, \cdots, d^{(M)})$ and $c = (c^{(1)}, \cdots, c^{(M)})$, where $M$ denotes the number of reasoning steps, $d^{(m)}$ is the descriptor of step $m$, and $c^{(m)}$ represents the corresponding detailed content. The autoregressive likelihood can be factorized hierarchically as follows:

$$p_\theta^H(d) = \prod_{m=1}^{M} p_\theta^H(d^{(m)} \mid d^{(1:m-1)}, c'^{(1:m-1)}), \quad p_\theta^L(c) = \prod_{m=1}^{M} p_\theta^L(c^{(m)} \mid d^{(1:m)}, c^{(1:m-1)}) \quad (2)$$

where $c'^{(m)}$ denotes a compressed summary of the detailed content $c^{(m)}$. We also experimented with removing the previous descriptors $d^{(1:m-1)}$ from Equation (2), but found that including them improves performance and facilitates training. The inference procedure is summarized in Algorithm 1.

**Architecture.** Figure 5 illustrates the architecture used to implement our MLR strategy. The model alternates between a high-level policy that produces step descriptors and a low-level policy that

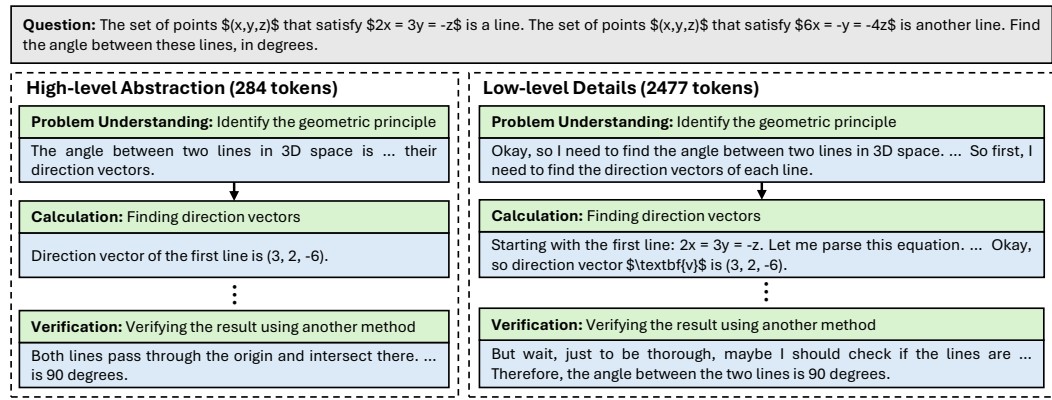

Figure 4: Illustration of MLR. MLR augments single-policy reasoning with a high-level policy that generates step descriptors (reasoning mode and subgoal), providing structured guidance to reduce long-horizon plan drift and facilitate more effective learning.

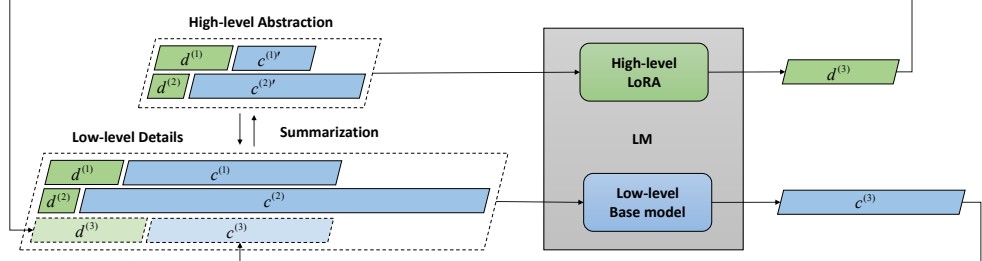

Figure 5: Overview of the architecture. The model alternates between generating high-level descriptors and the corresponding low-level content. Ablations motivating the design are in Section B.

generates the corresponding detailed content. The low-level policy is implemented with the base LM, which conditions on the sequence of prior descriptors and detailed contents, together with the current descriptor, to generate the next detailed content. The high-level policy is implemented with a lightweight LoRA module (Hu et al., 2022), which conditions on previous descriptors and summaries to produce the next descriptor. Since descriptors are much shorter than full reasoning content, this component remains compact and computationally efficient. The design rationale behind this architecture as well as ablation studies are provided in Section B. Additionally, we fine-tune an independent, lightweight LLM for summarization, which is shared across different base models.

## 3.2 ITERATIVE STEP-DPO WITH PROCESS-LEVEL PREFERENCES

To train our model effectively, we introduce an iterative Step-DPO pipeline that performs stepwise preference optimization with periodic data refresh.

**Supervised fine-tuning.** We collect long-CoT traces from DeepSeek-R1 (Guo et al., 2025) and use DeepSeek-V3 (Liu et al., 2024) to segment each trace into steps and annotate each step with a descriptor via in-context learning. We further distill each step's detailed reasoning into a compact summary that preserves the planning-relevant state. This yields aligned triples of descriptors, detailed contents, and summaries. We first full fine-tune the base LM on low-level contents, then freeze it and train a LoRA module on high-level descriptors. We also train a lightweight summarizer on the summary data and keep it fixed during iterative Step-DPO. See Section 4 for implementation details and Section B for design choices and ablations.

**Multi-level Step-DPO.** Following Lai et al. (2024), we optimize the MLR framework with the following stepwise preference loss $\mathcal{L}_{\text{step-DPO}}$:

$$-\mathbb{E}_{(x^{(m)}, y_+^{(m)}, y_-^{(m)}) \sim \mathcal{D}_{\text{pref}}} \frac{1}{M} \sum_{m=1}^{M} \log \sigma \Big[ \beta \big( \log \frac{p_\theta(y_+^{(m)} \mid x^{(m)})}{p_{\text{ref}}(y_+^{(m)} \mid x^{(m)})} - \log \frac{p_\theta(y_-^{(m)} \mid x^{(m)})}{p_{\text{ref}}(y_-^{(m)} \mid x^{(m)})} \big) \Big]. \quad (3)$$

where $(x^{(m)}, y_+^{(m)}, y_-^{(m)})$ is a preference triple at step $m$. During optimization, we maintain a low-level policy (the base LM) and a high-level policy (the LoRA adapter). For low-level preference pairs, we disable the LoRA adapter and update only the base LM parameters with $\big((d^{(1:m)}, c^{(1:m-1)}), c_+^{(m)}, c_-^{(m)}\big)$; for high-level pairs, we freeze the base LM and update only the LoRA parameters with $\big((d^{(1:m-1)}, c'^{(1:m-1)}), d_+^{(m)}, d_-^{(m)}\big)$.

**Multi-level update schemes.** A key challenge in jointly optimizing the two policies lies in designing an effective update scheme. We adopt an interleaved strategy: mini-batches of high-level and low-level examples are alternated, allowing the planner and executor to be trained jointly while preserving modularity. We compare this update scheme with alternatives in Section B.

**Iterative Step-DPO for online optimization.** Motivated by the benefits of on-policy data sampling in RL, we adopt an iterative Step-DPO framework for improved optimization. Specifically, in the $t$-th

iteration, we use the current policies to sample preference pairs to create the preference data $\mathcal{D}_{\text{pref}}^{(t)}$. We then obtain the next-iteration policies by minimizing $\mathcal{L}_{\text{step-DPO}}^{(t)}$:

$$-\mathbb{E}_{(x^{(m)}, y_+^{(m)}, y_-^{(m)}) \sim \mathcal{D}_{\text{pref}}^{(t)}} \frac{1}{M} \sum_{m=1}^{M} \log \sigma \Big[\beta\big(\log \frac{p_\theta(y_+^{(m)} \mid x^{(m)})}{p_\theta^{(t)}(y_+^{(m)} \mid x^{(m)})} - \log \frac{p_\theta(y_-^{(m)} \mid x^{(m)})}{p_\theta^{(t)}(y_-^{(m)} \mid x^{(m)})}\big)\Big]. \quad (4)$$

The procedure is summarized in Algorithm 2. More implementation details are provided in Section B.

**Process preference modeling.** A key component of our iterative Step-DPO pipeline is the process-level supervision for both the high-level descriptors $d^{(m)}$ and the low-level detailed contents $c^{(m)}$. Consider the full reasoning trajectory after a prefix $x^{(m)}$ as future tokens $\tau_{m+1:M} = (d^{(m+1)}, c^{(m+1)}, \ldots, d^{(M)}, c^{(M)})$, generated by a rollout policy $p_{\text{roll}}$. The survival probability of $x^{(m)}$ is

$$g(x^{(m)}) := \mathbb{E}_{\tau_{m+1:M} \sim p_{\text{roll}}(\cdot | x^{(m)})}\big[R(x^{(m)}, \tau_{m+1:M})\big], \quad (5)$$

where $R(x^{(m)}, \tau_{m+1:M}) \in \{0, 1\}$ indicates whether the final answer is correct.

Given an estimate of the survival probability $\hat{g}$, we construct preference data using a utility defined as the increment in log-survivability:

$$U(y^{(m)}) = \log \tilde{g}\big(x^{(m)}, y^{(m)}\big) - \log \tilde{g}(x^{(m)}). \quad (6)$$

where the survivability is clipped as $\tilde{g} = \text{clip}(\hat{g}, \varepsilon, 1 - \varepsilon)$ with $\varepsilon = 0.001$ for numerical stability. Intuitively, $U(y^{(m)})$ quantifies how the selected candidate changes the probability of eventual success relative to the preceding prefix. Then we impose the condition that the utility difference satisfies the following:

$$U(y_+^{(m)}) - U(y_-^{(m)}) = \log \tilde{g}\big(x^{(m)}, y_+^{(m)}\big) - \log \tilde{g}\big(x^{(m)}, y_-^{(m)}\big) > \delta, \quad (7)$$

where the margin threshold $\delta$ ensures the reliability of the preference data.

**Twisted Sequential Monte Carlo.** A key challenge of the above approach is computational cost: estimating future solvability by naive Monte-Carlo lookahead would require running the base model many times for each prefix. To address this, we adopt Twisted Sequential Monte Carlo (TSMC) to obtain *efficient* and *reliable* process-level signals. Importantly, our objective only requires *pairwise comparison* between candidate steps under the same prefix (i.e., generating accurate step-level preferences), rather than a fully calibrated estimate of the absolute survivability probability.

Given a prefix, the $k$-th particle at step $m-1$ has state $x_k^{(m-1)}$. We first sample a candidate step $y_k^{(m)} \sim p_{\text{roll}}\big(\cdot \mid x_k^{(m-1)}\big)$ and form the updated state $x_k^{(m)} = [x_k^{(m-1)}, y_k^{(m)}]$. Its importance weight is updated as

$$W_k^{(m)} = W_k^{(m-1)} \cdot \tilde{w}_k^{(m)}, \quad (8)$$

with incremental weight

$$\tilde{w}_k^{(m)} = G_m\big(x_k^{(m)}\big) \cdot \frac{p_\theta\big(x_k^{(m)} \mid x_k^{(m-1)}\big)}{p_{\text{roll}}\big(x_k^{(m)} \mid x_k^{(m-1)}\big)} \cdot \frac{\phi_m\big(x_k^{(m)}\big)}{\phi_{m-1}\big(x_k^{(m-1)}\big)}. \quad (9)$$

Here $W_k^{(m)}$ is the step-$m$ importance weight with $W_k^{(0)} = 1$. The potential function $G_m$ is defined as $G_m\big(x_k^{(m)}\big) = 1$ for $m < M$ and $G_M\big(x_k^{(M)}\big) = \mathbf{1}_{\text{correct}}\big(x_k^{(M)}\big)$, i.e., final answer correctness. $p_\theta$ denotes the base model, $p_{\text{roll}}$ is the rollout policy, and $\phi_m$ is a learned survivability critic at step $m$. When $p_{\text{roll}}$ is close to $p_\theta$, the importance correction term $p_\theta/p_{\text{roll}}$ and the twisting ratio $\phi_m/\phi_{m-1}$ become mild. In this regime, the incremental weight is dominated by the task potential $G_m(\cdot)$, and the resulting estimator provides a low-variance signal that is sufficient for ranking candidate steps and constructing step-level preference pairs. Finally, the survivability estimate is given by:

$$\hat{g}_K(x^{(m)}) = \frac{1}{K} \sum_{k=1}^{K} W_k^{(M_k)} = \frac{1}{K} \sum_{k=1}^{K} W_k^{(m)} \prod_{j=m+1}^{M_k} \tilde{w}_k^{(j)}. \quad (10)$$

where $K$ represents the number of particles, and $M_k$ denotes the termination step of particle $k$. In our use case, $\hat{g}_K(\cdot)$ is primarily used as a *relative* measure of future solvability to compare candidate steps under the same prefix.

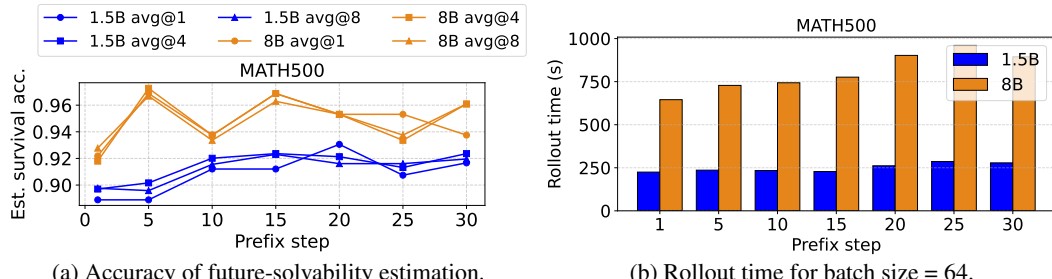

(a) Accuracy of future-solvability estimation.  (b) Rollout time for batch size = 64.

Figure 6: Monte Carlo rollout analysis. (a) Future-solvability estimation accuracy is similar between R1-Distill-LLaMA-8B and R1-Distill-Qwen-1.5B. (b) Runtime increases with model size, and R1-Distill-LLaMA-8B is substantially slower than R1-Distill-Qwen-1.5B. See Section B for implementation details.

To keep rollouts efficient while limiting distribution mismatch, we fine-tune a small LM on the same low-level SFT data and use it as $p_{\text{roll}}$. This is motivated by (i) a closer rollout distribution, (ii) the fact that preference construction only requires correct relative ranking, and (iii) substantially lower rollout cost. As shown in Figure 6, the small model provides comparable preference quality while being much faster. We further use an easy/hard split: for hard benchmarks where the small model lags far behind the base LM, we instead roll out with the base model (AIME24 and GPQA in practice).

## 4 EXPERIMENTS

**Datasets.** We evaluate on math (MATH500 (Hendrycks et al., 2021), AIME24 (MAA, 2024)), science (GPQA-Diamond (Rein et al., 2023)), and logical reasoning (BoardGameQA-Hard (Kazemi et al., 2023)); dataset overview and split protocols are provided in Section A. For training, we construct a multi-level dataset and divide it into two disjoint parts: SFT data and iterative preference data for Step-DPO. The SFT set contains about 80K examples produced using the multi-level decomposition procedure described in Section 3.2. We reserve an additional 10K training prompts for Step-DPO preference sampling. Details of dataset construction are provided in Section B.

**Implementation details.** We full fine-tune three base LMs (Qwen-2.5-1.5B (Yang et al., 2024a), Qwen-2.5-MATH-7B (Yang et al., 2024b), and LLaMA-3.1-8B (Grattafiori et al., 2024)) on low-level SFT data, then freeze them and train a lightweight LoRA adapter as the high-level policy. We also fine-tune Qwen2.5-0.5B-Instruct (Yang et al., 2024a) as a summarizer, shared across base models. Iterative Step-DPO is implemented with TRL (von Werra et al., 2020). Each round samples approximately 3K prompts; per prompt we select 4 steps and generate $M = 2$ candidates, scored by Equation (6). We use a Qwen-2.5-1.5B SFT model as the rollout policy for MATH500 and BoardGameQA, and the same base model for AIME24 and GPQA; we use $K = 8$ rollouts per prefix (see Figure 6 and section 4 for analysis and parameter studies). We form one preference pair per prefix by taking the largest utility gap and retaining it only if the margin exceeds $\delta = 0.4$. Updates use batch size 32 for $E = 4$ epochs with $\beta = 0.1$, and continuations are capped at 8,192 tokens. Additional details are in Section B.

**Baselines.** We compare against the following baselines: (i) the base model and the instruction-tuned model; (ii) outcome-reward RL applied directly to the base model (SimpleRL (Zeng et al., 2025));

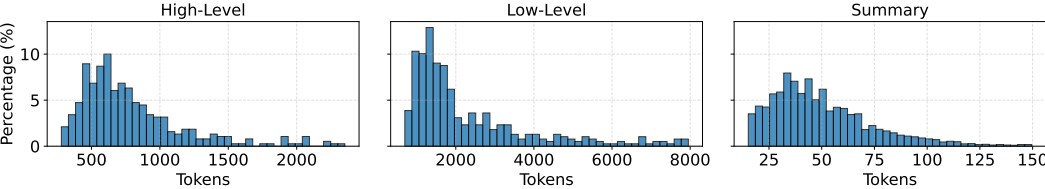

Figure 7: Length distributions for high-level trajectories, low-level trajectories, and distilled summaries in our SFT dataset. Trajectories longer than 8,192 tokens are discarded.

Table 1: Overall performance across benchmarks. The best performance for each task using the same base model is in bold. Asterisks (*) denote available results obtained from the official reports.

| Method | Math MATH500 Pass@1 | AIME24 Pass@1 | AIME24 Cons@32 | Science GPQA (Diamond) Pass@1 | Logic BoardGameQA (Hard) Pass@1 | Avg. Pass@1 |
|---|---|---|---|---|---|---|
| | | | Qwen-2.5-1.5B | | | |
| Base | $29.6 \pm 0.6$ | $0.0 \pm 0.0$ | $0.0 \pm 0.0$ | $21.2 \pm 1.1$ | $35.0 \pm 1.8$ | 21.5 |
| Instruct | $54.6 \pm 0.4$ | $3.3 \pm 1.5$ | $10.0 \pm 0.3$ | $25.2 \pm 1.4$ | $41.0 \pm 1.0$ | 31.0 |
| SimpleRL | $59.0^*$ | $6.7^*$ | $4.2^*$ | $-$ | $-$ | $-$ |
| DeepSeek-R1-Distill | $83.9^*$ | $28.9^*$ | $43.3 \pm 0.4$ | $33.8^*$ | $40.0 \pm 2.2$ | 47.7 |
| Plan-and-Solve | $60.8 \pm 1.0$ | $2.0 \pm 1.4$ | $6.2 \pm 0.4$ | $20.1 \pm 1.6$ | $44.6 \pm 1.6$ | 31.9 |
| SFT + DPO | $76.5 \pm 1.3$ | $12.0 \pm 1.9$ | $21.6 \pm 0.7$ | $27.6 \pm 1.6$ | $51.8 \pm 1.7$ | 42.0 |
| SFT + Step-DPO | $81.4 \pm 1.2$ | $24.0 \pm 2.0$ | $36.5 \pm 0.7$ | $29.0 \pm 1.7$ | $56.2 \pm 1.7$ | 47.8 |
| SFT + GRPO | $82.1 \pm 1.2$ | $25.2 \pm 2.2$ | $36.0 \pm 0.8$ | $30.2 \pm 1.5$ | $56.0 \pm 1.6$ | 48.4 |
| MLR (SFT only) | $62.0 \pm 1.2$ | $8.9 \pm 1.4$ | $13.3 \pm 0.4$ | $26.0 \pm 2.0$ | $46.4 \pm 1.5$ | 35.8 |
| MLR | $\mathbf{86.1} \pm 1.0$ | $\mathbf{31.2} \pm 1.0$ | $\mathbf{47.4} \pm 0.4$ | $\mathbf{37.6} \pm 1.9$ | $\mathbf{62.0} \pm 1.7$ | **54.2** |
| | | | Qwen-2.5-MATH-7B | | | |
| Base | $52.0 \pm 0.5$ | $2.0 \pm 1.0$ | $5.0 \pm 0.3$ | $20.5 \pm 1.1$ | $33.0 \pm 1.6$ | 26.9 |
| Instruct | $82.1 \pm 0.4$ | $16.7 \pm 1.8$ | $34.0 \pm 0.4$ | $27.8 \pm 1.3$ | $44.5 \pm 1.4$ | 42.8 |
| SimpleRL | $80.2^*$ | $40.0^*$ | $24.0^*$ | $-$ | $-$ | $-$ |
| DeepSeek-R1-Distill | $92.8^*$ | $55.5^*$ | $78.0 \pm 0.4$ | $49.1^*$ | $42.4 \pm 1.4$ | 60.0 |
| Plan-and-Solve | $85.6 \pm 0.9$ | $18.2 \pm 1.7$ | $34.9 \pm 0.5$ | $28.4 \pm 1.6$ | $52.1 \pm 1.5$ | 46.1 |
| SFT + DPO | $87.4 \pm 1.0$ | $36.0 \pm 1.8$ | $53.0 \pm 0.5$ | $36.0 \pm 1.6$ | $54.5 \pm 1.5$ | 53.4 |
| SFT + Step-DPO | $88.5 \pm 0.9$ | $48.5 \pm 1.9$ | $70.5 \pm 0.5$ | $48.0 \pm 1.7$ | $56.0 \pm 1.6$ | 60.3 |
| SFT + GRPO | $89.7 \pm 1.0$ | $46.5 \pm 1.9$ | $66.2 \pm 0.5$ | $46.0 \pm 1.8$ | $57.5 \pm 1.6$ | 59.9 |
| MLR (SFT only) | $86.3 \pm 1.0$ | $22.4 \pm 1.9$ | $40.5 \pm 0.5$ | $34.6 \pm 1.7$ | $54.8 \pm 1.6$ | 49.5 |
| MLR | $\mathbf{94.1} \pm 0.9$ | $\mathbf{58.8} \pm 1.8$ | $\mathbf{80.5} \pm 0.4$ | $\mathbf{51.2} \pm 1.5$ | $\mathbf{60.5} \pm 1.6$ | **66.2** |
| | | | Llama-3.1-8B | | | |
| Base | $13.6 \pm 0.4$ | $0.0 \pm 0.0$ | $0.0 \pm 0.0$ | $1.5 \pm 1.0$ | $2.0 \pm 1.1$ | 4.3 |
| Instruct | $51.9 \pm 0.2$ | $6.7 \pm 1.8$ | $13.3 \pm 0.2$ | $22.7 \pm 0.6$ | $40.0 \pm 1.2$ | 30.3 |
| SimpleRL | $23.0^*$ | $0.0^*$ | $0.2^*$ | $-$ | $-$ | $-$ |
| DeepSeek-R1-Distill | $89.1^*$ | $50.4^*$ | $70.0 \pm 0.4$ | $49.0^*$ | $46.0 \pm 3.8$ | 58.6 |
| Plan-and-Solve | $62.4 \pm 1.1$ | $12.3 \pm 1.8$ | $24.1 \pm 0.4$ | $31.0 \pm 1.6$ | $47.2 \pm 1.7$ | 38.2 |
| SFT + DPO | $74.1 \pm 1.5$ | $32.4 \pm 1.8$ | $52.0 \pm 0.6$ | $44.0 \pm 1.7$ | $56.0 \pm 1.7$ | 51.6 |
| SFT + Step-DPO | $82.4 \pm 1.3$ | $42.6 \pm 2.0$ | $61.2 \pm 0.5$ | $49.2 \pm 1.5$ | $62.1 \pm 1.4$ | 59.1 |
| SFT + GRPO | $86.5 \pm 1.4$ | $42.0 \pm 2.0$ | $61.0 \pm 0.5$ | $47.0 \pm 1.6$ | $64.5 \pm 1.5$ | 60.0 |
| MLR (SFT only) | $63.8 \pm 1.2$ | $20.2 \pm 2.0$ | $36.7 \pm 0.4$ | $36.2 \pm 1.8$ | $48.5 \pm 1.8$ | 42.2 |
| MLR | $\mathbf{91.5} \pm 1.3$ | $\mathbf{53.2} \pm 2.0$ | $\mathbf{73.3} \pm 0.4$ | $\mathbf{52.8} \pm 1.5$ | $\mathbf{67.0} \pm 1.4$ | **66.1** |

(iii) SFT-based distillation using vanilla long CoTs (R1-Distill (Guo et al., 2025)); (iv) planning-style prompting (Plan-and-Solve (Wang et al., 2023a)); (v) preference-optimization baselines trained on our SFT initialization (DPO (Rafailov et al., 2023) and Step-DPO (Lai et al., 2024)); and (vi) outcome-reward RL applied to the SFT initialization (GRPO (Shao et al., 2024)).

All baselines that we train ourselves (DPO, Step-DPO, GRPO, and Plan-and-Solve) use exactly the same training data budget; see Section B for details. Results for external baselines (Instruct, SimpleRL, and R1-Distill) are included as strong reference points. During evaluation, we use greedy decoding for Base and Instruct. For all other methods (including ours), we follow Guo et al. (2025) and use sampling-based decoding with temperature $0.6$ and top-$p$ $0.95$ to generate 8 responses per prompt. We report pass@1, and additionally cons@32 for AIME24.

**Empirical results.** We present representative outputs in Section C and error analysis in Section C. Table 1 summarizes overall performance, and Figure 9 shows how MLR evolves across training

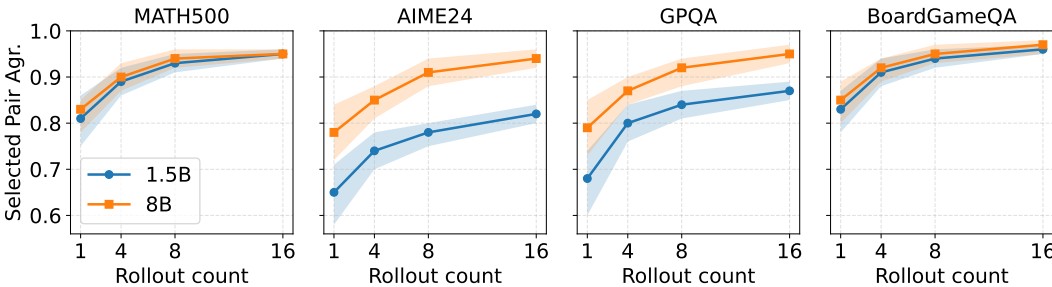

Figure 8: Agreement of selected preference pairs versus rollout count $K$ on (a) MATH500, (b) AIME24, (c) GPQA, and (d) BoardGameQA. For each $K$, we report the fraction of rollout-selected pairs whose preference direction matches an 8B reference using 16 rollouts. We use the base model as $p_{\mathrm{roll}}$ for AIME24 and GPQA.

Table 2: Ablation results using Qwen-2.5-1.5B. For the high-level SFT ablation, all methods use the same low-level SFT. For the hierarchical-level ablation, all variants share the same trained model. Our approach is highlighted in bold.

| Method | MATH500 Pass@1 | AIME24 Pass@1 | Cons@32 | Avg. Pass@1 |
|---|---|---|---|---|
| *Ablation of high-level SFT strategies* | | | | |
| **SFT (low) + LoRA (high)** | **62.0** ±1.2 | **8.9** ±1.4 | **13.3** ±0.4 | **35.5** |
| Base + LoRA (high) | 56.4 ±1.5 | 4.1 ±1.1 | 9.2 ±0.7 | 30.3 |
| SFT (high) | 59.8 ±1.3 | 6.5 ±1.2 | 11.0 ±0.5 | 33.2 |
| *Ablation of hierarchical levels* | | | | |
| **High-level + Low-level** | **86.1** ±1.0 | **31.2** ±1.0 | **47.4** ±0.4 | **58.7** |
| High-level only | 80.0 ±1.3 | 18.4 ±2.0 | 30.5 ±0.8 | 49.2 |
| Low-level only | 84.2 ±1.1 | 27.1 ±1.8 | 41.0 ±0.6 | 55.7 |

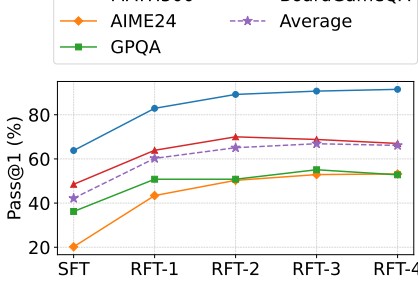

Figure 9: Performance of MLR over different training stages.

Table 3: Ablation results for the core components of MLR using LLaMA-3.1-8B.

| | Math | | | Science | Logic | |
|---|---|---|---|---|---|---|
| Method | MATH500 Pass@1 | AIME24 Pass@1 | Cons@32 | GPQA (Diamond) Pass@1 | BoardGameQA (Hard) Pass@1 | Avg. Pass@1 |
| Ours | **91.5** ±1.3 | **53.2** ±2.0 | **73.3** ±0.4 | **52.8** ±1.5 | **67.0** ±1.4 | **66.1** |
| DPO-only | 78.2 ±1.4 | 38.1 ±1.9 | 57.0 ±0.5 | 46.0 ±1.6 | 59.0 ±1.6 | 55.3 |
| Low-level policy + Step-DPO | 82.4 ±1.3 | 42.6 ±2.0 | 61.2 ±0.5 | 49.2 ±1.5 | 62.1 ±1.4 | 59.1 |
| Low-level policy + DPO | 74.1 ±1.5 | 32.4 ±1.8 | 52.0 ±0.6 | 44.0 ±1.7 | 56.0 ±1.7 | 51.6 |
| SFT-only | 63.8 ±1.2 | 20.2 ±2.0 | 36.7 ±0.4 | 36.2 ±1.8 | 48.5 ±1.8 | 42.2 |

stages. We compare instruction tuning, SFT distillation (vanilla long CoTs), preference optimization (DPO/Step-DPO), outcome-reward RL (GRPO), Plan-and-Solve, and MLR. Across benchmarks, MLR consistently outperforms all baselines, with its structured design enabling more effective reasoning on complex, long-horizon tasks. In addition, the iterative step-DPO procedure yields substantial gains over the SFT model. We also report response-length statistics: high-level trajectories are about 10–20% the length of low-level ones (Figure 11).

**Parameter studies.** We vary the rollout count $K$ and compare utilities from the 1.5B rollout policy against a stronger 8B reference using 16 rollouts (Figure 15). As expected, increasing $K$ reduces estimator variance, though at the cost of higher computation. To mitigate this overhead, we introduce a margin threshold $\delta$ when selecting preference pairs. We further measure the agreement of the selected preference pairs as a function of $K$, defined as the fraction whose preference direction agrees with the base model using 16 rollouts (Figure 8). Finally, we report model performance across training stages under different values of $K$ (Figure 12). Our default setting attains comparable final accuracy with lower rollout cost. Implementation details are provided in Section C.

**Ablation studies.** We ablate key components of MLR. We compare five configurations: (i) the full method, (ii) MLR trained with DPO instead of Step-DPO, (iii) low-level only with Step-DPO, (iv) low-level only with DPO, and (v) MLR with SFT only. Table 3 summarizes the results, which show that both multi-level modeling and step-level preferences contribute to performance. Figure 10 further reports preference accuracy over training. We additionally ablate the high-level SFT component (Table 2), evaluating two alternatives: (i) applying LoRA to the original base model and (ii) full-parameter SFT. A detailed discussion and implementation details are provided in Section B. We also ablate the hierarchical structure (Table 2), comparing (i) high-level–only and (ii) low-level–only variants. Further analysis and implementation details appear in Section C. Across all ablations, our full strategy yields the strongest performance.

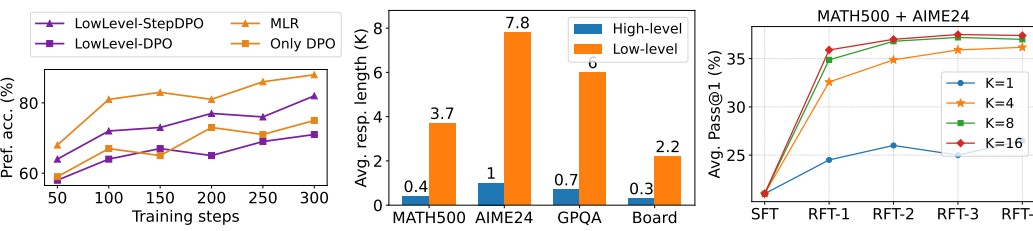

Figure 10: Preference accuracy.  Figure 11: Avg. resp. length.  Figure 12: Effect of $K$ in MLR.

## 5 RELATED WORK

**SFT-based reasoning distillation.** Reasoning models (OpenAI, 2024; Qwen-Team, 2024; Guo et al., 2025) have motivated extensive interest in replicating strong long-CoT performance. Several follow-up efforts primarily rely on SFT-based distillation, training smaller models to imitate long-CoT traces generated by stronger reasoning models (HuggingFace, 2025; Muennighoff et al., 2025). This line of work emphasizes data collection and curation to obtain high-quality reasoning trajectories, and has shown promising results in math and coding domains. However, SFT optimizes next-token imitation and thus passively follows demonstrations, providing limited exploration and limited credit assignment for long trajectories, where early mistakes can be amplified downstream.

**Outcome-reward RL for reasoning.** To go beyond imitation, recent RL-based pipelines optimize LMs using outcome rewards, with methods broadly falling into PPO-, GRPO-, and REINFORCE-based families (Schulman et al., 2017; Shao et al., 2024; Williams, 1992). PPO variants improve value learning and stability for long CoTs (e.g., value pretraining, modified advantage estimation) (Yuan et al., 2025; Yue et al., 2025). GRPO-style methods remove the critic and refine sampling/normalization/token-level objectives to improve stability at scale (Yu et al., 2025). REINFORCE variants reduce variance via KL penalties and centralized rewards (Hu, 2025; Kimi-Team et al., 2025). Across these methods, reward design typically combines outcome correctness with heuristic constraints such as format compliance and length control (Zhang et al., 2025a), and training often benefits from sampling heuristics such as curriculum-style difficulty scheduling (Zhang et al., 2025b) and rejection/sample filtering (Yu et al., 2025; He et al., 2025). Despite these advances, most pipelines still rely primarily on sparse outcome rewards, which remain weakly informative for long-horizon credit assignment.

**Process supervision and preference optimization.** Process-level supervision has been explored via process reward models (PRMs), which attempt to assign intermediate rewards to reasoning steps (Lightman et al., 2023; Wang et al., 2023b; Xiong et al., 2025). In practice, PRMs face several challenges: explicitly defining fine-grained steps is non-trivial; verifying intermediate correctness reliably is difficult; and training/maintaining a separate reward model introduces additional cost and risks (e.g., reward hacking) (Guo et al., 2025; Xiong et al., 2026). Preference optimization provides an alternative to explicit reward modeling. DPO learns from offline, trajectory-level preference pairs (Rafailov et al., 2023), while Step-DPO constructs step-wise preferences to better support long-chain reasoning (Lai et al., 2024). Online AI feedback can further enable preference-based updates beyond static datasets (Guo et al., 2024a). However, scaling to long CoTs remains challenging: constructing reliable step-level preferences often depends on strong teacher models or verifiers, which is costly and can be noisy on harder problems. This motivates scalable mechanisms for generating step-wise preferences without heavy reliance on external teachers. In this paper, we introduce a scalable TSMC-based approach to produce step-wise preferences for long-horizon supervision.

**Planning for long-horizon reasoning.** Beyond training signals, long-horizon reasoning with a single policy can suffer from plan drift and cascading failures. Planning-style prompting methods often follow a plan-then-execute paradigm, generating a full plan upfront and executing it as written (Xu et al., 2023; Wang et al., 2023a), which can propagate early errors when subtasks fail or new information emerges. In contrast, our work proposes multi-level reasoning (MLR), where a learned planner adapts its plan based on execution feedback, enabling mid-course revisions and improved robustness.

## 6 CONCLUSION

We introduced Multi-Level Reasoning (MLR), a two-level formulation for long-CoT generation that alternates between high-level step descriptors and low-level detailed reasoning, enabling scalable multi-policy modeling and adaptive planning over long horizons. To provide process-level training signals without a separate reward model, we proposed an iterative Step-DPO pipeline with TSMC-based preference construction. Across math, science, and logical reasoning benchmarks, MLR consistently improves over strong distillation, outcome-reward RL, and stepwise preference baselines, and degrades more slowly under long-horizon evaluation. Future work will explore extending MLR to tool-augmented settings, where external tools or knowledge are integrated into the reasoning process.

ACKNOWLEDGMENTS

This work is supported in part by DARPA SciFy program, Award No.HR001125C0302, and CISCO Systems, Inc.

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

## A  DATASET OVERVIEW

In this section, we summarize dataset statistics and split protocols for all benchmarks used in our study: MATH500 (Hendrycks et al., 2021; Lightman et al., 2023) and AIME24 (MAA, 2024) for math reasoning, GPQA-Diamond (Rein et al., 2023) for science reasoning, and BoardGameQA-Hard (Kazemi et al., 2023) for logical reasoning.

For each benchmark, we form a training pool and an evaluation set. The training pool is split into two disjoint subsets for supervised fine-tuning (SFT) and iterative preference learning. Evaluation examples are *strictly excluded* from SFT, preference construction, and rollouts, and are used only for final evaluation.

The MATH dataset (Hendrycks et al., 2021) contains 7,500 training and 5,000 test problems. MATH500 is a 500-problem subset of the MATH test set introduced by Lightman et al. (2023). We evaluate on MATH500 and add the remaining MATH test problems to the training pool; the MATH500 subset is excluded from SFT, preference learning, and rollouts.

The AIME dataset (MAA, 2024) is based on the American Invitational Mathematics Examination (AIME), a high-level mathematics contest administered by the Mathematical Association of America (MAA). We use a public AIME problem collection (1983–2024)[4], where each year typically contains 15 problems (before 2000) and 30 problems (afterwards). We use AIME 2024 as the evaluation set (AIME24), and all prior years as the training pool.

GPQA (Rein et al., 2023) is a multiple-choice QA dataset of difficult expert-written questions in biology, physics, and chemistry. It includes three subsets: Main (448 questions), Diamond (198 questions), and Extended (546 questions). We evaluate on the Diamond subset and combine the remaining subsets as the training pool. To prevent leakage, we remove any training examples that overlap with the evaluation set.

BoardGameQA (Kazemi et al., 2023) is a claim verification dataset with three labels (`proved`, `disproved`, `unknown`) that requires reasoning with conflicting information under preferences over rules. The dataset is divided into 15 subsets based on reasoning depth and levels of conflict or distractors, each with separate train/validation/test splits. We define BoardGameQA-Hard as the union of test splits from five challenging subsets: Main-depth3, DifficultConflict-depth2, HighConflict-depth2, KnowledgeHeavy-depth2, and ManyDistractors-depth2, yielding 500 evaluation examples. All remaining data from the benchmark are used as the training pool.

## B  IMPLEMENTATION DETAILS

**Dataset construction.**  We consider four publicly available reasoning benchmarks: MATH500, AIME24, GPQA-Diamond, and BoardGameQA-Hard. For each benchmark, we form a training pool and an evaluation set following Section A. The training pool is split into two disjoint subsets for supervised fine-tuning (SFT) and iterative preference learning (60%/40%). Evaluation examples are strictly excluded from SFT, preference construction, and rollouts, and are used only for final evaluation. In particular, for MATH we evaluate on MATH500 and add the remaining MATH test problems to the training pool, while excluding the MATH500 subset from all training components (SFT, preference learning, and rollouts).

To increase the diversity of training prompts for AIME and GPQA, we additionally generate synthetic problems using GPT-4o. For each seed problem, we sample multiple candidate questions and retain only those whose final answers are consistent across GPT-4o and DeepSeek-R1, i.e., both models solve the problem and produce the same final answer. We also remove any synthetic problems that overlap with the evaluation sets to prevent leakage. After constructing the training pools, the resulting SFT subsets contain 7.2K MATH, 4.0K AIME, 4.0K GPQA, and 5.0K BoardGameQA problems, and the preference-learning subsets contain 4.8K, 1.5K, 1.5K, and 2.0K problems, respectively.

All methods that we train ourselves (DPO, Step-DPO, GRPO, Plan-and-Solve, MLR, and all ablations in Table 3) use exactly the same data splits and budgets, ensuring a fair comparison. Results for external baselines (Instruct, SimpleRL, and DeepSeek-R1-Distill), marked with an asterisk in Table 1,

---

[4]https://www.kaggle.com/datasets/hemishveeraboina/aime-problem-set-1983-2024

are copied from their official reports and may rely on different training corpora; we include them as reference points.

Specifically, in the SFT phase, we generate multiple trajectories for each training problem by sampling four solutions from DeepSeek-R1, yielding approximately 80K filtered trajectories in total. Each accepted trajectory is then segmented into step-level chunks using DeepSeek-V3 via in-context learning, with each step annotated by a step descriptor. To enable multi-level reasoning, we further distill each step into a concise high-level summary, again using DeepSeek-V3 with in-context learning. All prompt templates used in this pipeline are provided in Section C. Figure 7 presents the resulting length distributions across low-level trajectories, high-level trajectories, and summaries.

**Summarization.** We train a separate summarization model to produce the compressed summary $c'^{(m)}$ for each low-level step $c^{(m)}$, which serves as the planner's conditioning context in Equation (2). The summarizer is Qwen2.5-0.5B-Instruct, fine-tuned with full-parameter SFT using AdamW (cosine schedule with linear warmup) and peak learning rate $1 \times 10^{-5}$, and shared across all base models. We keep the summarizer frozen during iterative Step-DPO. The prompt template is provided in Section C, and the summary length distribution is shown in Figure 7.

**Supervised fine-tuning.** We first fine-tune the base LM on low-level trajectories, then freeze it and train a parameter-efficient LoRA adapter for high-level planning. This design is motivated by two considerations. First, low-level trajectories are substantially longer and require modeling fine-grained token-level dependencies, for which full-parameter updates are beneficial. Second, high-level trajectories are shorter and more abstract, making them well suited for adapter tuning while reducing interference with the low-level executor. Training the planner on top of a fixed executor also reduces distribution shift between the planner's conditioning context and the executor's behavior. From an optimization standpoint, LoRA benefits from a stronger backbone after low-level SFT and constrains the number of trainable parameters, which can mitigate overfitting on high-level data. Operationally, the final system consists of a single base model plus a small LoRA adapter (less than 2% additional parameters), keeping deployment overhead low.

**Hyperparameters.** We train each base model on 80K multi-level examples using the AdamW optimizer with a cosine learning-rate schedule and linear warm-up (5% of total steps). The batch size is set to 256, with a peak learning rate of $2 \times 10^{-5}$, and we truncate sequences to 8,192 tokens. Training is performed for 3 epochs. For the base LM (low-level policy), the AdamW optimizer is used with a peak learning rate of $2 \times 10^{-5}$, while the high-level LoRA module ($r = 16$, $\alpha = 32$, target_modules=[q_proj, k_proj, v_proj, o_proj], no bias) is trained with a higher rate of $1 \times 10^{-4}$ (dropout=0.1) to allow faster adaptation. We further verified that adding MLP projections (up_proj, down_proj, gate_proj) yields only marginal improvements while substantially increasing the number of trainable parameters.

**Ablation study.** To further validate our design choices, we compare against two alternative training strategies for the high-level policy, while keeping the low-level training unchanged. This is important because low-level modeling requires full-parameter updates due to its longer and more complex reasoning trajectories; LoRA is insufficient for this component. We consider:

(i) LoRA on the original (non-SFT) base model: We directly apply LoRA tuning on the unfined-tuned Qwen-2.5-1.5B base model using only high-level trajectories.

- Base: Qwen-2.5-1.5B
- LoRA: $r = 16$, $\alpha = 32$, target_modules=[q_proj, k_proj, v_proj, o_proj], no bias.
- Optimization: AdamW with a cosine learning-rate schedule and linear warmup, a peak learning rate of $1 \times 10^{-4}$ and a a a LoRA dropout of 0.1.

(ii) Full-parameter SFT on high-level trajectories: We train a separate base model using full SFT on only high-level trajectories.

- Base: Qwen-2.5-1.5B
- Optimization: AdamW with a cosine learning-rate schedule and linear warmup, a peak learning rate of $1 \times 10^{-5}$.

We evaluate both variants on MATH500 and AIME24. Table 2 summarizes the results. Our default configuration (full SFT on low-level trajectories followed by LoRA tuning on high-level abstractions) achieves the highest accuracy, particularly on the harder AIME tasks that require deeper multi-step planning. We also observe that applying LoRA on top of the SFT-enhanced base model substantially eases optimization and mitigates the overfitting issues that arise when fully fine-tuning a separate base model using only high-level trajectories.

**Monte Carlo rollout analysis**  We analyze Monte Carlo rollout behavior using R1-Distill-LLaMA-8B and R1-Distill-Qwen-1.5B. Hidden CoTs are segmented into steps using \n\n. Estimation accuracy measures the fraction of prefixes for which rollouts correctly determine whether the prefix can still lead to a correct final solution.

For each partial trajectory, we assign a ground-truth survival label $y \in \{0, 1\}$ using extensive Monte Carlo lookahead with the base model: $y = 1$ if at least one rollout from the prefix reaches a correct final answer (i.e., the prefix is survivable), and $y = 0$ otherwise.

Using the fast rollout model, we draw $K$ continuations from each prefix and compute the estimated survivability

$$\hat{g}_K = \frac{1}{K} \sum_{k=1}^{K} \mathbf{1}_{\text{correct}}^{(k)}.$$

We then predict a survival label

$$\hat{y}_K = \begin{cases} 1, & \hat{g}_K > 0, \\ 0, & \text{otherwise.} \end{cases}$$

A prediction is correct when $\hat{y}_K = y$. Estimated survival accuracy for a given $K$ is the proportion of prefixes where this prediction matches ground truth.

We present the results in Figure 6. R1-Distill-LLaMA-8B and R1-Distill-Qwen-1.5B exhibit highly similar accuracy across prefix steps. As expected, runtime scales with model size: R1-Distill-LLaMA-8B is substantially slower than R1-Distill-Qwen-1.5B. All measurements are obtained on a single A100 GPU using vLLM.

**Iterative Step-DPO.**  The reference model for KL regularization in Step-DPO is the corresponding SFT checkpoint. We perform 4 training rounds, with approximately 3K prompts per round, as described in the main text. For each prompt, we sample $N = 4$ candidate continuations using temperature 0.7 and top-$p = 0.9$, and truncate continuations at 8,192 tokens to match the SFT context length and keep attention computation within our memory budget. Stepwise preference pairs are constructed from these candidates and stored in a replay buffer of size 50K; once the buffer is full, older entries are discarded in FIFO order. We optimize the policy with AdamW (learning rate $1 \times 10^{-5}$, weight decay 0.1, $\beta_1 = 0.9$, $\beta_2 = 0.95$), a global batch size of 256 preference pairs, gradient clipping with a maximum norm of 1.0, DPO temperature $\beta = 0.1$, and KL coefficient $\lambda_{\text{KL}} = 0.02$. At

---

**Algorithm 1:** Multi-Level Inference

1 **Inputs:** query $q$, high-level policy $\pi_{\theta_H}$, low-level policy $\pi_{\theta_L}$, summarizer $\pi_{\theta_S}$;
2 **Hyperparameter:** max steps $M$;
3 $m \leftarrow 1$;
4 **while** $m \leq M$ **do**
5 $\quad d^{(m)} \sim \pi_{\theta_H}(d \mid q, d^{(1:m-1)}, c'^{(1:m-1)})$;
6 $\quad c^{(m)} \sim \pi_{\theta_L}(c \mid q, d^{(1:m)}, c^{(1:m-1)})$;
7 $\quad c'^{(m)} \leftarrow \pi_{\theta_S}(d^{(m)}, c^{(m)})$;
8 $\quad$ **if** StopCriterion$(d^{(m)}, c^{(m)})$ **then**
9 $\quad\quad$ **break**;
10 $\quad m \leftarrow m + 1$;
11 **return** $(d^{(1:m)}, c^{(1:m)})$;

---

each round, we perform one epoch of updates over the current replay buffer. During optimization, we maintain a low-level policy (the base LM) and a high-level policy (the LoRA adapter). For low-level preference pairs, we disable the LoRA adapter and update only the base LM parameters; for high-level pairs, we freeze the base LM and update only the LoRA parameters. Mini-batches of low- and high-level examples are interleaved within each round, so that the executor and planner are optimized jointly while remaining modular. To improve sample efficiency, we apply a dynamic dropout strategy that filters "easy" prefixes, i.e., prefixes for which all candidates induce the same

---

**Algorithm 2:** Iterative Step-DPO

---

1 **Inputs:** Low-level policy $\pi_{\theta_L}$, high-level policy $\pi_{\theta_H}$; Reference models $\pi_{\text{ref}}^L$, $\pi_{\text{ref}}^H$; Fast rollout policy $\pi_{\text{roll}}$; RL prompts $\mathcal{D}_{\text{RL}}$.

2 **Hyperparams:** rounds $T$, prompts per round $N$, sample steps per prompt $M_s$, rollout count $K$, epochs $E$.

3 **for** $t = 1$ **to** $T$ **do**

4      Sample prompts $\{q_i\}_{i=1}^N \subset \mathcal{D}_{\text{RL}}$;

5      Initialize buffers $\mathcal{D}_{\text{pref-L}}^{(t)} \leftarrow \emptyset$, $\mathcal{D}_{\text{pref-H}}^{(t)} \leftarrow \emptyset$;

6      **foreach** $q$ **do**

7          $(\text{prefix}_H^{(t)}, \text{prefix}_L^{(t)}) \leftarrow \text{GENERATEPREFIXES}(\pi_{\theta_H}^{(t)}, \pi_{\theta_L}^{(t)}, q)$;

8          Randomly select a subset of steps $\mathcal{M}$ (size $M_s$) for evaluation;

9          **foreach** $m \in \mathcal{M}$ **do**

10              $\mathcal{D}_{\text{pref-L}}^{(t,m)} \leftarrow \text{COLLECTPAIR}(\pi_{\theta_L}^{(t)}, \text{prefix}_L^{(t)}[m])$;

11              $\mathcal{D}_{\text{pref-H}}^{(t,m)} \leftarrow \text{COLLECTPAIR}(\pi_{\theta_H}^{(t)}, \text{prefix}_H^{(t)}[m])$;

12              $\mathcal{D}_{\text{pref-L}}^{(t)} \leftarrow \mathcal{D}_{\text{pref-L}}^{(t)} \cup \mathcal{D}_{\text{pref-L}}^{(t,m)}$;

13              $\mathcal{D}_{\text{pref-H}}^{(t)} \leftarrow \mathcal{D}_{\text{pref-H}}^{(t)} \cup \mathcal{D}_{\text{pref-H}}^{(t,m)}$;

14      **if** $t > 1$ **then**

15          $\pi_{\text{ref}}^L \leftarrow \pi_{\theta_L}^{(t-1)}$; $\pi_{\text{ref}}^H \leftarrow \pi_{\theta_H}^{(t-1)}$;

16      **for** $e = 1$ **to** $E$ **do**

17          $\text{STEPDPOUPDATE}(\pi_{\theta_L}^{(t)}, \pi_{\text{ref}}^L, \mathcal{D}_{\text{pref-L}}^{(t)})$;

18          $\text{STEPDPOUPDATE}(\pi_{\theta_H}^{(t)}, \pi_{\text{ref}}^H, \mathcal{D}_{\text{pref-H}}^{(t)})$;

19 **return** $\pi_{\theta_L}^{(T)}, \pi_{\theta_H}^{(T)}$;

---

utility; the dropout rate increases linearly from $0.1$ to $0.9$ over training. All experiments are conducted on $4\times$ A100 GPUs (80GB) with `bf16` precision.

**Step-DPO update schemes.** We compare the proposed update scheme against alternatives under a matched training budget (same number of prompts, candidates, and optimization steps). In the *planner-only* variant, we freeze the SFT base LM and apply Step-DPO updates only to the high-level LoRA adapter for all preference pairs, thereby testing whether adapting the planner alone is sufficient once the executor has been trained. In a *round-based* variant, we first run Step-DPO for two rounds updating only the low-level policy (LoRA disabled), and then for two rounds updating only the high-level LoRA (base LM frozen), mirroring a coarse low-then-high schedule in the iterative phase. Empirically, our joint modular scheme, which interleaves low-level and high-level updates while restricting each preference type to its corresponding module, achieves the best overall performance on MATH500 and AIME24, suggesting that simultaneously refining the executor and planner, while keeping their parameter updates disentangled, is more effective than tuning either component in isolation.

**DPO baseline.** To isolate the effect of step-wise supervision, we train a standard outcome-level DPO baseline on the same prompt pool and with the same rollout configuration as Step-DPO. The reference model for KL regularization is the corresponding SFT checkpoint, and we run 4 training rounds with approximately 3K prompts per round. For each prompt, we sample $N = 4$ candidate continuations using temperature $0.7$ and top-$p = 0.9$, truncating each continuation at 8,192 tokens to match the SFT context length. Preference pairs are constructed at the trajectory level: we assign each candidate a scalar utility based on its final solution correctness and form DPO pairs from these outcome-level utilities, ignoring intermediate prefixes. The resulting preference pairs are stored in a replay buffer of size 50K with FIFO eviction, and we perform one epoch of DPO updates over the buffer per round. We optimize a single policy (no hierarchical separation) with AdamW (learning rate $1 \times 10^{-5}$, weight decay 0.1, $\beta_1 = 0.9$, $\beta_2 = 0.95$), using a global batch size of 256 preference pairs, gradient clipping with a maximum norm of 1.0, DPO temperature $\beta = 0.1$, and KL coefficient

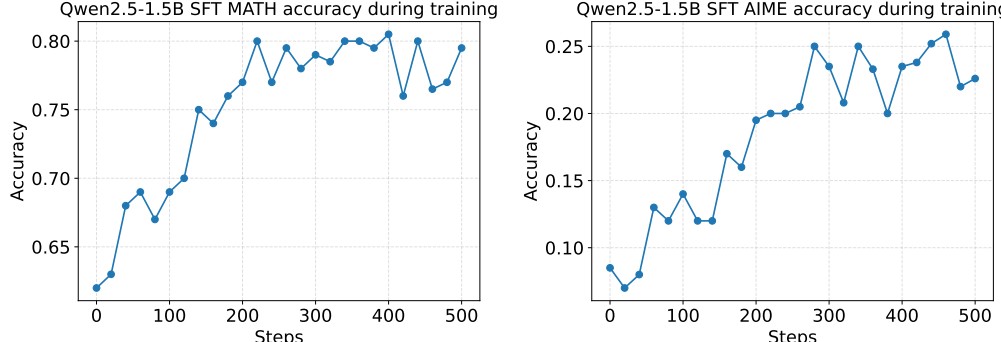

Figure 13: Validation accuracy of Qwen2.5-1.5B SFT during GRPO training. For each question, we sample 8 responses and calculate the overall average accuracy to ensure a stable evaluation.

$\lambda_{\mathrm{KL}} = 0.02$. All experiments are conducted on $4\times$ A100 GPUs (80GB) with `bf16` precision under a matched training budget to Step-DPO.

**GRPO baseline.** To compare MLR with a standard single-policy preference-optimization method, we train a GRPO baseline on the same prompt pool as Step-DPO. We first construct a *single-policy* SFT checkpoint by fine-tuning Qwen-2.5-1.5B on the processed low-level trajectories in which all step descriptors are removed from both inputs and targets, using the same optimizer, schedule, and token budget as our low-level SFT. Starting from this checkpoint, we apply full-parameter GRPO, keeping a frozen copy of the SFT model as the reference policy. We implement the baseline using the Verl (Sheng et al., 2024) framework and vLLM (Kwon et al., 2023) as the rollout backend. The actor and reference are both initialized from the same SFT checkpoint. For each prompt, we sample groups of $N = 4$ candidate continuations with temperature $0.7$ and top-$p = 0.9$, cap the maximum response length at 4,096 tokens to respect GPU memory limits, and assign a rule-based outcome reward of $1$ if the final answer is correct and $0$ otherwise. We optimize the actor with AdamW (learning rate $5 \times 10^{-7}$, weight decay $0.1$) under a KL-penalty objective with coefficient $\lambda_{\mathrm{KL}} = 0.02$, using `bf16` precision, gradient checkpointing, and FlashAttention (Dao, 2023) on $4\times$ A100 GPUs (80GB). We train for 4 epochs, using a global batch size of 32 (PPO mini-batch sizes 16, micro-batch sizes 2, respectively), and evaluate every 100 steps on the held-out validation split, selecting the checkpoint with the best validation Pass@1.

We visualize the validation accuracy of Qwen2.5-1.5B SFT during GRPO training in Figure 13. For each question, we sample 8 responses and report the average accuracy to obtain a stable estimate. The evolution of the average response length during GRPO is shown in Figure 14, and the final evaluation results are summarized in Table 1. Compared with our strategy, GRPO is less efficient for

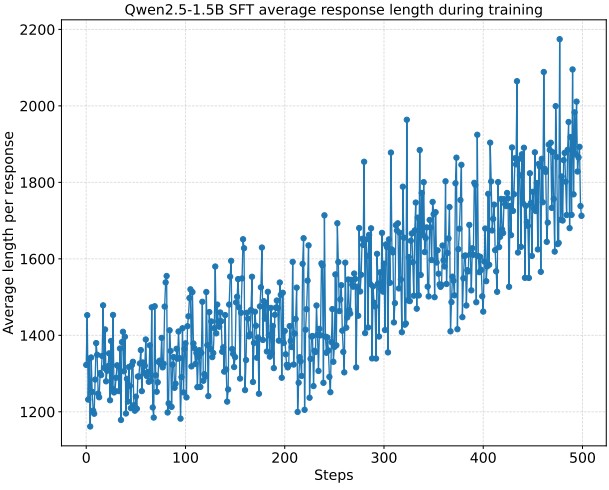

Figure 14: The average response length of Qwen2.5-1.5B SFT on the training set during GRPO.

long-horizon reasoning: outcome rewards are (i) *sparse*: for long trajectories, a single scalar signal is often insufficient to localize errors; and (ii) *computationally expensive*: generating full rollouts requires substantial memory and compute. When starting from fine-tuned models with long CoTs, these costs limit our ability to apply GRPO to larger base models. In contrast, our iterative Step-DPO procedure is easier to implement and control, and provides a more practical alternative for finetuning long-horizon reasoning policies.

We repeat the above protocol on a 7B model. We first obtain a *single-policy* SFT checkpoint by fine-tuning Qwen-2.5-MATH-7B on the same processed low-level trajectories (step descriptors removed from inputs and targets). Starting from this checkpoint, we run full-parameter GRPO with a frozen copy of the SFT model as the reference. Training uses FSDP sharding, `bf16` precision, activation checkpointing, and FlashAttention on $8\times$A100 (80GB), with a global batch size of 32 implemented as a micro-batch of 1 per GPU and 4 gradient-accumulation steps. Rollouts are generated with vLLM: for each prompt we sample $N{=}4$ candidates (temperature 0.7, top-$p{=}0.9$) and cap the maximum response length at 4,096 tokens. We optimize with AdamW (learning rate $5 \times 10^{-7}$, weight decay 0.1) under a KL penalty with coefficient $\lambda_{\mathrm{KL}}{=}0.02$, and evaluate every 100 steps, selecting the checkpoint with the best validation Pass@1.

We follow the same procedure with Llama-3.1-8B. A *single-policy* SFT checkpoint is first obtained on the same low-level trajectories, after which we apply full-parameter GRPO using a frozen reference initialized from the SFT checkpoint. We train with FSDP, `bf16`, activation checkpointing, and FlashAttention on $8\times$A100 (80GB), using a global batch size of 32 realized as a micro-batch of 1 per GPU and 4 gradient-accumulation steps. Rollouts use vLLM with $N{=}4$ candidates per prompt (temperature 0.7, top-$p{=}0.9$) and a 4,096-token cap. The optimizer, KL objective, evaluation cadence, and model selection criteria are identical to the 7B setting. For additional memory headroom, the frozen reference is sharded; when necessary, we load the reference in 8-bit for forward-only KL without changing any other hyperparameters.

**Plan-and-Solve baseline.** We compare against Plan-and-Solve (Wang et al., 2023a), which first proposes a concise, global plan and then executes the solution conditioned on that plan. The example prompt is given in Section C. For a fair comparison, we use the same backbone as our method and fine-tune two LoRA heads on top of it: a *planner* (Problem $\rightarrow$ Plan) and an *executor* (Problem $+$ Plan $\rightarrow$ Solution). At inference we follow the standard two-pass Plan-and-Solve pipeline: Pass-1 generates the plan; Pass-2 solves the problem conditioned on that plan.

**Training data creation.** Using the same training set as our method, we prompt a strong teacher model (DeepSeek-V3.2) to produce corresponding trajectories. We filter trajectories by final-answer correctness and basic format checks. We match the total number of accepted trajectories to our method (80K) to ensure a fair comparison.

**Training configuration.** Unless otherwise noted, we freeze the backbone and train LoRA adapters with identical hyperparameters for planner and executor.

- Backbone: Qwen-2.5-1.5B. LoRA: $r = 16$, $\alpha = 32$, target_modules=[q_proj, k_proj, v_proj, o_proj], no bias. Optimization: AdamW, cosine decay with $3\%$ warm-up, learning rate $1 \times 10^{-4}$.

- Backbone: Qwen-2.5-MATH-7B. Same LoRA configuration. Same optimization configuration except for learning rate $5 \times 10^{-5}$.

- Backbone: Llama-3.1-8B. Same LoRA configuration. Same optimization configuration except for learning rate $5 \times 10^{-5}$.

**Results.** Table 1 summarizes performance. Because Plan-and-Solve here is trained only with SFT, we compare it against MLR (SFT-only). Across all three backbones, our method outperforms Plan-and-Solve, with the largest margins on the harder benchmarks (AIME, GPQA). We observe that Plan-and-Solve often implicitly assumes all subtasks succeed as initially planned; errors in early steps can propagate, and the executor may partially deviate from the plan. In contrast, our approach learns a better planner that can adapt its plans based on execution signals, enabling revisions rather than committing to a fixed blueprint. This adaptive coupling between planner and executor yields more stable long-horizon reasoning than prompting a plan upfront and executing it verbatim.

**Evaluation.** During evaluation, we use greedy decoding for both the base model and the instruction fine-tuned model to produce more coherent and consistent CoTs. For all other baselines and our method, we follow the decoding protocol in Guo et al. (2025), using sampling-based decoding with a temperature of $0.6$ and a top-$p$ value of $0.95$ to generate 8 responses per prompt to reduce variance and repetition. For MLR, we employ a single base LM for both levels and switch the high-level LoRA adapter on or off depending on the generation stage (Algorithm 1). Specifically, we enable the high-level LoRA adapter to produce step descriptors (planning), and then disable the adapter to generate the corresponding low-level trajectories conditioned on these descriptors. The maximum generation length for all models is set to 16,384 tokens. Performance is measured using Pass@1 $= \frac{1}{k} \sum_{i=1}^{k} p_i$, where $p_i$ denotes the correctness of the $i$-th response. For AIME24, we also report consensus accuracy over 32 samples, denoted as cons@32.

## C  ADDITIONAL RESULTS

**Examples of MLR outputs.** In this section, we present additional results to further demonstrate and analyze the effectiveness of our method. We showcase representative output examples generated by MLR across different datasets (Section C). Each sample consists of a two-level reasoning trajectory, comprising shared reasoning steps annotated with both a step descriptor and corresponding step content. In the high-level trajectory, the step descriptor is generated by the high-level module, while the step content is produced by the compressor, which takes the low-level content as input and outputs a concise abstraction. In the low-level trajectory, the step descriptor is provided by the high-level module, and the step content is directly generated by the low-level base model.

**Error analysis.** To better understand the strengths and limitations of our framework, we conduct detailed error analysis. To further enhance verification and error localization, we incorporate auxiliary models (OpenAI's o1 and o1-mini) to assist in identifying potential reasoning flaws. Specifically, we first evaluate whether the auxiliary model can independently solve each task without access to the ground-truth final answer or reference solution. If the auxiliary model successfully produces the correct solution, we then use it to help analyze erroneous trajectories generated by our framework. The error analysis provided by the auxiliary model is subsequently reviewed and confirmed by human evaluators. Through this process, we identify several recurring error patterns: 1) High-level step descriptor errors: redundant branching (multiple step descriptors that pursue the same subtask), unclosed loops (steps are never marked as "complete," leading to repeated revisitation), dead-end retention (contradicted or unproductive exploratory branches are retained), copy-pasted fallback (guessed answers are repeated verbatim under different step descriptors). 2) Low-level step content errors: logical misapplication (misuse of domain-specific rules or principles), contradiction tolerance (inconsistent constraints are not resolved), repetitive reasoning (redundant inference chains without new contributions), failure to propagate known facts (previously inferred information is ignored in later steps), looping filler (verbose or stalled reasoning with redundant rephrasing).

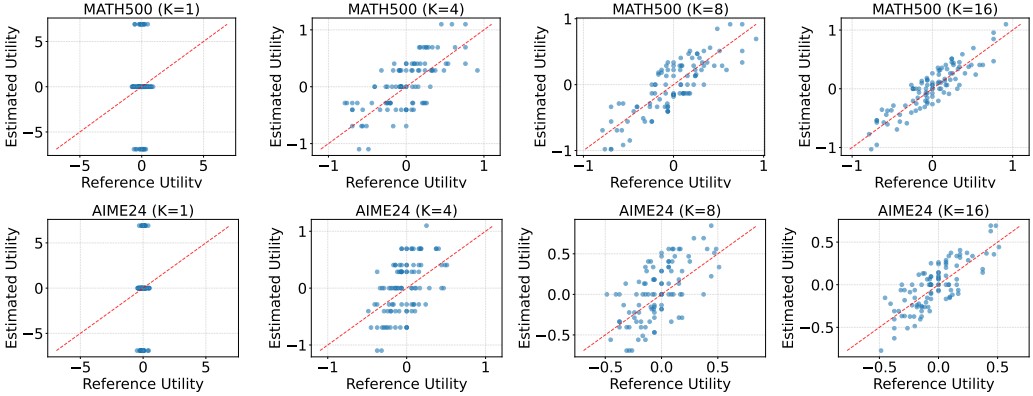

Figure 15: Effect of rollout count on the reliability of utility estimates. We compare utilities estimated by the 1.5B model under $K$ rollouts to reference utilities by the 8B model using 16 rollouts for 100 sampled prefixes from MATH500 and AIME24, respectively.

**Parameter studies on rollout count.**    In our iterative step-DPO, the rollout count $K$ directly affects the quality of the preference pairs. We first examine how $K$ influences the reliability of the utility estimates defined in Equation (6). To do so, we compare utilities estimated by the 1.5B model under various $K$ to reference utilities produced by the 8B model with 16 rollouts, using 100 sampled prefixes from MATH500 and AIME24, respectively (Figure 15). As expected, larger $K$ reduces estimator variance but increases computational cost. To control this overhead, we apply a margin threshold $\delta$ when selecting preference pairs which allows us to use smaller $K$ while maintaining reliability of the preference data. Next, we evaluate the agreement of selected preference pairs as a function of $K$, defined as the fraction of pairs whose preference direction agrees with the base model using 16 rollouts (Figure 8). For each $K$, we generate 100 preference pairs following Equation (7). We then recompute the reference utilities of both options and check whether the chosen response has higher reference utility than its alternative. Pairs that satisfy this condition are counted as agreed, and we report the average agreement for each $K$. We consider both the 1.5B model and the 8B base model as rollout policy. In practice, we use the base model as the rollout policy for AIME24 and GPQA. Finally, we study model performance across training stages under different rollout counts (Figure 12). Starting from the same SFT model, we generate the same number of preference pairs for each $K$ and all train for 4 epochs. We report performance on MATH500 and AIME24 throughout training. Overall, our setting achieves comparable accuracy while significantly reducing computational cost.

**Ablations on hierarchical levels.**    To investigate the role of different levels, we conduct an ablation study on the hierarchical structure. We consider two variants: (1) High-level only: the high-level module directly predicts summaries without invoking the low-level module; (2) Low-level only: the low-level module is required to predict both the high-level step descriptions and the detailed reasoning without guidance from the high-level module. The evaluation protocol matches our main setting, and the results are reported in Table 2. Our full method consistently outperforms both variants, especially on the challenging AIME24 dataset. The high-level-only variant underperforms because the planner lacks grounded execution learning, making direct summary prediction unreliable for difficult reasoning tasks. We show an erroneous example in Section C. The low-level-only variant is weaker because the absence of explicit high-level guidance causes the low-level module to drift and accumulate errors as the trajectory grows longer. Overall, these results demonstrate that our two-level design yields better performance on long-horizon reasoning tasks.

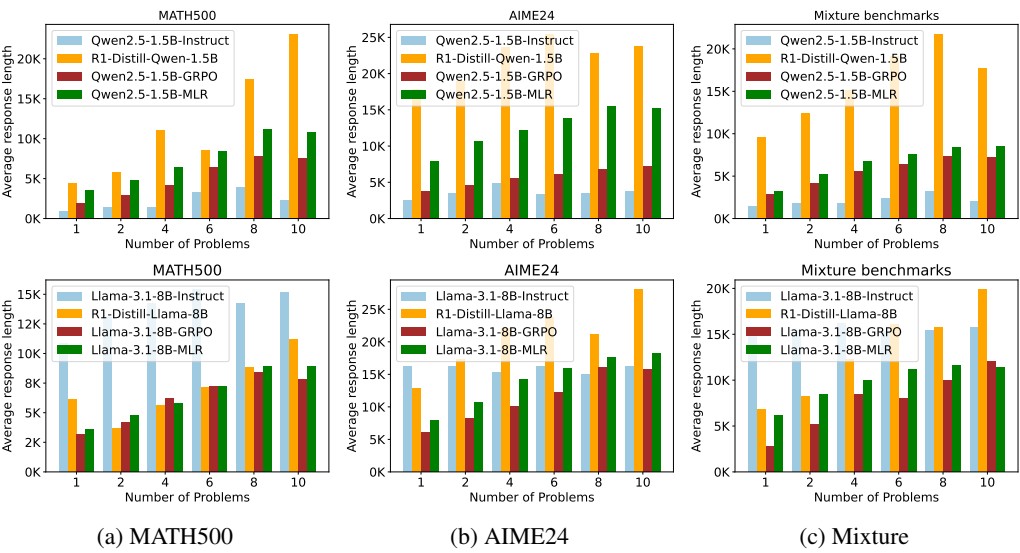

(a) MATH500          (b) AIME24          (c) Mixture

Figure 16: Long-horizon reasoning response length on (a) MATH500, (b) AIME24, and (c) Mixture benchmarks (MATH500, AIME24, GPQA, BoardGameQA). We simulate long-horizon reasoning by concatenating multiple problems in the prompt and report average accuracy. Compared with R1-Distill models, which are prone to getting lost in long trajectories and entering dead loops, MLR benefits from explicit planning and is able to solve all problems more effectively.

**Long-horizon reasoning test.** To further evaluate our method on long-horizon reasoning, we simulate a multi-question setting by concatenating multiple problems into a single prompt. We consider three configurations: (i) multiple problems from MATH500, (ii) multiple problems from AIME24, and (iii) mixed batches drawn from MATH500, AIME24, GPQA, and BoardGameQA. For each configuration, we randomly select 100 problems from the corresponding benchmarks and treat their questions as a mini-batch. The batch size varies from 1 to 10; for each size, we present all questions in the batch in one prompt and ask the LLM to solve them jointly. To mitigate cases where the model gets stuck on a single problem, we explicitly instruct it to skip a question if it cannot make progress. Prompt template can be found in Section C. For instruction-tuned models, we use greedy decoding. For the other baselines, for every mini-batch we sample 8 responses with temperature 0.6 and top_p = 0.95, using a maximum context length of 32K tokens, to reduce variance and then report accuracy averaged over the 100 problems. Note that all methods are trained only in the single-question setting, so this constitutes a fully out-of-distribution evaluation. Figure 1 shows the results: MLR consistently degrades more slowly than all baselines across datasets and base models, highlighting the benefits of explicit planning for long-horizon reasoning. We further report the corresponding length distributions for different scenarios in Figure 16.

---

**Example: Long-horizon Plan Failures (DeepSeek-R1-Distill-Llama-8B on MATH500)**

**Prompt:**
What is the least positive integer multiple of 30 that can be written with only the digits 0 and 2?

**Response:**
<think>
Okay, so I need to find the smallest positive integer multiple of 30 that can be written using only the digits 0 and 2. Hmm, let me think about how to approach this.

First, I remember that a multiple of 30 must be divisible by both 3 and 10. So, let's break down what that means.

Divisibility by 10: For a number to be divisible by 10, it must end with a 0. That makes sense because any number ending in 0 is a multiple of 10. So, whatever number I come up with must end with a 0. That means the last digit is fixed as 0.

Divisibility by 3: For a number to be divisible by 3, the sum of its digits must be a multiple of 3. Since the number can only have digits 0 and 2, the sum of the digits will just be the number of 2s multiplied by 2. So, the total sum is 2 times the number of 2s. This sum needs to be divisible by 3.

So, putting that together, I need a number that consists of one or more 2s followed by a 0. The number of 2s should be such that the total sum (which is 2 times the number of 2s) is divisible by 3.

Let me formalize that. Let's say the number has 'n' number of 2s followed by a 0. So, the number would be 222...20, with 'n' 2s and ending with a 0.

The sum of the digits is 2n, and we need 2n to be a multiple of 3. So, 2n = 0 mod 3. Which means n must be a multiple of 3 because 2 and 3 are coprime (they have no common divisors other than 1). So, n needs to be divisible by 3.

Therefore, the smallest number of 2s needed is 3, right? Because 3 is the smallest positive integer divisible by 3.

So, if n = 3, the number would be 2220. Let me check that.

2220 is a multiple of 30? Let's see. 2220 divided by 30 is 74, which is an integer. So, yes, 2220 is a multiple of 30.

But is it the smallest?

Wait, maybe I can find a smaller number with fewer digits? Let me think. The number must end with a 0, so the last digit is fixed. The other digits can be 0 or 2. But if I try to make a smaller number, maybe with fewer digits, but still only 0s and 2s.

Let me consider numbers of different lengths and see if they can satisfy the conditions.

Starting with the smallest possible number: 1 digit. It would have to be 0, but 0 is not positive. So, no good.

2 digits: The number would be 20. Is 20 a multiple of 30? 20 divided by 30 is less than 1, so no. Also, 20 is not a multiple of 30.

3 digits: Let's see. The possible numbers are 200, 220, and 222.

200: 200 divided by 30 is approximately 6.666, not an integer. 220 divided by 30 is about 7.333, not integer. 222 divided by 30 is 7.4, which isn't integer either. So, none of these are multiples of 30.

4 digits: Let's list possible numbers. They end with 0, so the last digit is 0, and the first three digits are either 0 or 2.

Possible numbers: 2000, 2200, 2220, 2222, 2020, 2022, 2202, 2220, 2222, etc. Wait, but actually, the first three digits can be any combination of 0s and 2s. So, all numbers of the form 2220, 2202, 2022, 0222, but 0222 is not a valid number since it starts with 0. So, the numbers are 2220, 2202, 2022.

Check each:

2220: As before, 2220 / 30 = 74, which is integer. So, that's a multiple.

2202: 2202 / 30 is 73.4, which isn't integer.

2022: 2022 / 30 is 67.4, which isn't integer.

So, the only 4−digit number that works is 2220.

But earlier, with 3 digits, we had 2220 as well. Wait, so 2220 is a 4−digit number, but 3−digit numbers didn't work. So, 2220 is the first one.

Wait, but hold on. Maybe there's a 3−digit number that's a multiple of 30, but I haven't considered all possibilities. Let me check again.

Wait, 2220 is a 4−digit number, but what about a 3−digit number? 222 is 222. 222 divided by 30 is 7.4, which isn't integer. 2220 is 74. So, maybe 2220 is the first one.

Wait, but hold on. Maybe a 3−digit number with two 2s and a 0. Let's see: 220, 202, 022. But 022 isn't a 3−digit number, it's just 22. So, 220 is 220, which is 220 / 30 ˜= 7.333, not integer. 202 is 202 / 30 ˜= 6.733, also not integer. So, no luck.

So, in the 3−digit numbers, there's no multiple of 30 that can be formed with 0s and 2s. So, moving on to 4−digit numbers.

Wait, but 2220 is a 4−digit number, but is that the smallest? Wait, let me think. Is there a smaller number with 4 digits, but starting with 2, and having 2s and 0s, but not necessarily 3 2s? Wait, no, because if you have 2 2s, then 220, 202, 022. None of these are divisible by 30.

So, the only 4−digit number is 2220, which is 2220. But wait, maybe a 5−digit number? No, 2220 is already a 4−digit number, which is smaller than a 5−digit number. So, 2220 is smaller.

But wait, hold on. Let me consider 2220. It's 2220, but is there a smaller number with 4 digits that can be formed with 0s and 2s? Wait, 2220 is 2220. But what about 2220, 2202, 2022, etc. Wait, none of the other permutations of 2s and 0s in 4 digits are multiples of 30. So, 2220 is the only one.

But wait, is 2220 the minimal? Because 2220 is a 4−digit number, but is there a 3−digit number that's a multiple of 30? Wait, I don't think so, because 2220 is the first multiple of 30 with only 2s and 0s.

Wait, hold on. Let me think differently. Maybe instead of 2220, there's a smaller number.

Wait, 2220 is 2220, but if I can find a number with fewer digits, like 220, but 220 is not a multiple of 30. 220 divided by 30 is 7.333. Not integer.

Wait, 202: 202 / 30 is 6.733. Not integer. 200: 200 / 30 is 6.666. Not integer. 222: 222 / 30 is 7.4. Not integer. So, none of these are multiples of 30.

Wait, so maybe 2220 is the first one. But 2220 is a 4−digit number. Is there a 3−digit number that can be formed with 2s and 0s that is a multiple of 30? Let me think.

Wait, 2220 is the first one. So, 2220 is 2220. But 2220 is 2220, which is 2220. But if I can write a 3−digit number with 2s and 0s that's a multiple of 30, that would be smaller than 2220.

Wait, 2220 is 4 digits, but 222 is 3 digits. But 222 is not a multiple of 30. So, no.

Wait, maybe 220 is 220, but that's 220, which is not a multiple of 30. 200 is 200, which is not a multiple of 30. 222 is 222, which is not a multiple of 30. So, yeah, no luck.

So, 2220 is the first one.

Wait, but hold on. Let me think again. If I can have a number that is 200...0, but with some 2s in between. But the sum of digits must be a multiple of 3, so as we saw earlier, the number of 2s must be a multiple of 3.

So, with 3 2s, we get 2220, which is 2220. But is there a way to have 2s in other positions? Like 2020, 2002, 2022, etc. But as we saw, those are not multiples of 30.

Wait, 2020: sum of digits is $2 + 0 + 2 + 0 = 4$, which is not a multiple of 3, so it's not divisible by 3, hence not divisible by 30. Similarly, 2002: $2 + 0 + 0 + 2 = 4$, same thing. 2200: $2 + 2 + 0 + 0 = 4$, same. 2220: $2 + 2 + 2 + 0 = 6$, which is a multiple of 3. So, 2220 is the first one.

So, perhaps 2220 is the minimal.

Wait, but hold on. Let me check if 2220 is indeed the minimal. Maybe a number with 4 digits but starting with 2, and with other digits as 2s and 0s, but the number of 2s is more than 3? But no, because 2220 is the first one where the number of 2s is 3, which is the minimal multiple of 3.

Wait, but maybe 2220 is the minimal, but is there a smaller number? Let me think.

Wait, 2220 is 2220, which is a 4−digit number. Is there a 3−digit number with 2s and 0s that is a multiple of 30? As we saw, 2220 is the first one. So, 2220 is the minimal.

Wait, but hold on. Maybe I can have a number like 2200, which is 2200, but the sum is 4, which isn't a multiple of 3. So, not divisible by 3. So, no.

Wait, 2202: sum is $2 + 2 + 0 + 2 = 6$, which is a multiple of 3, but 2202 divided by 30 is 73.4, which isn't integer. So, not a multiple.

Similarly, 2022: sum is $2 + 0 + 2 + 2 = 6$, which is a multiple of 3, but 2022 / 30 is 67.4, not integer. So, no.

2002: sum is 4, not multiple of 3.

2220: sum is 6, which is multiple of 3, and 2220 / 30 is 74, which is integer. So, that works.

Therefore, 2220 is the minimal number.

Wait, but hold on. Let me think if there's a smaller number.

Wait, 2220 is 2220, which is a 4−digit number. Is there a 3−digit number that can be formed with 2s and 0s, which is a multiple of 30? As we saw, 2220 is the first one, so no, there's no smaller number.

Wait, but hold on again. Let me think about 2220. Is it the minimal, or is there a smaller number with 4 digits but starting with 2, and having 2s and 0s, but arranged in a different way?

Wait, for example, 2020: sum is 4, which is not multiple of 3, so not divisible by 3. 2002: same. 2200: same. So, no.

So, 2220 is the minimal.

Wait, but hold on, let me think of another approach. Maybe think of numbers in base 10, with digits only 0 and 2, and divisible by 30.

So, the number must end with 0, and the sum of digits must be a multiple of 3.

So, as we saw, the number must have a number of 2s equal to 3, 6, 9, etc.

Therefore, the minimal number is 2220, which is 3 digits.

Wait, 2220 is 4 digits. So, 2220 is 2220, which is 4 digits, and it's the first one.

But wait, 2220 is a 4−digit number. Is there a 3−digit number? 2220 is 2220, which is 2220. But 222 is 3 digits, but 222 is not a multiple of 30.

Wait, so 2220 is the minimal.

Wait, but I just thought of another way. Maybe if we can have a number with a 0 at the end, and two 2s, but also have another 2 somewhere in the middle. But the sum of digits must be a multiple of 3, so $2 + 2 + 0 = 4$, which isn't a multiple of 3. So, that's not good. So, 2220 is the minimal.

Wait, but 2220 is a 4−digit number, but 2220 is 2220, which is 2220, which is 2220. Hmm, but 2220 is the minimal.

Wait, hold on, but is 2220 the minimal? Let me think of 20220, but that's a 5−digit number, which is larger than 2220, so not helpful.

Wait, so 2220 is 2220, which is 4 digits, and it's the minimal.

Wait, but hold on, maybe 20220 is a 5−digit number, but 2220 is smaller.

Wait, so 2220 is 2220, which is 4 digits, and it's smaller than 20220.

Therefore, 2220 is the minimal.

Wait, but now, hold on, another thought. Maybe a number with 2220 is 2220, but is there a number with 2220 in a different arrangement? For example, 20220 is a 5−digit number, but 2220 is 4 digits, so 2220 is smaller.

Therefore, 2220 is the minimal.

Wait, but let me just confirm once again. 2220 is 2220, which is 2220. Let me see if any number with fewer digits can be formed.

Wait, 2220 is 4 digits. 222 is 3 digits, but 222 is not a multiple of 30. 2220 is 4 digits, and it's a multiple of 30.

Wait, so 2220 is the minimal.

Wait, but wait, 20220 is a 5−digit number, which is 20220, which is 20220. But 2220 is smaller.

Wait, but hold on, 2220 is a 4−digit number, which is 2220, so 2220 is smaller than 20220, which is a 5−digit number.

Therefore, 2220 is the minimal.

Wait, but I just thought of another approach. Maybe think of the number as starting with 2, followed by 2s and 0s.

So, for example, 2220, 2202, 2022, 2002, 2220, etc. But as we saw, only 2220 is a multiple of 30.

Wait, but 2220 is a 4−digit number, which is 2220. 2220 is 2220, which is 2220, so it's a 4−digit number.

Wait, but let me think of the number 2220. Is that the minimal?

Wait, 2220 is 2220, which is 2220. So, 2220 is a multiple of 30, but is there a smaller number?

Wait, 2220 is 2220, which is 2220, which is 2220, which is 2220.

Wait, but 2220 is 2220, which is 2220, so is that the minimal? Or is there a number with fewer digits?

Wait, 2220 is 4 digits. 222 is 3 digits. 222 is not a multiple of 30. So, 2220 is the minimal.

Wait, but wait, another idea: 2220 is 2220, which is 2220, which is 2220, which is 2220.

Wait, but if I can have a number with 2s and 0s, but with leading 2s and trailing 0s, but in between, perhaps 2s can be placed in different positions.

Wait, but as we saw, the sum of digits must be a multiple of 3, so the number of 2s must be a multiple of 3.

Therefore, the minimal number is 2220, which is 2220.

Wait, so 2220 is 4 digits, and it's the minimal. So, 2220 is the answer.

Wait, but wait, I just thought of something else. What if the number is 20220? That's 5 digits, but it's 20220, which is 20220. 2220 is 2220, which is smaller. So, 2220 is smaller.

Wait, but 2220 is 2220, which is a 4−digit number. 20220 is 20220, which is a 5−digit number, so 2220 is smaller.

Therefore, 2220 is the minimal.

Wait, but I'm confused because 2220 is 2220, which is 2220. So, that's a 4−digit number, and it's the minimal.

Wait, but hold on, let me just confirm if 2220 is indeed the minimal.

Wait, 2220 is 2220, which is 2220. Let me check if 2220 is a multiple of 30. 2220 divided by 30 is 74, which is an integer. So, yes, it is.

So, 2220 is the minimal.

Wait, but hold on, 2220 is 2220, which is 2220, which is a 4−digit number.

But, is there a smaller number?

Wait, 2220 is 2220, which is 2220. But, 2220 is 2220, which is 2220. So, it's 2220, which is 2220. So, 2220 is the minimal.

Wait, but wait, 2220 is 2220, which is 2220, which is a 4−digit number. So, 2220 is 2220, which is 2220.

Wait, so that's the minimal.

Wait, but wait, hold on, 2220 is 2220, which is 2220, which is 2220. So, it's 2220.

Wait, so is 2220 the minimal.

Wait, but hold on, another thought. Let me think of numbers with 2s and 0s, but in different positions, but the sum of digits must be a multiple of 3.

So, 2220 is 2220, which is 2220, sum is 6, which is a multiple of 3.

Wait, but if I have a number like 20220, which is 5 digits, sum is 6, which is multiple of 3, but 20220 is 20220, which is larger than 2220.

So, 2220 is the minimal.

Wait, but hold on, another idea. Maybe 20220 is 20220, which is 20220, which is a 5−digit number, which is 20220, which is larger than 2220, which is 4−digit.

So, 2220 is smaller.

Wait, so 2220 is the minimal.

Wait, but, wait, let me think of the number 2220. It's 2220, which is 2220, which is 2220. So, 2220 is a 4−digit number.

But, 2220 is 2220, which is 2220, so that's 2220.

Wait, so, yeah, 2220 is the minimal.

Wait, but, hold on, I just thought of another number. Let's say 2220 is 2220, which is 2220. But, 2220 is 2220, which is 2220. So, is there a smaller number?

Wait, 2220 is 2220, which is 2220, which is 2220, which is 2220. So, 2220 is 2220, which is 2220. So, 2220 is 2220, which is 2220.

Wait, so, that's the minimal.

Wait, so, conclusion: 2220 is the minimal.

Wait, but hold on, I just thought of something else. Maybe a number like 2220 is 2220, but 2220 is 2220, which is 2220, which is 2220, which is 2220.

Wait, so, yeah, 2220 is 2220, which is 2220, which is 2220.

Wait, so, 2220 is 2220, which is 2220, which is 2220.

Wait, so, yeah, 2220 is 2220, which is 2220.

Wait, so, that's 2220, which is 2220, which is 2220.

Wait, but, hold on, let me think again.

Wait, 2220 is 2220, which is 2220, which is 2220.

Wait, but 2220 is 2220, which is 2220.

Wait, so, yeah, 2220 is 2220, which is 2220.

Wait, so, 2220 is 2220, which is 2220.

Wait, so, that's 2220, which is 2220.

Wait, so, yeah, 2220 is 2220.

...

---

**Prompt: Boundary Detection**

[SYSTEM PROMPT]
You are a reasoning−trajectory segmenter.
Decide whether two adjacent paragraphs belong to the SAME subgoal.
Return JSON only.

[USER PROMPT]
Task: Determine if two adjacent paragraphs belong to the same subgoal.

Rules:
− YES if both paragraphs pursue the same immediate objective.
− NO if the next paragraph introduces a new objective/stage (e.g., starts a new computation, introduces a new intermediate target, or shifts to verification/finalization).
− Self−correction without changing the objective => YES.

Problem:
{problem}

Context before the current paragraph:
{prev_context}

Current paragraph:
{cur_paragraph}

Next paragraph:
{next_paragraph}

Output JSON schema:
{
   "decision": "YES" or "NO",
   "confidence": 0.0 to 1.0,
   "reason": "..."
}

---

**Prompt: High-level Abstraction Labelling**

[SYSTEM PROMPT]
You are a high−level abstraction labeler.
Given a merged reasoning chunk, output exactly one reasoning mode, one subgoal, and one summary.
Use neutral factual style and avoid personal pronouns.

Return JSON only.

[USER PROMPT]
Task: Label this merged chunk as ONE subgoal.

Reasoning mode candidates:
− ProblemUnderstanding: restate the goal, identify givens/unknowns, clarify constraints.
− Planning: outline substeps, decompose, decide the next objective (without executing).
− Recall: state a relevant definition/theorem/formula/fact needed for the solution.
− Derivation: derive a non−trivial intermediate claim/lemma (not a direct computation).
− Calculation: perform algebra/arithmetic/symbolic manipulation or explicit computation.
− CaseAnalysis: explore alternatives or split into cases without abandoning the current direction.
− Verification: check correctness/consistency (plug−in, sanity check, dimension/bounds, cross−check).
− ErrorCorrection: fix a detected mistake while keeping the overall approach.
− Backtracking: explicitly abandon/revise a prior approach/assumption and switch direction.
− Synthesis: combine intermediate results to reach the final conclusion.
− Finalization: present the final answer or finalize formatting.
− Other

Constraints:
− English.
− Subgoal: action + object style, <= 30 words.
− Avoid repeating prior subgoal names (paraphrase if needed).
− If the chunk mainly corrects a prior step, set subgoal to: Refine <prior subgoal> (paraphrase if needed).
− Summary: neutral and factual; avoid personal pronouns (I/We/You/They/The user); preserve key intermediate results and decisions.

Problem:
{problem}

Previous steps:
{history_text}

Merged chunk:
{merged_text}

Output JSON schema:
{
   "reasoning_mode": "...",
   "subgoal": "...",
   "summary": "..."
}

---

## Prompt: MLR (high-level)

[SYSTEM PROMPT]
You are a high−level planner. Given the problem and completed history, generate one reasoning mode and one next subgoal.
Return only the two required lines (or DONE). Do not provide low−level solution details.

[USER PROMPT]
Instruction:
Generate the next reasoning mode and next subgoal.

Rules:
− Output exactly two lines in this format:
   reasoning_mode: <one label>
   next_subgoal: <text>
− If the problem is already solved, output exactly: DONE
− reasoning_mode must be one of: {ProblemUnderstanding, Planning, Recall, Derivation, Calculation, CaseAnalysis, Verification, ErrorCorrection, Backtracking, Synthesis, Finalization, Other}

– next_subgoal must be action + object style, <= 30 words
– Do not repeat previous subgoals; if unavoidable, paraphrase
– Keep it minimal and actionable

Problem:
{problem}

Completed history:
{history_text}

Continue:

---

**Prompt: MLR (low-level)**

[SYSTEM PROMPT]
You are a low−level executor.
Focus strictly on the given reasoning mode and the current subgoal.
Do not start future subgoals.
Output only the reasoning content (no headings, no labels).
End your output with a single line: <END_STEP>

[USER PROMPT]
Continue solving the problem. Only complete the current subgoal.

Problem:
{problem}

Existing reasoning:
{reasoning_prefix}

Reasoning mode for this step:
{reasoning_mode}

Current subgoal:
{subgoal}

Continue:

---

**Prompt: MLR (summarization)**

[SYSTEM PROMPT]
You are a step summarizer for hierarchical planning.
Return only the summary text (no headings, no bullets, no extra text).

[USER PROMPT]
Summarize the current step into neutral and factual text for high−level planning.

Keep: key intermediate results/values, assumptions/constraints, method choice, and any corrections.
Drop: long derivations, repetitive explanations, and formatting.
Do not propose next steps or new reasoning. No personal pronouns.

Problem:
{problem}

Completed history:
{history_text}

Current step:
Mode: {reasoning_mode}
Subgoal: {subgoal}
Content:
{step_text}

Summary:

---

**Example: MLR Output (high-level) - MATH500**

**Problem:**
The set of points $(x, y, z)$ that satisfy $2x = 3y = -z$ is a line.
The set of points $(x, y, z)$ that satisfy $6x = -y = -4z$ is another line.
Find the angle between these lines, in degrees.

**Solution:**
Mode: ProblemUnderstanding
Subgoal: Identify principle for angle between 3D lines
Summary:
The angle between two lines in 3D space is determined by the angle between their direction vectors.

--------------------------------------------------------------------------------

Mode: Calculation
Subgoal: Compute direction vector of line 1
Summary:
Direction vector of the first line is (3, 2, −6).

--------------------------------------------------------------------------------

Mode: Calculation
Subgoal: Compute direction vector of line 2
Summary:
Direction vector is (2, −12, −3).

--------------------------------------------------------------------------------

Mode: Calculation
Subgoal: Compute angle via dot product
Summary:
Direction vectors: (3, 2, −6) and (2, −12, −3).
Dot product: 3∗2 + 2∗(−12) + (−6)∗(−3) = 6 −24 + 18 = 0.
Since the dot product is zero, the vectors are perpendicular.
Therefore, the angle between the lines is 90 degrees.

--------------------------------------------------------------------------------

Mode: Verification
Subgoal: Verify perpendicularity and angle
Summary:
− Both lines pass through the origin and intersect there.
− Their direction vectors are perpendicular (dot product is zero).
− Therefore, the angle between the two lines is 90 degrees.

--------------------------------------------------------------------------------

Mode: Finalization
Subgoal: State final angle
Summary:

Thus, the final answer is $\boxed{90^\circ}$.

---

**Example: MLR Output (low-level) - MATH500**

**Problem:**
The set of points $(x, y, z)$ that satisfy $2x = 3y = -z$ is a line.
The set of points $(x, y, z)$ that satisfy $6x = -y = -4z$ is another line.
Find the angle between these lines, in degrees.

**Solution:**
Mode: ProblemUnderstanding
Subgoal: Identify principle for angle between 3D lines
Content:
Okay, so I need to find the angle between two lines in 3D space. The lines are given by the equations 2x = 3y = −z and 6x = −y = −4z. Hmm, let me recall how to approach this. I think the angle between two lines is determined by the angle between their direction vectors. So first, I need to find the direction vectors of each line.

--------------------------------------------------------------------------------

Mode: Calculation
Subgoal: Compute direction vector of line 1
Content:

Starting with the first line: 2x = 3y = −z. Let me parse this equation. It's saying that 2x, 3y, and −z are all equal to each other. Let me denote this common value as some parameter, say t. So, 2x = t, 3y = t, −z = t. Then, solving for x, y, z in terms of t:

x = t/2, y = t/3, z = −t.

So the parametric equations for the first line are:

x = (1/2)t,

y = (1/3)t,

z = −t.

Therefore, the direction vector of the first line is given by the coefficients of t, which is (1/2, 1/3, −1). But to make it easier, maybe I can multiply by 6 to eliminate the denominators. Multiplying each component by 6 gives (3, 2, −6). So the direction vector is (3, 2, −6). Let me check that: starting from 2x = 3y = −z, if I set t = 6x, then 3y = 6x => y = 2x, and −z = 6x => z = −6x. So the direction vector is (x, y, z) = (x, 2x, −6x), which is in the direction of (1, 2, −6). Wait, but earlier I got (3, 2, −6). Hmm, maybe I made a mistake there. Let me clarify.

If 2x = 3y = −z, then let me set 2x = k. Then 3y = k => y = k/3, and −z = k => z = −k. So x = k/2, y = k/3, z = −k. So the direction vector is (k/2, k/3, −k). To make this a direction vector, we can take k = 6 (the least common multiple of denominators 2 and 3). Then x = 3, y = 2, z = −6. So the direction vector is (3, 2, −6). So that's correct. Okay, so direction vector **v** is (3, 2, −6).

────────────────────────────────────────────────────────────────────

Mode: Calculation
Subgoal: Compute direction vector of line 2
Content:
Now the second line: 6x = −y = −4z. Similarly, let's set 6x = m. Then −y = m => y = −m, and −4z = m => z = −m/4. So x = m/6, y = −m, z = −m/4. Therefore, the direction vector is (m/6, −m, −m/4). Again, to eliminate denominators, multiply by 12 (the least common multiple of 6 and 4). Then x = 2, y = −12, z = −3. So the direction vector is (2, −12, −3). Alternatively, simplifying further, we can factor out a common factor. Let's see, 2, −12, −3. The greatest common divisor is 1, so maybe that's the simplest form. Alternatively, we can divide by 1, so the direction vector is (2, −12, −3). Alternatively, maybe we can write it as (2, −12, −3) or simplify by dividing by a common factor. Let's check: 2 and −12 have a common factor of 2, but −3 doesn't. So maybe (2, −12, −3) is the simplest. Alternatively, if we factor out a 1, it's still the same. So direction vector **u** is (2, −12, −3).

────────────────────────────────────────────────────────────────────

Mode: Calculation
Subgoal: Compute angle via dot product
Content:
Now, to find the angle between the two lines, we need the angle between their direction vectors **v** and **u**. The formula for the angle $\theta$ between two vectors is:

$\cos \theta = (\mathbf{v} \cdot \mathbf{u})/(||\mathbf{v}||||\mathbf{u}||)$

So first, let's compute the dot product of **v** and **u**. **v** is (3, 2, −6), **u** is (2, −12, −3).

Dot product = (3)(2) + (2)(−12) + (−6)(−3) = 6 −24 + 18 = 6 −24 is −18, −18 +18 is 0. Wait, the dot product is zero? That would mean the vectors are perpendicular, so the angle between them is 90 degrees. But that seems surprising. Let me double−check the calculations.

$\mathbf{v} \cdot \mathbf{u}$ = (3)(2) + (2)(−12) + (−6)(−3) = 6 −24 + 18.

6 −24 is −18, −18 +18 is 0. Yep, that's correct. So the dot product is indeed zero, which implies that the vectors are orthogonal. Therefore, the angle between them is 90 degrees. Hmm, so is that the answer? 90 degrees? That seems straightforward, but let me confirm once again.

Alternatively, maybe I made a mistake in determining the direction vectors. Let me check the direction vectors again.

First line: 2x = 3y = −z. Let me parametrize it differently. Let's set 2x = k. Then 3y = k => y = k/3, and −z = k => z = −k. So x = k/2, y = k/3, z = −k. So direction vector is (k/2, k/3, −k). To eliminate fractions, multiply by 6: (3, 2, −6). Correct.

Second line: $6x = −y = −4z$. Let's set $6x = m$. Then $−y = m => y = −m$, and $−4z = m => z = −m/4$. So $x = m/6$, $y = −m$, $z = −m/4$. Direction vector is $(m/6, −m, −m/4)$. Multiply by 12: $(2, −12, −3)$. Correct.

So direction vectors are $(3, 2, −6)$ and $(2, −12, −3)$. Their dot product is $3*2 + 2*(−12) + (−6)*(−3) = 6 −24 + 18 = 0$. So yes, they are perpendicular. Therefore, the angle between the lines is 90 degrees.

−−−−−−−−−−−−−−−−−−−−−−−−−−−−−−−−−−−−−−−−−−−−−−−−−−−−−−−−−−−−−−−−−−−−−−−−−−−−−−

Mode: Verification
Subgoal: Verify perpendicularity and angle
Content:
But wait, just to be thorough, maybe I should check if the lines are indeed skew lines or intersecting lines. If they intersect, then the angle between them is still defined as the angle between their direction vectors. If they are skew lines, the angle is still the same. So regardless, the angle between them is determined by their direction vectors.

Alternatively, maybe I can check if the lines are coplanar or not, but since they are in 3D space, unless they are parallel or intersecting, they are skew. But since their direction vectors are not scalar multiples, they are not parallel, so they are skew lines. But the angle between them is still the angle between their direction vectors.

Alternatively, maybe the problem is designed such that the angle is 90 degrees, but let me verify once more.

Alternatively, maybe I can check with another method. Let me compute the cross product of the direction vectors to find the sine of the angle, but since the dot product is zero, the angle must be 90 degrees.

Alternatively, if the dot product is zero, the vectors are perpendicular, so the angle is 90 degrees.

Alternatively, maybe the problem is designed to have a right angle, but I need to confirm.

Alternatively, maybe I made a mistake in the direction vectors. Let me check again.

First line: $2x = 3y = −z$. Let me take a point on the line. Let me choose x=0. Then $3y = −z$. Let me set y=0, then z=0. So the point $(0,0,0)$ is on the line. Then, direction vector is $(1/2, 1/3, −1)$ as found earlier. So direction vector is $(1/2, 1/3, −1)$. So if I take t=1, the point is $(1/2, 1/3, −1)$. So direction vector is $(1/2, 1/3, −1)$. Multiplying by 6 gives $(3, 2, −6)$. Correct.

Second line: $6x = −y = −4z$. Let me take x=0. Then $−y = −4z => y = 4z$. Let me set z=0, then y=0. So the point $(0,0,0)$ is on this line as well. Wait, both lines pass through the origin? Then they intersect at the origin. But if they intersect, then they are coplanar, but in 3D space, two lines that intersect are coplanar, but the angle between them is still determined by their direction vectors. However, if they intersect, the angle between them is the angle between their direction vectors, which is 90 degrees. So even if they intersect, the angle between them is 90 degrees. So that's consistent.

But wait, if both lines pass through the origin, and they intersect at the origin, but their direction vectors are perpendicular, then the angle between them is 90 degrees. So that's correct.

Alternatively, maybe I can check with another point. Let's take a point on the first line. For example, take x=1. Then $2*1=3y => y=3/2$, and $−z=3*1 => z=−3$. So the point $(1, 3/2, −3)$. Similarly, take x=1 on the second line. Then $6*1=−y => y=−6$, and $−4z = −6 => z=1.5$. So the point $(1, −6, 1.5)$. Now, check if these two points are connected by a line that is perpendicular to both direction vectors. But maybe that's overcomplicating. Since the direction vectors are perpendicular, the angle is 90 degrees.

Alternatively, maybe I can check the angle using another formula. The angle between two lines is given by the angle between their direction vectors. Since the dot product is zero, the angle is 90 degrees. So I think that's correct.

Therefore, the angle between the two lines is 90 degrees.

−−−−−−−−−−−−−−−−−−−−−−−−−−−−−−−−−−−−−−−−−−−−−−−−−−−−−−−−−−−−−−−−−−−−−−−−−−−−−−

Mode: Finalization
Subgoal: State final angle
Content:

Thus, the final answer is $\boxed{90°}$.

**Example: MLR Output (high-level) - AIME24**

**Problem:**

Find the number of triples of nonnegative integers $(a, b, c)$ satisfying $a + b + c = 300$ and $a^2b + a^2c + b^2a + b^2c + c^2a + c^2b = 6,000,000$.

**Solution:**

Mode: ProblemUnderstanding

Subgoal: Restate constraints and rewrite expression

Summary:

Triples (a, b, c) of nonnegative integers must satisfy:

$- a + b + c = 300$

$- a^2b + a^2c + b^2a + b^2c + c^2a + c^2b = 6,000,000$

The sum $a^2b + a^2c + b^2a + b^2c + c^2a + c^2b$ can be rewritten as $a^2(b + c) + b^2(a + c) + c^2(a + b)$.

----------------------------------------------------------------

Mode: Verification

Subgoal: Verify factoring into grouped terms

Summary:

Factoring terms:

$- a^2b + a^2c = a^2(b + c)$

$- b^2a + b^2c = b^2(a + c)$

$- c^2a + c^2b = c^2(a + b)$

So, the expression simplifies to $a^2(b + c) + b^2(a + c) + c^2(a + b)$.

----------------------------------------------------------------

Mode: Calculation

Subgoal: Substituting the sum constraint

Summary:

Expression after substitution:

$a^2(300 - a) + b^2(300 - b) + c^2(300 - c)$

Main equation:

$a^2(300 - a) + b^2(300 - b) + c^2(300 - c) = 6,000,000$

----------------------------------------------------------------

Mode: Calculation

Subgoal: Expanding and simplifying the equation

Summary:

Expanded expression:

$300(a^2 + b^2 + c^2) - (a^3 + b^3 + c^3) = 6,000,000$

This relates to the sum $a + b + c = 300$.

----------------------------------------------------------------

Mode: Derivation

Subgoal: Expressing in terms of symmetric sums

Summary:

S1 = a + b + c = 300

S2 = ab + bc + ca

S3 = abc

$a^2 + b^2 + c^2 = S1^2 - 2S2$

$a^3 + b^3 + c^3 = S1^3 - 3S1S2 + 3S3$

Substituting into the equation:

$300(S1^2 - 2S2) - (S1^3 - 3S1S2 + 3S3) = 6,000,000$

----------------------------------------------------------------

Mode: Calculation

Subgoal: Plugging in the known value and simplifying

Summary:

First term: $27,000,000 - 600S2$

Second term: $-27,000,000 + 900S2 - 3S3$

Combine: $27,000,000 - 600S2 - 27,000,000 + 900S2 - 3S3 = 6,000,000$

Simplify: $300S2 - 3S3 = 6,000,000$

Divide by 3: $100S2 - S3 = 2,000,000$

––––––––––––––––––––––––––––––––––––––––––––––––––––––––––––––––––––––––––

Mode: Planning
Subgoal: Analyzing the new equation and possible values
Summary:
Equation: $100S_2 - S_3 = 2,000,000$
Given: $S_1 = a + b + c = 300$, $S_2 = ab + bc + ca$, $S_3 = abc$
Need: Integer solutions for nonnegative $a, b, c$ with $a + b + c = 300$ and $100S_2 - S_3 = 2,000,000$.
Approach: Consider cases with symmetry (e.g., two variables equal or all equal) to find possible solutions.

––––––––––––––––––––––––––––––––––––––––––––––––––––––––––––––––––––––––––

Mode: CaseAnalysis
Subgoal: Checking the case where all variables are equal
Summary:
If $a = b = c = 100$, then $a + b + c = 300$, $S_2 = 30,000$, $S_3 = 1,000,000$, and $100S_2 - S_3 = 2,000,000$, which satisfies the condition.
So, $(100, 100, 100)$ is a solution, and since all are equal, there is only one such triple.

––––––––––––––––––––––––––––––––––––––––––––––––––––––––––––––––––––––––––

Mode: CaseAnalysis
Subgoal: Checking the case where two variables are equal
Summary:
Case: a = b ≠ c
c = 300 − 2a

S2 = −3a^2 + 600a
S3 = a^2(300 − 2a)

Equation:
a^3 − 300a^2 + 30,000a − 1,000,000 = 0
Factor: (a − 100)^3 = 0 ⇒ a = 100 is the only solution.

Conclusion: The only solution in this case is a = b = c = 100; no other solutions when two variables are equal.

––––––––––––––––––––––––––––––––––––––––––––––––––––––––––––––––––––––––––

Mode: CaseAnalysis
Subgoal: Considering the case where all variables are distinct
Summary:
All variables distinct:
Given $S_1 = 300$, $100S_2 - S_3 = 2,000,000$, and $S_3 = 100S_2 - 2,000,000$.
Since $S_3 = abc \geq 0$, $S_2 \geq 20,000$.
Maximum $S_2 = 30,000$, minimum $S_2 = 0$.
So, for all variables distinct, $S_2$ must be between 20,000 and 30,000, and $S_3 = 100S_2 - 2,000,000$.
The original equation reduces to $300S_2 - 3S_3 = 6,000,000$, confirming the earlier result.

––––––––––––––––––––––––––––––––––––––––––––––––––––––––––––––––––––––––––

Mode: Derivation
Subgoal: Factoring the equation for further simplification
Summary:
Given 100S2 − S3 = 2,000,000 and S1 = 300,
S3 = 100S2 − 2,000,000, and S3 = abc, so
abc = 100(ab + bc + ca) − 2,000,000.

Rearranged and factored,
(a − 100)(b − 100)(c − 100) = 0.

––––––––––––––––––––––––––––––––––––––––––––––––––––––––––––––––––––––––––

Mode: Derivation
Subgoal: Interpreting the factored equation and finding solutions
Summary:
At least one of a, b, or c must be 100.
The other two variables sum to 200 (since a + b + c = 300).
All nonnegative integer triples (a, b, c) where one is 100 and the other two sum to 200 are solutions.
This condition is equivalent to the original equation, so all such triples satisfy both equations.

––––––––––––––––––––––––––––––––––––––––––––––––––––––––––––––––––––––––––

Mode: Calculation
Subgoal: Counting the number of valid triples
Summary:
Case 1: a = 100 ⇒ 201 triples
Case 2: b = 100 ⇒ 201 triples
Case 3: c = 100 ⇒ 201 triples

Triple (100,100,100) is counted 3 times (once in each case).
Using inclusion–exclusion:
Total triples = $3 \times 201 - 3 + 1 = 601$.

––––––––––––––––––––––––––––––––––––––––––––––––––––––––––––––––––––––––––––––––

Mode: Verification
Subgoal: Verifying the counting with a small example
Summary:
Listed all possible triples: (1,0,2), (1,1,1), (1,2,0), (0,1,2), (2,1,0), (0,2,1), (2,0,1) – total of 7 triples.

Using inclusion–exclusion: $3 \times 3 - 3 \times 1 + 1 = 9 - 3 + 1 = 7$.

Conclusion: The formula works for this small example.

––––––––––––––––––––––––––––––––––––––––––––––––––––––––––––––––––––––––––––––––

Mode: Finalization
Subgoal: Extract the final answer
Summary:

Thus, the final answer is $\boxed{601}$.

## Example: MLR Output (low-level) - AIME24

**Problem:**
Find the number of triples of nonnegative integers $(a, b, c)$ satisfying $a + b + c = 300$ and
$a^2 b + a^2 c + b^2 a + b^2 c + c^2 a + c^2 b = 6{,}000{,}000$.

**Solution:**
Mode: ProblemUnderstanding
Subgoal: Restate constraints and rewrite expression
Content:
Okay, so the user is trying to solve a problem involving finding triples (a, b, c) of nonnegative integers that satisfy two conditions: a + b + c = 300 and a^2b + a^2c + b^2a + b^2c + c^2a + c^2b = 6,000,000. They started by considering factoring the expression and noticed that it can be rewritten as a^2(b + c) + b^2(a + c) + c^2(a + b).

––––––––––––––––––––––––––––––––––––––––––––––––––––––––––––––––––––––––––––––––

Mode: Verification
Subgoal: Verify factoring into grouped terms
Content:
First, I need to verify if their initial approach is correct. Let's check the expression:

Original expression: a^2b + a^2c + b^2a + b^2c + c^2a + c^2b

Factoring terms:

a^2b + a^2c = a^2(b + c)
b^2a + b^2c = b^2(a + c)
c^2a + c^2b = c^2(a + b)
So yes, the expression simplifies to a^2(b + c) + b^2(a + c) + c^2(a + b). That's a good start.

––––––––––––––––––––––––––––––––––––––––––––––––––––––––––––––––––––––––––––––––

Mode: Calculation
Subgoal: Substituting the sum constraint
Content:
Next, since a + b + c = 300, we can substitute (b + c) with (300 − a), similarly for the others. Let's apply that substitution:

Expression becomes: a^2(300 − a) + b^2(300 − b) + c^2(300 − c)

So the equation is: a^2(300 − a) + b^2(300 − b) + c^2(300 − c) = 6,000,000

––––––––––––––––––––––––––––––––––––––––––––––––––––––––––––––––––––––––––––––––

Mode: Calculation
Subgoal: Expanding and simplifying the equation
Content:
Now, the user might need to find a way to relate this to the sum $a + b + c = 300$. Perhaps expanding the terms or looking for symmetry could help. Let's consider expanding each term:

For each variable x (where x is a, b, or c), the term x^2(300 − x) = 300x^2 − x^3. So the entire expression becomes:

300(a^2 + b^2 + c^2) − (a^3 + b^3 + c^3) = 6,000,000

So we have:

300(a^2 + b^2 + c^2) − (a^3 + b^3 + c^3) = 6,000,000

─────────────────────────────────────────────────────────────────────

Mode: Derivation
Subgoal: Expressing in terms of symmetric sums
Content:
Now, we also know that a + b + c = 300. Let's denote S1 = a + b + c = 300, S2 = ab + bc + ca, and S3 = abc. We can express a^2 + b^2 + c^2 in terms of S1 and S2:

a^2 + b^2 + c^2 = (a + b + c)^2 − 2(ab + bc + ca) = S1^2 − 2S2

Similarly, the sum of cubes can be expressed using the identity:

a^3 + b^3 + c^3 = (a + b + c)^3 − 3(a + b + c)(ab + bc + ca) + 3abc = S1^3 − 3S1S2 + 3S3

Substituting these into the equation:

300(S1^2 − 2S2) − (S1^3 − 3S1S2 + 3S3) = 6,000,000

─────────────────────────────────────────────────────────────────────

Mode: Calculation
Subgoal: Plugging in the known value and simplifying
Content:
Plugging in $S1 = 300$:

300(300^2 − 2S2) − (300^3 − 900S2 + 3S3) = 6,000,000

Let's compute each term step by step:

First term: 300∗(300^2 − 2S2) = 300∗(90,000 − 2S2) = 27,000,000 − 600S2

Second term: −(300^3 − 900S2 + 3S3) = −27,000,000 + 900S2 − 3S3

Combining both terms:

27,000,000 − 600S2 −27,000,000 + 900S2 − 3S3 = 6,000,000

Simplify:

(27,000,000 −27,000,000) + (−600S2 + 900S2) −3S3 = 6,000,000

Which simplifies to:

$300S2 − 3S3 = 6,000,000$

Divide both sides by 3:

100S2 − S3 = 2,000,000

─────────────────────────────────────────────────────────────────────

Mode: Planning
Subgoal: Analyzing the new equation and possible values
Content:
So now we have an equation involving S2 and S3: 100S2 − S3 = 2,000,000

But we also know that S1 = 300, S2 = ab + bc + ca, and S3 = abc. So we need another relation between S2 and S3. However, with three variables and two equations (S1, S2, S3), we might need to find possible integer solutions.

Alternatively, perhaps we can consider that a, b, c are nonnegative integers summing to 300, and we need to find the number of triples such that 100S2 − S3 = 2,000,000.

This seems complicated. Maybe there's a symmetry or specific values that satisfy this equation. Let's consider possible cases where two variables are equal, or all three are equal.

─────────────────────────────────────────────────────────────────────────────
Mode: CaseAnalysis
Subgoal: Checking the case where all variables are equal
Content:
First, check if all three are equal: $a = b = c$. Then $a + b + c = 3a = 300 \Rightarrow a = 100$. So $(100, 100, 100)$. Let's check if this satisfies the second equation.

Compute $S2 = ab + bc + ca = 3*100*100 = 30,000$

$S3 = 100^3 = 1,000,000$

Then $100 S2 - S3 = 100*30,000 - 1,000,000 = 3,000,000 - 1,000,000 = 2,000,000$. Hey, that works! So $(100, 100, 100)$ is a solution. But the problem asks for the number of triples, so this is one solution.

But there might be others. Let's check if permutations of this count, but since all are equal, there's only one such triple.
─────────────────────────────────────────────────────────────────────────────
Mode: CaseAnalysis
Subgoal: Checking the case where two variables are equal
Content:
Now, consider cases where two variables are equal. Let's say $a = b \neq c$. Then $a + a + c = 300 \Rightarrow 2a + c = 300 \Rightarrow c = 300 - 2a$.

Now, compute S2 and S3:

$S2 = a^2 + 2ac = a^2 + 2a(300 - 2a) = a^2 + 600a - 4a^2 = -3a^2 + 600a$

$S3 = a^2 c = a^2(300 - 2a)$

Now plug into $100 S2 - S3 = 2,000,000$:

$100*(-3a^2 + 600a) - a^2(300 - 2a) = 2,000,000$

Compute each term:

$100*(-3a^2 + 600a) = -300a^2 + 60,000a$

Subtract $a^2(300 - 2a) = -300a^2 + 2a^3$

So total equation:

$-300a^2 + 60,000a - 300a^2 + 2a^3 = 2,000,000$

Combine like terms:

$2a^3 - 600a^2 + 60,000a - 2,000,000 = 0$

Divide both sides by 2:

$a^3 - 300a^2 + 30,000a - 1,000,000 = 0$

Now, we need to solve this cubic equation for integer a, where a must be a nonnegative integer such that $c = 300 - 2a \geq 0 \Rightarrow a \leq 150$.

Let's try possible integer roots. Possible rational roots are factors of 1,000,000 divided by factors of 1 (leading coefficient). So possible roots are divisors of 1,000,000. Let's test $a = 100$:

$100^3 - 300*100^2 + 30,000*100 - 1,000,000 = 1,000,000 - 3,000,000 + 3,000,000 - 1,000,000 = 0$. So $a = 100$ is a root. That corresponds to the case where $a = b = 100$, $c = 100$, which we already considered.

Factor out $(a - 100)$:

Using polynomial division or synthetic division:

Divide $a^3 - 300a^2 + 30{,}000a - 1{,}000{,}000$ by $(a - 100)$.

Using synthetic division:

$100 \mid 1\ -300\ 30{,}000\ -1{,}000{,}000$

$\qquad 100\ -20{,}000\ 1{,}000{,}000$

$\underline{\qquad\qquad\qquad\qquad\qquad\qquad\qquad}$

$\qquad 1\ -200\ 10{,}000\ 0$

So the cubic factors as $(a - 100)(a^2 - 200a + 10{,}000)$

Now solve $a^2 - 200a + 10{,}000 = 0$

Discriminant $D = 40{,}000 - 40{,}000 = 0$. So $a = (200)/2 = 100$. Thus, the cubic factors as $(a - 100)^3$. So the only solution is $a = 100$. Therefore, in the case where two variables are equal, the only solution is when all three are equal. So no other solutions in this case.

────────────────────────────────────────────────────────────────────────────

Mode: CaseAnalysis
Subgoal: Considering the case where all variables are distinct
Content:
Next, consider the case where all three variables are distinct. Let's see if there are any solutions where a, b, c are different.

This might be more complex. Let's think if there's a way to parameterize the variables or use Vieta's formulas. Since we have $S1 = 300$, $S2$, and $S3$ related by $100S2 - S3 = 2{,}000{,}000$, perhaps we can express $S3$ in terms of $S2$: $S3 = 100S2 - 2{,}000{,}000$.

But we also know that for nonnegative integers a, b, c, $S3 = abc$ must be nonnegative. So $100S2 - 2{,}000{,}000 \geq 0 \Rightarrow S2 \geq 20{,}000$.

But $S2 = ab + bc + ca$. Given that $a + b + c = 300$, what's the maximum and minimum possible $S2$?

The maximum $S2$ occurs when two variables are as large as possible. For example, if one variable is 0, then $S2 = ab$ where $a + b = 300$, which is maximized when $a = b = 150$, giving $S2 = 150*150 = 22{,}500$. If all variables are equal, $S2 = 3*100*100 = 30{,}000$. Wait, but if $a = b = c = 100$, then $S2 = 3*100*100 = 30{,}000$. Wait, but earlier when we considered two variables equal, $S2$ was $-3a^2 + 600a$, which for a=100 gives $S2 = 30{,}000$. So $S2$ can range depending on the distribution.

Wait, actually, $S2$ can be calculated as $(a + b + c)^2 - (a^2 + b^2 + c^2)/2$, but maybe it's better to think in terms of maximizing/minimizing $S2$ given $a + b + c = 300$.

The maximum of $S2$ occurs when two variables are as large as possible, which would be when one variable is 0, and the other two sum to 300. Then $S2 = (150)(150) = 22{,}500$. Wait, but earlier when all three are equal, $S2 = 3*100*100 = 30{,}000$, which is larger. So actually, the maximum $S2$ occurs when all variables are equal. Because $S2 = ab + bc + ca$ is maximized when the variables are equal, by the inequality that $ab + bc + ca \leq (a + b + c)^2 / 3$, which for $a + b + c = 300$ gives $S2 \leq 300^2 / 3 = 30{,}000$, achieved when $a = b = c = 100$.

Similarly, the minimum $S2$ occurs when one variable is 300 and the others are 0, giving $S2 = 0$. So $S2$ ranges from 0 to 30,000.

But in our case, $S2$ must be at least 20,000 because $S3 = 100S2 - 2{,}000{,}000 \geq 0 \Rightarrow S2 \geq 20{,}000$. So $S2$ is between 20,000 and 30,000.

But how does this help us? We need to find triples (a, b, c) such that $S2$ is between 20,000 and 30,000, and $S3 = 100S2 - 2{,}000{,}000$. Additionally, since a, b, c are nonnegative integers summing to 300, we can model this as solving for a, b, c with these constraints.

Alternatively, perhaps there's a substitution or another identity that can be used. Let's revisit the original equation:

$a^2b + a^2c + b^2a + b^2c + c^2a + c^2b = 6{,}000{,}000$

Notice that this can be written as ab(a + b) + bc(b + c) + ca(c + a). Wait, but that's similar to the earlier expression. Alternatively, factor differently:

= ab(a + b) + bc(b + c) + ca(c + a)

But since a + b + c = 300, we can write a + b = 300 − c, etc. So:

= ab(300 − c) + bc(300 − a) + ca(300 − b)

Which expands to:

300ab − abc + 300bc − abc + 300ca − abc

= 300(ab + bc + ca) − 3abc

So the equation becomes:

$300 S_2 - 3 S_3 = 6{,}000{,}000$

Which simplifies to $100 S_2 - S_3 = 2{,}000{,}000$, which matches what we had earlier.

−−−−−−−−−−−−−−−−−−−−−−−−−−−−−−−−−−−−−−−−−−−−−−−−−−−−−−−−−−−−−−−−−−−−−−−−−−−−−−−−

Mode: Derivation
Subgoal: Factoring the equation for further simplification
Content:
So we have $100 S_2 - S_3 = 2{,}000{,}000$, and $S_1 = 300$. So $S_3 = 100 S_2 - 2{,}000{,}000$.

But $S_3 = abc$, so $abc = 100(ab + bc + ca) - 2{,}000{,}000$.

This seems like a complicated equation to solve for integers a, b, $c \geq 0$ with a + b + c = 300.

But perhaps there's a way to parameterize this. Let's consider that $abc = 100(ab + bc + ca) - 2{,}000{,}000$.

Rearranging:

abc − 100ab − 100bc − 100ca = −2,000,000

Add 10000(a + b + c) to both sides:

abc − 100ab − 100bc − 100ca + 10000(a + b + c) = −2,000,000 + 10000∗300

Compute the right−hand side:

−2,000,000 + 3,000,000 = 1,000,000

Now, the left−hand side can be factored:

abc − 100ab − 100bc − 100ca + 10000a + 10000b + 10000c

Factor terms:

= abc + (−100ab −100bc −100ca) + 10000(a + b + c)

Notice that this resembles (a − 100)(b − 100)(c − 100) expanded. Let's check:

Expand (a − 100)(b − 100)(c − 100):

= abc − 100ab − 100ac − 100bc + 10,000a + 10,000b + 10,000c − 1,000,000

Compare to our left−hand side:

Left−hand side after adding 10000(a + b + c):

abc −100ab −100bc −100ca + 10,000a + 10,000b + 10,000c

So the difference between this and the expanded (a −100)(b −100)(c −100) is the constant term −1,000,000. Therefore:

(a − 100)(b − 100)(c − 100) = abc −100ab −100ac −100bc +10,000a +10,000b +10,000c −1,000,000

But our left−hand side is:

abc −100ab −100bc −100ca +10,000a +10,000b +10,000c = (a −100)(b −100)(c −100) + 1,000,000

Therefore, the equation becomes:

(a −100)(b −100)(c −100) + 1,000,000 = 1,000,000

So:

(a −100)(b −100)(c −100) = 0
────────────────────────────────────────────────────────────────────────────
Mode: Derivation
Subgoal: Interpreting the factored equation and finding solutions
Content:
Ah! This simplifies things. Therefore (a − 100)(b − 100)(c − 100) must be zero. Which means at least one of the factors must be zero. Therefore, at least one of a, b, c must be equal to 100.

So the solutions are all triples (a, b, c) where at least one of a, b, or c is 100, and the other two variables sum to 200 (since a + b + c = 300).

Therefore, the problem reduces to finding all nonnegative integer triples (a, b, c) such that one of them is 100, and the other two sum to 200. However, we also need to ensure that the original equation holds, but since we derived that this condition is equivalent to the original equation, all such triples will satisfy both equations.
────────────────────────────────────────────────────────────────────────────
Mode: Calculation
Subgoal: Counting the number of valid triples
Content:
Now, let's count the number of such triples.

Case 1: a = 100. Then b + c = 200. The number of nonnegative integer solutions for (b, c) is 201 (from b=0 to b=200). However, since triples are ordered, each solution here corresponds to a unique triple (100, b, c).

Case 2: b = 100. Similarly, a + c = 200. Number of solutions: 201.

Case 3: c = 100. Similarly, a + b = 200. Number of solutions: 201.

However, we have to subtract the overlaps where two variables are 100. For example, if a = 100 and b = 100, then c = 100, which is counted in all three cases. There are three such overlapping triples: (100,100,100), (100,100,100), etc., but since it's the same triple, it's only counted once in each case. Wait, actually, when we count the three cases, the triple (100,100,100) is included in all three cases. So total overlaps are:

Each triple where two variables are 100 is counted in two cases. For example, (100,100,100) is counted in all three cases. Wait, no, if a = 100, b = 100, then c = 100, so it's counted once in each case. Wait, no, when a = 100, b can be 100, which gives c = 100. Similarly for the other cases. So the triple (100,100,100) is counted three times in total (once in each case). Similarly, any triple where exactly two variables are 100 would be counted twice. Wait, but if two variables are 100, say a = 100 and b = 100, then c = 100, which is the same as all three being 100. So there are no triples where exactly two variables are 100 because that forces the third to also be 100. Therefore, the only overlap is the triple (100,100,100), which is counted three times.

Therefore, using inclusion−exclusion:

Total triples = 3∗201 − 3∗1 + 1∗1 = 603 − 3 + 1 = 601.
────────────────────────────────────────────────────────────────────────────
Mode: Verification
Subgoal: Verifying the counting with a small example
Content:
But let's check with a small example. Suppose we have variables summing to 3 instead of 300, and we want to count triples where one variable is 1. Then:

Case 1: a=1, b + c =2 $\Rightarrow$ (1,0,2), (1,1,1), (1,2,0)

Case 2: b=1, a + c =2 $\Rightarrow$ (0,1,2), (1,1,1), (2,1,0)

Case 3: c=1, a + b =2 $\Rightarrow$ (0,2,1), (1,1,1), (2,0,1)

Total listed triples: (1,0,2), (1,1,1), (1,2,0), (0,1,2), (2,1,0), (0,2,1), (2,0,1). That's 7 triples. Using inclusion–exclusion:

$3*3 - 3*1 + 1 = 9 - 3 + 1 = 7$. Correct. So the formula works.
––––––––––––––––––––––––––––––––––––––––––––––––––––––––––––––––––––––––––––––––––––––
Mode: Finalization
Subgoal: Extract the final answer
Content:

Thus, the final answer is $\boxed{601}$ .

---

### Example: MLR Output (high-level) - GPQA (Diamond)

**Problem:**
Q: 3–(hydroxymethyl)–5–(prop–1–en–2–yl)cyclohexan–1–one is treated with sodium hydride, followed by benzyl bromide, forming product 1. 1 is treated with p–toluenesulfonyl hydrazide in the presence of a catalytic amount of HCl, forming product 2. 2 is treated at low temperature with n–butyllithium, followed by aqueous ammonium chloride, forming product 3. 3 is stirred with Pd/C under a hydrogen atmosphere, forming product 4. What is the structure of product 4?

Choices:
A. (((3–isopropylcyclohexyl)methoxy)methyl)benzene
B. (3–isopropylcyclohexyl)methanol
C. N'–(3–(hydroxymethyl)–5–isopropylcyclohexyl)–4–methylbenzenesulfonohydrazide
D. 3–((benzyloxy)methyl)–1–butyl–5–isopropylcyclohexan–1–ol

**Solution:**
Mode: ProblemUnderstanding
Subgoal: Understanding the problem and outlining the approach
Summary:
Need to determine the structure of product 4 in the reaction sequence.
The starting material is 3–(hydroxymethyl)–5–(prop–1–en–2–yl)cyclohexan–1–one.
– The cyclohexanone ring has a hydroxymethyl group at position 3.
– At position 5, there is a prop–1–en–2–yl group, which is CH2=C(CH3)– (a methyl group attached to a double bond).
– The structure is a cyclohexanone ring with these two substituents at positions 3 and 5.
––––––––––––––––––––––––––––––––––––––––––––––––––––––––––––––––––––––––––––––––––––––
Mode: Derivation
Subgoal: Analyzing the first reaction: NaH and benzyl bromide
Summary:
NaH deprotonates the hydroxymethyl group to form an alkoxide, which then reacts with benzyl bromide to form a benzyl ether. Product 1 is 3–(benzyloxymethyl)–5–(prop–1–en–2–yl)cyclohexan–1–one, where the hydroxymethyl group is converted to a benzyloxymethyl group.
––––––––––––––––––––––––––––––––––––––––––––––––––––––––––––––––––––––––––––––––––––––
Mode: Derivation
Subgoal: Analyzing the second reaction: p–toluenesulfonyl hydrazide and HCl
Summary:
Product 1 reacts with p–toluenesulfonyl hydrazide (TsNHNH2) and catalytic HCl to form a tosylhydrazone. The ketone group (cyclohexanone) reacts with the hydrazide, losing water and forming a hydrazone. Product 2 is the tosylhydrazone of the ketone, with all other substituents (including prop–1–en–2–yl) unchanged. The structure is N'–(3–(benzyloxymethyl)–5–(prop–1–en–2–yl)cyclohexylidene)–4–methylbenzenesulfonohydrazide.
––––––––––––––––––––––––––––––––––––––––––––––––––––––––––––––––––––––––––––––––––––––
Mode: Derivation
Subgoal: Analyzing the third reaction: n–butyllithium and aqueous ammonium
Summary:
Product 2 is treated with n–butyllithium and aqueous ammonium chloride (Shapiro reaction), converting the ketone–derived hydrazone into an alkene. The cyclohexanone ring becomes a cyclohexene ring with a double

bond between C1 and C2. Product 3 is a cyclohexene with a benzyloxymethyl group at C3 and an isopropenyl group at C5.
———————————————————————————————————————————————

Mode: Derivation
Subgoal: Analyzing the fourth reaction: hydrogenation with Pd/C and H2
Summary:
– Hydrogenation with Pd/C and $H_2$ saturates all double bonds.
– The cyclohexene ring becomes a cyclohexane ring.
– The isopropenyl group ($CH_2$=C($CH_3$)–) is reduced to an isopropyl group ($CH_2$CH($CH_3$)–).
– Product 4: cyclohexane ring with a benzyloxymethyl group at C3 and an isopropyl group at C5.
– The ring is fully saturated; no other changes occur.
———————————————————————————————————————————————

Mode: Synthesis
Subgoal: Recapping the sequence and mapping substituent positions
Summary:
Starting material: cyclohexan–1–one
Position 3: hydroxymethyl → benzyloxymethyl (after NaH/benzyl bromide)
Position 1: ketone → tosylhydrazone → double bond (after Shapiro reaction)
Ring: cyclohexene (double bond between 1 and 2)
Position 5: prop–1–en–2–yl (CH2=C(CH3)–)
Final product: 3–(benzyloxymethyl)–5–isopropylcyclohexane
———————————————————————————————————————————————

Mode: Verification
Subgoal: Evaluating the effect of hydrogenolysis on the benzyl ether
Summary:
The benzyloxymethyl group should remain after hydrogenation with Pd/C, as benzyl ethers are generally stable under these conditions. Therefore, product 4 should have a benzyloxymethyl group at position 3 and isopropyl at position 5 on cyclohexane. None of the options match this structure.
———————————————————————————————————————————————

Mode: Verification
Subgoal: Re–examining the possibility of benzyl ether cleavage under hydrogenation
Summary:
– Ketone is at position 1 on the cyclohexane ring.
– Substituents at positions 3 (benzyloxymethyl) and 5 (isopropyl) are two positions apart.
– After Shapiro elimination, double bond forms between positions 1 and 2.
– Hydrogenation restores the single bond; substituents remain at positions 3 and 5.
– In product 4, cyclohexane ring has benzyloxymethyl at position 3 and isopropyl at position 5.
– This arrangement does not match any of the answer choices.
———————————————————————————————————————————————

Mode: Verification
Subgoal: Considering the effect of hydrogenolysis on the benzyl ether group
Summary:
Step 1: alkylation to benzyloxymethyl–correct.
Step 2: tosylhydrazone formation at ketone–correct.
Step 3: Shapiro reaction forms alkene between positions 1 and 2–correct.
Step 4: hydrogenation saturates both the 1–2 alkene and isopropenyl group to isopropyl.
Final product: 3–(benzyloxymethyl)–5–isopropylcyclohexane.
None of the options exactly match this structure.
———————————————————————————————————————————————

Mode: Backtracking
Subgoal: Realizing benzyl ether cleavage under hydrogenation conditions
Summary:
Starting material: 3–(hydroxymethyl)–5–(prop–1–en–2–yl)cyclohexan–1–one.
– Tosylhydrazone forms at position 1 (the ketone).
– Elimination forms an alkene between positions 1 and 2.
– Substituents at 3 (hydroxymethyl) and 5 (prop–1–en–2–yl) remain.
– Hydrogenation reduces the double bond (1,2) and the prop–1–en–2–yl group at 5.
– Final product: cyclohexane ring with substituents at 3 (hydroxymethyl) and 5 (isopropyl).
– Question: how to name this compound.
———————————————————————————————————————————————
Mode: Recall
Subgoal: Applying IUPAC nomenclature and lowest locant rule
Summary:

After renumbering to give the substituents the lowest possible numbers, the hydroxymethyl group is at position 1 and the isopropyl at position 3. The correct IUPAC name is 1−(hydroxymethyl)−3−isopropylcyclohexane, also named as (3−isopropylcyclohexyl)methanol. Therefore, option B is correct.

────────────────────────────────────────────────────────────────────────────

Mode: Finalization

Subgoal: Final conclusion and answer selection

Summary:

Numbering changes to give the lowest possible numbers, repositioning the substituents as 1 and 3 instead of 3 and 5.

Correct answer is B .

---

## Example: MLR Output (low-level) - GPQA (Diamond)

**Problem:**

Q: 3−(hydroxymethyl)−5−(prop−1−en−2−yl)cyclohexan−1−one is treated with sodium hydride, followed by benzyl bromide, forming product 1. 1 is treated with p−toluenesulfonyl hydrazide in the presence of a catalytic amount of HCl, forming product 2. 2 is treated at low temperature with n−butyllithium, followed by aqueous ammonium chloride, forming product 3. 3 is stirred with Pd/C under a hydrogen atmosphere, forming product 4. What is the structure of product 4?

Choices:
A. (((3−isopropylcyclohexyl)methoxy)methyl)benzene
B. (3−isopropylcyclohexyl)methanol
C. N'−(3−(hydroxymethyl)−5−isopropylcyclohexyl)−4−methylbenzenesulfonohydrazide
D. 3−((benzyloxy)methyl)−1−butyl−5−isopropylcyclohexan−1−ol

**Solution:**

Mode: ProblemUnderstanding

Subgoal: Understanding the problem and outlining the approach

Content:

Okay, so I need to figure out the structure of product 4 in this reaction sequence. Let me start by breaking down each step carefully.

First, the starting material is 3−(hydroxymethyl)−5−(prop−1−en−2−yl)cyclohexan−1−one. Let's draw that out. The cyclohexanone ring has a hydroxymethyl group at position 3 and a prop−1−en−2−yl group (which is an allyl group attached at the middle carbon, so it's actually a methyl group attached to a double bond, right? So like CH2−C(CH2)− maybe?

Wait, prop−1−en−2−yl would be CH2=C(CH3)−, right? So the substituent is a methyl group attached to a double bond. So the cyclohexanone ring has that at position 5 and the hydroxymethyl at position 3.

────────────────────────────────────────────────────────────────────────────

Mode: Derivation

Subgoal: Analyzing the first reaction: NaH and benzyl bromide

Content:

Then, the first step is treatment with sodium hydride, followed by benzyl bromide. Sodium hydride is a strong base, so maybe it deprotonates something. The hydroxymethyl group has an −OH. Deprotonating that would give an alkoxide, which is a good nucleophile. Then benzyl bromide would act as an alkylating agent. So the benzyl group (Bn) would be attached to the oxygen, making a benzyl ether. So product 1 would be 3−(benzyloxymethyl)−5−(prop−1−en−2−yl)cyclohexan−1−one. That makes sense because the hydroxymethyl becomes benzyloxymethyl after alkylation.

────────────────────────────────────────────────────────────────────────────

Mode: Derivation

Subgoal: Analyzing the second reaction: p−toluenesulfonyl hydrazide and HCl

Content:

Next, product 1 is treated with p−toluenesulfonyl hydrazide (TsNHNH2) with catalytic HCl. That's typically used to form a tosylhydrazone. The ketone group (cyclohexanone) would react with the hydrazide. The mechanism is that the ketone attacks the hydrazide, losing water and forming a hydrazone. So product 2 would be the tosylhydrazone of the ketone. So the cyclohexanone becomes a cyclohexylidene group linked to the tosylhydrazide. So structure would be N'−(3−(benzyloxymethyl)−5−(prop−1−en−2−yl)cyclohexylidene)−4−methylbenzenesulfonohydrazide.

Wait, but maybe the prop−1−en−2−yl is still present? The reaction is only on the ketone, so the other substituents remain.

________________________________________________________________

Mode: Derivation
Subgoal: Analyzing the third reaction: n−butyllithium and aqueous ammonium
Content:
Third step: product 2 is treated with n−butyllithium at low temperature, then aqueous ammonium chloride. This sounds like the Shapiro reaction. The Shapiro reaction uses tosylhydrazones treated with a strong base like n−BuLi to form alkenes via elimination. The mechanism involves deprotonation and elimination of the tosylhydrazide group, forming a double bond. So the cyclohexanone ring, which had the hydrazone, would become a cyclohexene ring?

Wait, but the starting material already has an allyl group (prop−1−en−2−yl).

Wait, maybe the Shapiro reaction here is converting the ketone−derived hydrazone into an alkene. But the original ketone is at position 1 (cyclohexan−1−one). So after Shapiro, that would form a double bond between C1 and C2? So the cyclohexanone becomes cyclohexene with the substituents. So product 3 would be 3−(benzyloxymethyl)−5−(prop−1−en−2−yl)cyclohexene, but the position of the double bond would be between C1 and C2?

Wait, maybe I need to think about regiochemistry here. The Shapiro reaction typically gives less substituted alkenes because it's a concerted elimination. But in the cyclohexane ring, the original ketone is at position 1. So when eliminating, the hydrazone is at C1, and the adjacent hydrogen is at C2. So elimination would form a double bond between C1 and C2. But the ring already has substituents at C3 (benzyloxymethyl) and C5 (prop−1−en−2−yl). So after elimination, the ring becomes a cyclohexene with double bond between C1 and C2. So product 3 is 1,2−cyclohexenyl with substituents at C3 and C5. But the prop−1−en−2−yl is another substituent.

Wait, but maybe the prop−1−en−2−yl is an isopropenyl group (CH2=C(CH3)−). So product 3 would have a cyclohexene ring (double bond between 1 and 2), a benzyloxymethyl group at 3, and an isopropenyl group at 5. Then, when treated with n−BuLi, but after that, aqueous ammonium chloride. Wait, the Shapiro reaction usually forms alkenes via deprotonation and elimination. But maybe in this case, after forming the hydrazone, treatment with n−BuLi leads to deprotonation and elimination of the tosylhydrazide, giving the alkene. So product 3 would be the cyclohexene derivative with those substituents.
________________________________________________________________

Mode: Derivation
Subgoal: Analyzing the fourth reaction: hydrogenation with Pd/C and H2
Content:
Then, product 3 is stirred with Pd/C under hydrogen. Hydrogenation. So any double bonds would be hydrogenated. The cyclohexene ring's double bond (from the Shapiro reaction) would be hydrogenated to a single bond. Also, the prop−1−en−2−yl group (isopropenyl, CH2=C(CH3)−, so that's a vinyl group with a methyl substituent on the double bond) would be hydrogenated, making it a prop−2−yl group (isopropyl group).

Wait, prop−1−en−2−yl is CH2=C(CH3)−, so hydrogenation would make CH2CH(CH3). So the isopropenyl becomes isopropyl. Also, the cyclohexene from the Shapiro reaction (double bond between 1 and 2) would become cyclohexane again. So product 4 would have a cyclohexane ring with substituents: at C3, benzyloxymethyl; at C5, isopropyl.

But wait, after hydrogenation, the ring is fully saturated. Also, the hydroxymethyl group at 3 is now a benzyloxymethyl. But after hydrogenation, is there any other change?
________________________________________________________________

Mode: Synthesis
Subgoal: Recapping the sequence and mapping substituent positions
Content:
Wait, let's recap: Starting material: cyclohexan−1−one (so position 1 is the ketone). Substituents: at 3, hydroxymethyl; at 5, prop−1−en−2−yl (CH2=C(CH3)−).

Step 1: deprotonate hydroxymethyl's OH with NaH, then benzyl bromide gives benzyloxymethyl at position 3. Step 2: form tosylhydrazone at the ketone (position 1). So now, the hydrazone is at position 1. Step 3: Shapiro reaction (n−BuLi) eliminates the tosylhydrazide, forming a double bond between position 1 and 2. So the ring becomes cyclohexene (positions 1 and 2 double bond). So now, the substituents are at 3 (benzyloxymethyl), 5 (CH2=C(CH3)−), and the ring has a double bond between 1 and 2. Step 4: hydrogenation with Pd/C and H2. The double bonds in the molecule (both the cyclohexene ring's 1,2−double bond and the prop−1−en−2−yl group's double bond) would be hydrogenated. So the cyclohexene becomes cyclohexane (single bond between 1 and 2), and the prop−1−en−2−yl becomes isopropyl (CH(CH3)2). So the substituent at position 5 becomes isopropyl. So product 4 would be 3−(benzyloxymethyl)−5−isopropylcyclohexane.

Wait, but cyclohexane is a six−membered ring. The substituents are at positions 3 and 5. But then, after hydrogenation, the ring is fully saturated. Also, the hydroxymethyl was at 3, which became benzyloxymethyl. But after hydrogenation, is there any other change?

────────────────────────────────────────────────────────────────────────────

Mode: Verification
Subgoal: Evaluating the effect of hydrogenolysis on the benzyl ether
Content:
Wait, but in the final hydrogenation step, maybe the benzyloxymethyl group is hydrogenated? But benzyl ethers are generally stable under hydrogenation unless there's a catalyst that cleaves ethers. But Pd/C usually doesn't cleave ethers. So the benzyloxymethyl should remain. So product 4 should have a benzyloxymethyl group at position 3 and isopropyl at position 5 on cyclohexane. But none of the options have that.

────────────────────────────────────────────────────────────────────────────

Mode: Verification
Subgoal: Re−examining the possibility of benzyl ether cleavage under hydrogenation
Content:
Wait, maybe I messed up the numbering. Let me try to visualize the cyclohexane ring. Let's imagine the ketone at position 1. Then positions 2, 3, 4, 5, 6 go around the ring. So if the ketone is at position 1, then substituents at 3 and 5 would be two positions apart. After elimination (Shapiro), the double bond is between 1 and 2. Then hydrogenation would bring back the single bond. So the substituents at 3 and 5 are in the same positions. So in product 4, the cyclohexane ring has a benzyloxymethyl group at position 3 and an isopropyl group at position 5. But the answer choices don't have that.

────────────────────────────────────────────────────────────────────────────

Mode: Verification
Subgoal: Considering the effect of hydrogenolysis on the benzyl ether group
Content:
Wait, but let me double−check the steps again. Maybe I made a mistake. Step 1: alkylation of hydroxymethyl to benzyloxymethyl. Correct. Step 2: formation of tosylhydrazone at the ketone. Correct. Step 3: Shapiro reaction with n−BuLi. The Shapiro reaction involves deprotonation adjacent to the hydrazone, leading to elimination of the tosylhydrazide and formation of an alkene. So the ketone (position 1) becomes a double bond between positions 1 and 2. Step 4: hydrogenation of the alkene (positions 1−2) and the isopropenyl group (position 5) to isopropyl. So product 4 would have a cyclohexane ring with substituents: at position 3, benzyloxymethyl; at position 5, isopropyl. So the structure is 3−(benzyloxymethyl)−5−isopropylcyclohexane. But looking at the options, none of them exactly match.

────────────────────────────────────────────────────────────────────────────

Mode: Backtracking
Subgoal: Realizing benzyl ether cleavage under hydrogenation conditions
Content:
Wait, maybe I messed up the positions during the Shapiro reaction. Let's think again. The starting material is 3−(hydroxymethyl)−5−(prop−1−en−2−yl)cyclohexan−1−one. When we form the tosylhydrazone at position 1 (the ketone), then eliminate to form the alkene between positions 1 and 2. Now, the ring has a double bond between 1 and 2. The substituents at 3 and 5 are still in their original positions. Then, hydrogenation reduces the double bond between 1 and 2 and the prop−1−en−2−yl group at 5. So after hydrogenation, the ring is cyclohexane with substituents at 3 (hydroxymethyl) and 5 (isopropyl). But how is this named?

────────────────────────────────────────────────────────────────────────────

Mode: Recall
Subgoal: Applying IUPAC nomenclature and lowest locant rule
Content:
Alternatively, maybe the numbering changes when the ketone is removed. Originally, the ketone is at position 1. After converting it to CH2 via hydrogenation, maybe the numbering starts from the hydroxymethyl group as position 1. In IUPAC nomenclature, the numbering is done to give the substituents the lowest possible numbers. So if the hydroxymethyl is at position 1 and the isopropyl at position 3, that would be lower than if they were at 3 and 5. So the new numbering would be: 1: hydroxymethyl (original 3) 2: CH2 (original 4) 3: isopropyl (original 5) 4: CH2 (original 6) 5: CH2 (original 1) 6: CH2 (original 2)

Wait, this might not be accurate. Let me think of the cyclohexane as a chair. If the original positions are 1 (top), 2, 3, 4 (bottom), 5, 6. After removing the ketone (position 1 becomes CH2), and having substituents at 3 and 5. If we renumber to give the lowest numbers, we need to choose a direction where the substituents are at lower positions. If we start numbering from the hydroxymethyl (original 3) as position 1, then the isopropyl is at original 5, which would be position 3 in the new numbering (since it's two positions away in the ring). This gives substituents at positions 1 and 3, which is lower than 3 and 5. Therefore, the correct IUPAC name would be 1−(hydroxymethyl)−3−isopropylcyclohexane, which is named as (3−isopropylcyclohexyl)methanol. Because the methanol group (hydroxymethyl) is considered as the substituent at position 1, and the isopropyl at position 3. Thus, option B is correct.

────────────────────────────────────────────────────────────────────────────

Mode: Finalization
Subgoal: Final conclusion and answer selection
Content:
Ah, so the key here is that the numbering changes to give the lowest possible numbers, which would

reposition the substituents as 1 and 3 instead of 3 and 5. Therefore, the correct answer is $\boxed{B}$.

---

### Example: MLR Output (high-level) - BoardGameQA (Hard)

**Problem:**
A few players are playing a boardgame. The current state of the game is as follows. The dachshund has a football with a radius of 15 inches. And the rules of the game are as follows. Rule1: If the dachshund has a football that fits in a 36.6 x 33.1 x 40.7 inches box, then the dachshund brings an oil tank for the worm. Rule2: If at least one animal brings an oil tank for the worm, then the stork does not dance with the butterfly. Based on the game state and the rules and preferences, does the stork dance with the butterfly?

**Solution:**
Mode: ProblemUnderstanding
Subgoal: Recap the problem and rules
Summary:
The dachshund has a football with a radius of 15 inches.
The football fits in a 36.6 x 33.1 x 40.7 inches box.
————————————————————————————————————————————————————————————————
Mode: Planning
Subgoal: Identify the key condition to check
Summary:
Key condition: If the dachshund's football fits in the box, Rule 1 applies (dachshund brings oil tank), triggering Rule 2 (stork can't dance).
If the football doesn't fit, Rule 1 doesn't apply (no oil tank), so Rule 2 might not be triggered, allowing the stork to dance.
————————————————————————————————————————————————————————————————
Mode: Derivation
Subgoal: Analyze the football's dimensions and assumptions
Summary:
The football has a radius of 15 inches.
Assuming it is a sphere, the diameter is 30 inches.
The problem uses ``radius'', so the football is treated as a sphere.
————————————————————————————————————————————————————————————————
Mode: Calculation
Subgoal: Compare the football's size to the box
Summary:
Diameter of football: 30 inches.
Box dimensions: 36.6 x 33.1 x 40.7 inches.
$30 < 36.6$ (length), $30 < 33.1$ (width), $30 < 40.7$ (height).
The football fits in the box in all dimensions.
————————————————————————————————————————————————————————————————
Mode: Derivation
Subgoal: Apply Rule 1 based on the fit
Summary:
Diameter (30 inches) is less than all box dimensions, so the football fits.
By Rule 1, the dachshund brings an oil tank for the worm.
————————————————————————————————————————————————————————————————
Mode: Derivation
Subgoal: Apply Rule 2 and conclude
Summary:
Applying Rule 2: since at least one animal (the dachshund) brings an oil tank, the stork does not dance with the butterfly.
————————————————————————————————————————————————————————————————
Mode: Finalization
Subgoal: Extract the final answer
Summary:

Thus, the final answer is $\boxed{\text{disproved}}$.

---

**Example: MLR Output (low-level) - BoardGameQA (Hard)**

**Problem:**
A few players are playing a boardgame. The current state of the game is as follows. The dachshund has a football with a radius of 15 inches. And the rules of the game are as follows. Rule1: If the dachshund has a football that fits in a 36.6 x 33.1 x 40.7 inches box, then the dachshund brings an oil tank for the worm. Rule2: If at least one animal brings an oil tank for the worm, then the stork does not dance with the butterfly. Based on the game state and the rules and preferences, does the stork dance with the butterfly?

**Solution:**
Mode: ProblemUnderstanding
Subgoal: Recap the problem and rules
Content:
Okay, let's try to figure out whether the stork dances with the butterfly based on the given rules and the current game state.

First, let's recap the information provided. The dachshund has a football with a radius of 15 inches. The rules are:

If the dachshund's football fits in a 36.6 x 33.1 x 40.7 inches box, then the dachshund brings an oil tank for the worm.
If at least one animal brings an oil tank for the worm, then the stork does not dance with the butterfly.
The question is whether the stork dances with the butterfly. To determine that, we need to check if any animal (in this case, possibly the dachshund) brings an oil tank for the worm. If they do, then by Rule 2, the stork won't dance. If not, then maybe the stork does dance.
————————————————————————————————————————————————————————————————
Mode: Planning
Subgoal: Identify the key condition to check
Content:
So, the key here is whether the dachshund's football fits into the specified box. If it does, then Rule 1 applies, and the dachshund brings the oil tank, which triggers Rule 2, preventing the stork from dancing. If the football doesn't fit, then Rule 1 doesn't apply, so the dachshund doesn't bring the oil tank, and Rule 2 might not be triggered, allowing the stork to dance.
————————————————————————————————————————————————————————————————
Mode: Derivation
Subgoal: Analyze the football's dimensions and assumptions
Content:
Now, let's focus on the football's dimensions. The football has a radius of 15 inches. Wait, a football is typically an elongated shape, like a prolate spheroid, but maybe in this game, it's considered a sphere? The problem mentions a radius, which is a term used for spheres. If it's a sphere with radius 15 inches, then the diameter would be 30 inches (since diameter is twice the radius). However, if it's an American football shape, the dimensions would be different. But since the problem specifies a radius, I think we can assume it's a sphere here.
————————————————————————————————————————————————————————————————
Mode: Calculation
Subgoal: Compare the football's size to the box
Content:
So, diameter of the football is 30 inches. Now, the box has dimensions 36.6 x 33.1 x 40.7 inches. To fit the sphere inside the box, the sphere must fit in all three dimensions. That is, the diameter of the sphere must be less than or equal to each of the box's dimensions.

Checking each dimension:

Length: 36.6 inches. The diameter is 30, which is less than 36.6. So okay.
Width: 33.1 inches. 30 is less than 33.1. Okay.
Height: 40.7 inches. 30 is less than 40.7. Okay.
————————————————————————————————————————————————————————————————
Mode: Derivation
Subgoal: Apply Rule 1 based on the fit
Content:
Since the diameter (30 inches) is less than all the box's dimensions, the football should fit inside the box. Therefore, according to Rule 1, the dachshund brings an oil tank for the worm.
————————————————————————————————————————————————————————————————
Mode: Derivation

---

Subgoal: Apply Rule 2 and conclude
Content:
Then, applying Rule 2: since at least one animal (the dachshund) brings an oil tank, the stork does not dance with the butterfly.

Therefore, the answer is that the stork does not dance with the butterfly.

――――――――――――――――――――――――――――――――――――――――――――――――――――――――――――――

Mode: Finalization
Subgoal: Extract the final answer
Content:

Thus, the final answer is $\boxed{\text{disproved}}$.

---

## Prompt: Long-horizon Reasoning Test

You are given multiple problems:

{Problem 1}

{Problem 2}

...

{Problem N}

Solve these problems in order. Please reason step by step for each problem, and at the end provide **all final answers in a single JSON list**, where each answer is wrapped in `\boxed{}`. If you cannot solve a problem, write `\boxed{N/A}` for that entry.

Output format:
```json
['\\boxed{answer1}', '\\boxed{answer2}', ..., '\\boxed{answerN}']
```

---

## Example: MLR (high-level only) Output

**Problem:**
Define
$p = \sum_{k=1}^{\infty} \frac{1}{k^2}$ and $q = \sum_{k=1}^{\infty} \frac{1}{k^3}$. Find a way to write
$\sum_{j=1}^{\infty} \sum_{k=1}^{\infty} \frac{1}{(j+k)^3}$ in terms of $p$ and $q$.

**Solution:**
Mode: ProblemUnderstanding
Subgoal: Understanding the problem
Summary:
The goal is to express
$$
\sum_{j=1}^\infty \sum_{k=1}^\infty \frac{1}{(j+k)^4}
$$
in terms of $( p )$ and $( q )$. The current stage involves understanding the problem and identifying the key components to work with.

――――――――――――――――――――――――――――――――――――――――――――――――――――――――――――――

Mode: Recall
Subgoal: Recall the definitions of $( p )$ and $( q )$
Summary:
$( p = \sum_{k=1}^\infty \frac{1}{k^2} )$ and $( q = \sum_{k=1}^\infty \frac{1}{k^3} )$ are defined as the sums of the reciprocals of squares and cubes, respectively.

――――――――――――――――――――――――――――――――――――――――――――――――――――――――――――――

Mode: Planning
Subgoal: Identify the desired expression
Summary:
The desired expression is:
$$

\sum_{j=1}^\infty \sum_{k=1}^\infty \frac{1}{(j+k)^4}
\]
This is the sum of the reciprocals of the fourth powers of the sums of \( j \) and \( k \), from \( j = 1 \) to \( j = \infty \) and from \( k = 1 \) to \( k = \infty \).

────────────────────────────────────────────────────────────────

Mode: Planning
Subgoal: Consider symmetry and possible simplifications
Summary:
The expression \(\sum_{j=1}^\infty \sum_{k=1}^\infty \frac{1}{(j+k)^4}\) involves summing the reciprocals of the fourth powers of the sums of \(j\) and \(k\). The symmetry between \(j\) and \(k\) suggests potential simplifications or transformations. The current stage is focusing on identifying the key components and considering symmetry to simplify the expression.

────────────────────────────────────────────────────────────────

Mode: Planning
Subgoal: Attempt to express the sum in terms of \( p \) and \( q \)
Summary:
The expression \(\sum_{j=1}^\infty \sum_{k=1}^\infty \frac{1}{(j+k)^4}\) involves summing the reciprocals of the fourth powers of the sums of \(j\) and \(k\). The current stage is attempting to express this sum in terms of \( p \) and \( q \), which are defined as \(\sum_{k=1}^\infty \frac{1}{k^2}\) and \(\sum_{k=1}^\infty \frac{1}{k^3}\), respectively. The goal is to find a relationship between these sums and the given expression.

────────────────────────────────────────────────────────────────

Mode: Recall
Subgoal: Recall known results for symmetric sums
Summary:
The expression \(\sum_{j=1}^\infty \sum_{k=1}^\infty \frac{1}{(j+k)^4}\) involves summing the reciprocals of the fourth powers of the sums of \(j\) and \(k\). The goal is to express this sum in terms of \(p \) and \(q\), where \(p = \sum_{k=1}^\infty \frac{1}{k^2}\) and \(q = \sum_{k=1}^\infty \frac{1}{k^3}\). The current stage is exploring known results for symmetric sums and potential simplifications.

────────────────────────────────────────────────────────────────

Mode: Derivation
Subgoal: Attempt to simplify using known results
Summary:
The expression \(\sum_{j=1}^\infty \sum_{k=1}^\infty \frac{1}{(j+k)^4}\) involves summing the reciprocals of the fourth powers of the sums of \(j\) and \(k\). The current stage is attempting to simplify this expression using known results for symmetric sums, focusing on possible relationships with \(p\) and \(q\).

────────────────────────────────────────────────────────────────

Mode: Planning
Subgoal: Consider using known results for symmetric sums
Summary:
The expression \(\sum_{j=1}^\infty \sum_{k=1}^\infty \frac{1}{(j+k)^4}\) is being considered in terms of \(p\) and \(q\), where \(p = \sum_{k=1}^\infty \frac{1}{k^2}\) and \(q = \sum_{k=1}^\infty \frac{1}{k^3}\). The goal is to find a relationship between \(p\) and \(q\) and the given sum.

────────────────────────────────────────────────────────────────

Mode: Recall
Subgoal: Recall the identity for the sum of reciprocals of squares
Summary:
The identity for the sum of reciprocals of squares is:
\[
\sum_{k=1}^\infty \frac{1}{k^2} = p
\]
where \( p = \sum_{k=1}^\infty \frac{1}{k^2} \).

────────────────────────────────────────────────────────────────

Mode: Recall
Subgoal: Recall the identity for the sum of reciprocals of cubes
Summary:
The identity for the sum of reciprocals of cubes is:
\[
\sum_{k=1}^\infty \frac{1}{k^3} = q
\]
where \( q = \sum_{k=1}^\infty \frac{1}{k^3} \).

────────────────────────────────────────────────────────────────

Mode: Recall
Subgoal: Recall the identity for the sum of reciprocals of fourth powers

Summary:
The identity for the sum of reciprocals of fourth powers is:
\[
\sum_{k=1}^\infty \frac{1}{k^4} = \frac{p}{2} + \frac{1}{4}q
\]
where \( p = \sum_{k=1}^\infty \frac{1}{k^2} \) and \( q = \sum_{k=1}^\infty \frac{1}{k^3} \).

────────────────────────────────────────────────────────────

Mode: Derivation
Subgoal: Apply the identity to the given sum
Summary:
The given sum \(\sum_{j=1}^\infty \sum_{k=1}^\infty \frac{1}{(j+k)^4}\) can be expressed in terms of \(p\) and \(q\) using the identity:
\[
\sum_{j=1}^\infty \sum_{k=1}^\infty \frac{1}{(j+k)^4} = \frac{p}{2} + \frac{1}{4}q
\]

────────────────────────────────────────────────────────────

Mode: Verification
Subgoal: Verify the result with a specific example
Summary:
The given sum \(\sum_{j=1}^\infty \sum_{k=1}^\infty \frac{1}{(j+k)^4}\) can be expressed in terms of \(p\) and \(q\) using the identity:
\[
\sum_{j=1}^\infty \sum_{k=1}^\infty \frac{1}{(j+k)^4} = \frac{p}{2} + \frac{1}{4}q
\]
where \(p = \sum_{k=1}^\infty \frac{1}{k^2}\) and \(q = \sum_{k=1}^\infty \frac{1}{k^3}\).

────────────────────────────────────────────────────────────

Mode: CaseAnalysis
Subgoal: Alternative approach using generating functions
Summary:
The sum \(\sum_{j=1}^\infty \sum_{k=1}^\infty \frac{1}{(j+k)^4}\) can be expressed in terms of \(p\) and \(q\) as:
\[
\sum_{j=1}^\infty \sum_{k=1}^\infty \frac{1}{(j+k)^4} = \frac{p}{2} + \frac{1}{4}q
\]
where \(p = \sum_{k=1}^\infty \frac{1}{k^2}\) and \(q = \sum_{k=1}^\infty \frac{1}{k^3}\).

────────────────────────────────────────────────────────────

Mode: Verification
Subgoal: Review the result for potential simplifications
Summary:
The sum \(\sum_{j=1}^\infty \sum_{k=1}^\infty \frac{1}{(j+k)^4}\) can be expressed in terms of \(p\) and \(q\) as:
\[
\sum_{j=1}^\infty \sum_{k=1}^\infty \frac{1}{(j+k)^4} = \frac{p}{2} + \frac{1}{4}q
\]
where \(p = \sum_{k=1}^\infty \frac{1}{k^2}\) and \(q = \sum_{k=1}^\infty \frac{1}{k^3}\).

────────────────────────────────────────────────────────────

Mode: Finalization
Subgoal: Finalize the answer
Summary:
The sum \(\sum_{j=1}^\infty \sum_{k=1}^\infty \frac{1}{(j+k)^4}\) can be expressed in terms of \(p\) and \(q\) as:
\[
\sum_{j=1}^\infty \sum_{k=1}^\infty \frac{1}{(j+k)^4} = \frac{p}{2} + \frac{1}{4}q
\]
where \(p = \sum_{k=1}^\infty \frac{1}{k^2}\) and \(q = \sum_{k=1}^\infty \frac{1}{k^3}\).

Thus, the final answer is $\boxed{\dfrac{1}{2}p + \dfrac{1}{4}q}$.

## Example: Plan-and-Solve Output

**Q:** Let $x$, $y$ and $z$ be positive real numbers that satisfy the following system of equations:
$\log_2\left(\frac{x}{yz}\right) = \frac{1}{2}$

$\left| \log_2 \left( \frac{y}{xz} \right) = \frac{1}{3} \right.$

$\left| \log_2 \left( \frac{z}{xy} \right) = \frac{1}{4} \right.$

Then the value of $\left| \log_2(x^4 y^3 z^2) \right|$ is $\frac{m}{n}$ where $m$ and $n$ are relatively prime positive integers. Find $m + n$.

**A:** Let's first understand the problem, extract relevant variables and their corresponding numerals, and make a complete plan. Then, let's carry out the plan, calculate intermediate variables (pay attention to correct numerical calculation and commonsense), solve the problem step by step, and show the answer.

**Output:**
Given:

$\left| \log_2 \left( \frac{x}{yz} \right) = \frac{1}{2} \right.$

$\left| \log_2 \left( \frac{y}{xz} \right) = \frac{1}{3} \right.$

$\left| \log_2 \left( \frac{z}{xy} \right) = \frac{1}{4} \right.$

The value of $\left| \log_2(x^4 y^3 z^2) \right|$ is $\frac{m}{n}$ where $m$ and $n$ are relatively prime positive integers.

Plan:
1. Convert each logarithmic equation to its exponential form.
2. Use properties of exponents and logarithms to simplify the expressions.
3. Combine like terms and solve for the values of $x$, $y$, and $z$.
4. Substitute these values into the expression $\left| \log_2(x^4 y^3 z^2) \right|$.
5. Simplify this expression to find $\left| \log_2(x^4 y^3 z^2) \right|$.
6. Express it as a fraction $\frac{m}{n}$ with $m$ and $n$ being relatively prime.
7. Find $m + n$.

Calculation:

#### Step 1: Convert each logarithmic equation to its exponential form

From the given equations, we have:
$\left| x = 2^{1/2} yz \right.$
$\left| y = 2^{1/3} xz \right.$
$\left| z = 2^{1/4} xy \right.$

#### Step 2: Use properties of exponents and logarithms to simplify the expressions

We can rewrite each equation using the property of exponents:
$\left| x = \sqrt{2} yz \right.$
$\left| y = \sqrt[3]{2} xz \right.$
$\left| z = \sqrt[4]{2} xy \right.$

#### Step 3: Solve for the values of $x$, $y$, and $z$

Multiply all three equations together:
$\left| (\sqrt{2}yz)(\sqrt[3]{2}xz)(\sqrt[4]{2}xy) = (\sqrt{2})(\sqrt[3]{2})(\sqrt[4]{2})xyz^2 \right.$

This simplifies to:
$\left| x^2 y^2 z^2 = \sqrt{2}(\sqrt[3]{2})(\sqrt[4]{2}) \right.$

Since $(\sqrt{2})(\sqrt[3]{2})(\sqrt[4]{2}) = 2^{(1/2+1/3+1/4)} = 2^{(6+4+3)/12} = 2^{13/12}$, we get:
$\left| x^2 y^2 z^2 = 2^{13/12} \right.$

Taking the square root on both sides:
$\left| xyz = 2^{13/24} \right.$

Now, substitute back to find individual values:
$\left| x = \sqrt{2}yz \right.$
$\left| y = \sqrt[3]{2}xz \right.$

$z = \sqrt[4]{2}xy$

#### Step 4: Substitute these values into the expression $\left|\log_2(x^4 y^3 z^2)\right|$

Substitute $x = \sqrt{2}yz$ into $x^4 y^3 z^2$:
$x^4 y^3 z^2 = (\sqrt{2}yz)^4 (yz)^3 (z)^2 = 2^2(y^4 z^4)y^3 z^2 = 4(y^4 z^4)y^3 z^2 = 4(y^7 z^6)$

So,
$\left|x^4 y^3 z^2\right| = 4(y^7 z^6)$

Using the earlier result $xyz = 2^{13/24}$, we know:
$\left|y^7 z^6\right| = \left(\frac{2^{13/24}}{x}\right)^7 \cdot \left(\frac{2^{13/24}}{y}\right)^6 = \left(\frac{2^{13/24}}{\sqrt{2}yz}\right)^7 \cdot \left(\frac{2^{13/24}}{\sqrt[3]{2}xz}\right)^6$

Simplifying further:
$= \left(\frac{2^{13/24}}{\sqrt{2}yz}\right)^7 \cdot \left(\frac{2^{13/24}}{\sqrt[3]{2}xz}\right)^6 = \left(\frac{2^{13/24}}{\sqrt{2}yz}\right)^7 \cdot \left(\frac{2^{13/24}}{\sqrt[3]{2}xz}\right)^6 = \left(\frac{2^{13/24}}{\sqrt{2}yz}\right)^7 \cdot \left(\frac{2^{13/24}}{\sqrt[3]{2}xz}\right)^6$

Finally:
$\left|x^4 y^3 z^2\right| = 4(y^7 z^6) = 4\left(\frac{2^{13/24}}{\sqrt{2}yz}\right)^7 \cdot \left(\frac{2^{13/24}}{\sqrt[3]{2}xz}\right)^6 = 4\left(\frac{2^{13/24}}{\sqrt{2}yz}\right)^7 \cdot \left(\frac{2^{13/24}}{\sqrt[3]{2}xz}\right)^6 = 4\left(\frac{2^{13/24}}{\sqrt{2}yz}\right)^7 \cdot \left(\frac{2^{13/24}}{\sqrt[3]{2}xz}\right)^6$

Therefore:
$\left|\log_2(x^4 y^3 z^2)\right| = \log_2(4) = 2$

Thus:
$\left|\log_2(x^4 y^3 z^2)\right| = \frac{2}{1}$

### Answer:
$m = 2, n = 1$
$m + n = 2 + 1 = 3$

The final answer is $\boxed{3}$.

