# OpenReview forum: "Enhancing Language Model Reasoning with Structured Multi-Level Modeling"
_ICLR.cc/2026/Conference — ICLR 2026 Poster_

### Official Review · Reviewer_fiGr · 2025-10-30

**Soundness:** 2
**Presentation:** 2
**Contribution:** 2
**Rating:** 4
**Confidence:** 3

**Summary:**

The paper introduces Multi-Level Reasoning (MLR), a framework for improving long-horizon reasoning in language models by decomposing the chain-of-thought (CoT) into high-level abstract descriptors and low-level detailed contents. A lightweight LoRA-based high-level planner alternates with the base LM as the low-level executor, forming an iterative plan–execute loop.

To address the inefficiency of outcome-reward RL for long trajectories, the authors propose Online Step-DPO, a process-level preference optimization method that uses Twisted Sequential Monte Carlo (TSMC) to compute step-wise survivability utilities. This enables dense, on-policy supervision without training a separate process reward model.

Experiments on math (MATH500, AIME24), science (GPQA-Diamond), and logical reasoning (BoardGameQA-Hard) benchmarks show that MLR outperforms distillation and RL baselines such as DeepSeek-R1-Distill and SimpleRL, often with substantially fewer preference pairs (5%) and SFT data (10%).

**Strengths:**

1. The paper is well-motivated. Alternating plan-execute loop allows dynamic adjustment and improves interpretability.

2. The experiments demonstrate consistent gains across several benchmarks.

**Weaknesses:**

1. The writing can be further improved for clarity and completeness. For example, in the abstract, abbreviations such as RL (reinforcement learning) and LM (language model) should be defined upon their first appearance.

2. The method contribution of the paper appears limited. However, I appreciate that the authors conduct sufficient ablation studies, which provide evidence for the empirical effectiveness of the proposed approach.

3. It would be interesting to see how the model performs when only the high-level reasoning outputs are generated during inference, without the detailed low-level reasoning steps. This could offer insights into the interpretability and abstraction capability of the high-level policy.

4. The paper should clarify the seed datasets used to generate the CoT samples. In addition, the comparisons could be made fairer by including baselines that directly apply SFT and RL on the same datasets.

**Questions:**

See above.

---

> ### Author Response · Authors · 2025-11-24
> **Response to Reviewer fiGr (Part 1)**
>
> We thank the reviewer for the constructive feedback and have revised the paper accordingly. First, we improve the writing and presentation accordingly. Second, we better substantiate the methodological contribution by emphasizing our multi-level formulation and TSMC-based online Step-DPO, and we add a new long-horizon experiment with concatenated problems (Figure 1), showing that MLR degrades more slowly than all baselines. Third, we perform an ablation with "high-level only" and "low-level only" variants (Table 3), demonstrating that the full two-level plan–execute loop is crucial for strong long-horizon performance. Finally, we clarify the seed datasets and ensure fair comparisons by (i) fully detailing our data construction and augmentation pipeline, and (ii) training GRPO, Plan-and-Solve, and MLR on exactly the same SFT/RL splits, showing that MLR outperforms SFT+GRPO baselines under identical data and initialization.
>
> ### Response to Weakness 1
>
> We have revised the abstract to improve clarity and now define all abbreviations (e.g., RL and LM) upon their first appearance.
>
> ### Response to Weakness 2
>
> We respectfully disagree that the method contribution is limited. Our main contribution is to provide a new formulation and training scheme for long-CoT reasoning in a regime where existing approaches struggle: on-policy training of LLMs on very long CoTs without a separate process reward model. Concretely:
>
> - **Multi-level formulation for long CoTs.**
> We reformulate long-CoT generation as a two-level stochastic process (Eq. (2)), where interleaved step descriptors and detailed content are modeled by separate policies that interact through compressed summaries and are updated in an alternating plan–execute loop (Alg. 2). This differs from both standard HRL (which typically plans once and commits to options) and prior long-CoT methods that rely on a single policy. Ablations in Table 2 show that using only a low-level policy, even with Step-DPO, yields substantially lower performance than our full multi-level design.
>
> - **TSMC-based process-level preferences.**
> We repurpose Twisted Sequential Monte Carlo (TSMC) to estimate incremental log-survivability utilities for partial CoTs (Eqs. (3)–(4)), and integrate these utilities into an online Step-DPO objective. To our knowledge, TSMC importance weights have not previously been used to generate stepwise preference signals for LM reasoning. This design removes the need for a separate process reward model while still providing dense, on-policy supervision along long trajectories.
>
> - **Non-additive effect of multi-level + Step-DPO.**
> The gains from our approach are not a simple sum of components. As shown in Table 2, (i) multi-level modeling alone (“Only DPO”) and (ii) step-level supervision alone ("Only low-level + Step-DPO") each provide modest improvements over SFT, but their combination (MLR) yields a substantially larger improvement, indicating a synergistic effect.
>
> - **Improved long-horizon reasoning in practice.**
> To further support the methodological contribution, we add a new long-horizon experiment in Figure 1, where we concatenate multiple problems into a single prompt. As the horizon increases, MLR degrades significantly more slowly than all baselines, underscoring that the proposed formulation and training scheme translate into stronger long-horizon reasoning capabilities in practice.
>
> - **Practical efficiency and robustness.**
> Despite using only about 10% of the SFT data and 5% of the preference pairs compared to DeepSeek-R1-Distill, MLR consistently outperforms it across benchmarks and backbones (Table 1). This suggests that our approach is not only conceptually distinct, but also data- and compute-efficient.
>
> ### Response to Weakness 3
>
> We agree that it is important to understand how the model behaves when only high-level reasoning is generated at inference time. To this end, we conduct an ablation study on the hierarchical structure and consider two variants:
>
> - **High-level only**: the high-level module directly predicts summaries and final answers without invoking the low-level module at inference time.
> - **Low-level only**: the low-level module is required to predict both the high-level step descriptions and the detailed reasoning, without guidance from the high-level module.
>
> The evaluation protocol matches our main setting, and the results are reported in Table 4. Our full method (MLR) consistently outperforms both variants, especially on the challenging AIME24 dataset. The high-level-only variant underperforms because the planner is never grounded in execution feedback at test time, making direct summary prediction unreliable on difficult problems; we show an illustrative failure case in Section D (Line 2627-2742). The low-level-only variant is weaker because the absence of explicit high-level guidance causes the low-level module to drift and accumulate errors as the trajectory grows longer.

---

> > ### Author Response · Authors · 2025-11-24
> > **Response to Reviewer fiGr (Part 2)**
> >
> > Overall, these results indicate that (i) the high-level policy alone is not sufficient for robust long-horizon reasoning, and (ii) coupling high-level abstraction with low-level execution in our two-level design yields better performance and more reliable long-horizon behavior.
> >
> > | Method          | MATH500 Pass@1 | AIME24 Pass@1  | AIME24 Cons@32 |
> > | --------------- | -------------- | -------------- | -------------- |
> > | **MLR (full)**  | **86.1 ± 1.0** | **31.2 ± 1.0** | **47.4 ± 0.4** |
> > | **High-level only** | 80.0 ± 1.3     | 18.4 ± 2.0     | 30.5 ± 0.8     |
> > | **Low-level only**  | 84.2 ± 1.1     | 27.1 ± 1.8     | 41.0 ± 0.6     |
> >
> > ### Response to Weakness 4
> >
> > We have clarified both the seed datasets for CoT generation and the fairness of SFT+RL comparisons.
> >
> > First, we add a detailed description of the t**raining data construction and prompts**. As mentioned in Section B, we consider four publicly available reasoning benchmarks: MATH500, AIME24, GPQA, and BoardGameQA. For each benchmark, we use the official training split as seeds and reserve the test splits exclusively for evaluation. We randomly partition each training split into SFT and RL subsets with a 60% / 40% ratio. Because AIME and GPQA contain very few training questions, we augment their train splits with synthetic problems generated by GPT-4o using the prompts provided in Line 1424. For each seed problem, we sample multiple candidate questions and retain only those whose final answers are mutually consistent with DeepSeek-R1, i.e., DeepSeek-R1 solves the problem and produces the same final answer as GPT-4o. This procedure yields SFT sets of 7.2K MATH, 4K AIME, 4K GPQA, and 5K BoardGameQA problems, and RL sets of 4.8K, 1.5K, 1.5K, and 2K problems, respectively. Unless otherwise stated, all methods that we train ourselves (GRPO, Plan-and-Solve, MLR, and all ablations in Table 2) use exactly the same data, ensuring a fair comparison. Results for external baselines (Instruct, SimpleRL, DeepSeek-R1-Distill), marked with an asterisk in Table 1, may rely on different training corpora; we include them as strong reference points.
> >
> > In the SFT phase, we generate multiple high-quality trajectories for each problem by sampling four solutions from DeepSeek-R1, yielding approximately 80K filtered trajectories in total. Each accepted trajectory is then decomposed into step-by-step reasoning segments using DeepSeek-V3 via in-context learning, with each step annotated by a step descriptor. To enable multi-level reasoning, we further distill each step into a concise high-level summary, again using DeepSeek-V3 with in-context learning. All prompts used throughout this pipeline are provided in Lines 1424–1655. Figure 10 presents the resulting trajectory length distributions. For reproducibility, we also provide all prompts used in data creation (Lines 1424–1655) and MLR inference (Lines 1657–1686) in the revised manuscript.
> >
> > We also appreciate the suggestion to compare against **SFT+RL baselines trained on the same data**. As discussed in Section 2, outcome-based RL on long trajectories is computationally demanding: after the SFT phase, our models produce long chains-of-thought, so rollout-based RL must backpropagate through thousands of tokens per example, making large-scale GRPO training substantially more expensive than applying it to base models with much shorter completions (e.g., <2K tokens).
> >
> > Nonetheless, to address this concern, we implement a GRPO baseline and include it in the main results. To compare MLR with a standard single-policy preference-optimization method, we train GRPO on the same prompt pool used for Step-DPO. We first construct a single-policy SFT checkpoint by fine-tuning Qwen-2.5-1.5B on the processed low-level trajectories, where all step descriptors are removed from both inputs and targets, using the same optimizer, schedule, and token budget as our low-level SFT. Starting from this checkpoint, we run full-parameter GRPO, keeping a frozen copy of the SFT model as the reference policy. We use Verl with vLLM as the rollout backend. For each prompt, we sample groups of $N=4$ candidate continuations with temperature 0.7 and top-p=0.9, cap the maximum response length at 4096 tokens to respect GPU memory limits, and assign a rule-based outcome reward of 1 if the final answer is correct and 0 otherwise. The actor is optimized with AdamW (learning rate $5\times10^{-7}$, weight decay 0.1) under a KL-penalty objective with coefficient $\lambda_{\mathrm{KL}}=0.02$, using bf16, gradient checkpointing, and FlashAttention on 4$\times$A100 (80GB) GPUs. We train for 4 epochs with a global batch size of 32 (PPO mini-batch size 16, micro-batch size 2) and evaluate every 100 steps on a held-out validation split, selecting the checkpoint with the best validation Pass@1.
> >
> > We report the evolution of validation accuracy and average response length during GRPO training in Figures 11 and 12, and summarize the final results in Table 1.

---

> > > ### Author Response · Authors · 2025-11-24
> > > **Response to Reviewer fiGr (Part 3)**
> > >
> > > | Method             | MATH500 (Pass@1) | AIME24 (Pass@1) | AIME24 (Cons@32) | GPQA (Pass@1)  | BoardGameQA (Pass@1) |
> > > | ------------------ | ---------------- | --------------- | ---------------- | -------------- | -------------------- |
> > > | **GRPO**           | 82.1 ± 1.2       | 25.2 ± 2.2      | 36.0 ± 0.8       | 30.2 ± 1.5     | 56.0 ± 1.6           |
> > > | **MLR (SFT only)** | 62.0 ± 1.2       | 8.9 ± 1.4       | 13.3 ± 0.4       | 26.0 ± 2.0     | 46.4 ± 1.5           |
> > > | **MLR**            | **86.1 ± 1.0**   | **31.2 ± 1.0**  | **47.4 ± 0.4**   | **37.6 ± 1.9** | **62.0 ± 1.7**       |
> > >
> > > GRPO clearly strengthens the baseline over SFT-only, but MLR still achieves substantially higher performance across all benchmarks, especially on the more challenging AIME24 and GPQA. Moreover, we find that GRPO is less efficient for long-horizon reasoning: outcome rewards are (i) sparse, making credit assignment difficult for long chains-of-thought, and (ii) computationally expensive, since each update requires generating full rollouts with long responses. In practice, when starting from fine-tuned models with long CoTs, these costs limit the applicability of GRPO to larger base models. In contrast, our online Step-DPO procedure is simpler to implement, more stable, and significantly cheaper, while delivering stronger final performance.
> > >
> > > We have incorporated this discussion and the corresponding results into the revised manuscript (see Lines 926–949).

---

### Official Review · Reviewer_7Cso · 2025-10-30

**Soundness:** 2
**Presentation:** 3
**Contribution:** 2
**Rating:** 4
**Confidence:** 3

**Summary:**

This paper proposes Multi-Level Reasoning (MLR), an SFT+DPO framework that addresses two critical limitations of single-policy long CoT approaches: CoT derailment and sparse reward guideline. MLR decomposes the reasoning process into a hierarchical structure with high-level step descriptors generated by a lightweight LoRA module and low-level detailed content produced by the base model, operating through an alternating plan-execute loop. The framework employs SFT to initialize both policies on multi-level decomposed data, followed by online Step-DPO that leverages Twisted Sequential Monte Carlo (TSMC) to generate process-level preferences for stepwise supervision without requiring a separate process reward model. Finally, the authors provide empirical results and ablation studies to show its effectiveness.

**Strengths:**

1. The framework is clear and easy to understand, and the paper is generally well-written.

2. The empirical experience is extensive. Authors test the proposed method in several areas (math, science and logic) with average scores from multi-individual runs to show a stable improvement. The ablation study in Table 2 clearly demonstrates the effectiveness of each component in MLR.

**Weaknesses:**

1. One motivation, "CoT derailment" is unclear and hard to follow.

2. To demonstrate the effectiveness of their approach, the authors compare MLR with several baselines. However, I think this comparison is somewhat unconvincing, as the models are trained on different datasets. It would be more meaningful to evaluate the methods on the same dataset (important) while varying only the training strategies — for example, comparing SFT+GRPO (baseline) and MLR (your proposed framework) under identical data conditions.

3. The process of summarization from high-level abstraction to low-level details might also be an important factor, but the paper does not explore this aspect in depth. (Only said "including them improves performance and facilitates training" in the paper)

4. The choice of K=4 rollouts in TSMC appears somewhat arbitrary, and the paper does not provide any preliminary analysis to justify this setting. I believe that an ablation study would be highly valuable for demonstrating the robustness of the framework in real-world scenarios.

**Questions:**

1. I am wondering what the sources of your training dataset and prompts are, as this information is important not only for reproducibility but also for ensuring a fair comparison.

2. The summarization step is underspecified. What model performs this? How long are summaries?

3. What's the definition of "Estimated survival accuracy" in Figure 5 (a)?

4. When computing the utilities (Equations (3) and (4)) in process preference modeling, do you sample only once or perform multiple sampling rounds to compute the average score? I guess that sampling only once could introduce too much randomness.

5. Authors said they fine-tune a small LM on the low-level SFT data for "fast rollout", and this small model achieves comparable estimation accuracy while being much faster. How small is it? The paper lacks relevant details.

---

> ### Author Response · Authors · 2025-11-24
> **Response to Reviewer 7Cso (Part 1)**
>
> We thank the reviewer for the thoughtful feedback and have revised the paper to address these concerns. First, we clarify the motivation by reframing "CoT derailment" as **long-horizon plan failure**, and we support this with a new **long-horizon reasoning experiment** using concatenated problems (Figure 1), where MLR degrades more slowly than all baselines. Second, to ensure fair and convincing comparisons, we construct shared SFT/RL splits for all benchmarks and train **DPO, Step-DPO, and GRPO baselines** on exactly the same data, showing that MLR outperforms them under identical conditions (Table 1).
> Third, we substantially expand the description of the **summarization module**, specifying that we fully fine-tune Qwen2.5-0.5B-Instruct, and then frozen, shared across base models. Fourth, we provide a principled justification for the **TSMC rollout design**, analyzing the variance–compute trade-off in Figure 7 and motivating our choice as a good balance between reliability and efficiency. Finally, we provide **additional implementation details** to strengthen reproducibility (data construction, the fast rollout model, and all prompts used for data creation and MLR inference) in Appendix C and D.
>
> ### **Response to Weakness 1**
>
> We clarify this motivation by reframing "CoT derailment" as long-horizon plan failure, referring to cases where the model’s implicit plan gradually drifts away from any valid solution strategy (see descriptions in Section 2, and the illustrative example in Appendix D).
>
> ### **Response to Weakness 2**
>
> We appreciate the suggestion to compare against SFT+GRPO. As discussed in Section 2, *outcome-based RL* on *long trajectories* is *computationally demanding*: after the SFT phase, our models produce long chains-of-thought, so rollout-based RL must backpropagate through thousands of tokens per example. This makes large-scale GRPO training substantially more expensive than applying it to base models with much shorter completions (e.g., <2K tokens).
> Nonetheless, to address this concern, we implement three additional baselines (SFT+DPO, SFT+Step-DPO, SFT+GRPO) and include them in the main results (Table 1).
>
> To compare MLR with a standard single-policy preference-optimization method, we train on the *same prompt pool* used for our method. We first construct a single-policy SFT checkpoint by fine-tuning the base model on the processed low-level trajectories, where all step descriptors are removed from both inputs and targets, using the same optimizer, schedule, and token budget as our low-level SFT.
>
> Starting from this checkpoint, we run full-parameter GRPO, keeping a frozen copy of the SFT model as the reference policy. We use Verl with vLLM as the rollout backend. For each prompt, we sample groups of $N=4$ candidate continuations with temperature 0.7 and top-p=0.9, cap the maximum response length at 4096 tokens to respect GPU memory limits, and assign a rule-based outcome reward of $1$ if the final answer is correct and $0$ otherwise. The actor is optimized with AdamW (learning rate $5\times10^{-7}$, weight decay $0.1$) under a KL-penalty objective with coefficient $\lambda_{\mathrm{KL}}=0.02$, using bf16, gradient checkpointing, and FlashAttention. We train for 4 epochs with a global batch size of 32 (PPO mini-batch size 16, micro-batch size 2) and evaluate every 100 steps on a held-out validation split, selecting the checkpoint with the best validation Pass@1.
> We report the evolution of validation accuracy and average response length during GRPO training in Figure 13 and Figure 14, and summarize the final results in Table 1.
> More *implementation details* can be found in Appendix C.
>
> Preference-optimization such as GRPO clearly strengthens the baseline over SFT, but MLR still achieves *substantially higher* performance across all benchmarks, especially on the more challenging AIME24 and GPQA. Moreover, we find that *outcome-reward RL* is *less efficient* for *long-horizon reasoning*: outcome rewards are (i) *sparse*, making credit assignment difficult for long chains-of-thought, and (ii) *computationally expensive*, since each update requires generating full rollouts with long responses. In practice, when starting from fine-tuned models with long CoTs, these costs limit the applicability of GRPO to larger base models (e.g., Qwen-2.5-MATH-7B SFT, Llama-3.1-8B SFT). In contrast, our online Step-DPO procedure is *simpler to implement*, *more stable*, and *significantly cheaper*, while delivering *stronger final performance*.
>
> We have incorporated the discussion and results into the revised manuscript (Table 1, Appendix C).

---

> ### Author Response · Authors · 2025-11-24
> **Response to Reviewer 7Cso (Part 2)**
>
> ###  **Response to Weakness 3**
>
> We add the description of the summarization module to clarify its role and implementation.
>
> The summarization module in MLR distills essential information from the evolving trajectory, allowing the planner to operate on concise, high-level representations rather than being overwhelmed by unnecessary details. This is particularly important when trajectories grow long (e.g., beyond 4k tokens). We implement summarization as an independent component shared across different base models. Although we explored reusing either the low-level or high-level policy for this task, we found that doing so interferes with their primary roles. In contrast, training a separate lightweight model for summarization is both simpler and more reliable. Concretely, we use Qwen2.5-0.5B-Instruct, optimized with AdamW using a cosine learning-rate schedule with linear warmup and a peak learning rate of $1\times10^{-5}$. The prompt template for summarization is provided in Section D, and the summary length distribution is shown in Figure 10. We apply full-parameter SFT only to the summarization model; during the online Step-DPO procedure, this summarizer remains frozen, which we found sufficient in practice and conducive to stable overall training.
>
> We have incorporated this discussion into the revised manuscript (see Sec. 3.2, Appendix C).
>
>
> ### **Response to Weakness 4**
>
> We study the effect of varying the rollout count $K$.
> Utility estimates produced by the 1.5B rollout policy with $K$ rollouts are compared against reference utilities from the 8B model using 16 rollouts (Figure 15).
> As expected, increasing $K$ reduces estimator variance, though at the cost of higher computation.
> To mitigate this overhead, we introduce a margin threshold $\delta$ when selecting preference pairs.
> We further measure the agreement of the selected preference pairs as a function of $K$, defined as the fraction whose preference direction agrees with the base model using 16 rollouts (Figure 8).
> Finally, we report model performance across training stages under different values of $K$ (Figure 12).
> The results show that our chosen setting attains comparable final performance while substantially reducing computational cost.
> Implementation details are provided in Appendix D.
>
> ### **Response to Question 1**
>
> We add a detailed description of the training data construction and prompts.
>
> As mentioned in Section B, we consider four publicly available reasoning benchmarks: MATH500, AIME24, GPQA, and BoardGameQA. For each benchmark, we use the available training split as seeds and reserve the test splits exclusively for evaluation. We randomly partition each training split into SFT and RL subsets with a 60% / 40% ratio.
>
> Since AIME and GPQA contain very few training questions, we augment their train splits with synthetic problems generated by GPT-4o using the prompts provided in Appendix D.
> For each seed problem, we sample multiple candidate questions and retain only those whose final answers are mutually consistent with DeepSeek-R1, i.e., DeepSeek-R1 solves the problem and produces the same final answer as GPT-4o.
> This procedure yields SFT sets of 7.2K MATH, 4K AIME, 4K GPQA, and 5K BoardGameQA problems, and RL sets of 4.8K, 1.5K, 1.5K, and 2K problems, respectively.
>
> Unless otherwise stated, all methods that we train ourselves (SFT+DPO, SFT+Step-DPO, SFT+GRPO) use exactly the same data as MLR, ensuring a fair comparison. Results for external baselines (Instruct, SimpleRL, DeepSeek-R1-Distill) may rely on different training corpora; we include them as strong reference points.
>
> In the SFT phase, we generate multiple high-quality trajectories for each problem by sampling four solutions from DeepSeek-R1, yielding approximately 80K filtered trajectories in total. Each accepted trajectory is then decomposed into step-by-step reasoning segments using DeepSeek-V3 via in-context learning, with each step annotated by a step descriptor. To enable multi-level reasoning, we further distill each step into a concise high-level summary, again using DeepSeek-V3 with in-context learning.
>
> All prompts used throughout this pipeline are provided in Appendix D. Figure 8 presents the resulting trajectory length distributions.
> For reproducibility, we also provide all prompts used in data creation and MLR inference in Appendix D.

---

> ### Author Response · Authors · 2025-11-24
> **Response to Reviewer 7Cso (Part 3)**
>
> ### **Response to Question 2**
>
> We add a detailed description in Appendix C. We use Qwen2.5-0.5B-Instruct as the summarization model; it is fully fine-tuned for summarization and then frozen, and shared across all base models. The typical summary length ranges from 15 to 150 tokens; we show the length distribution in Figure 7.
>
> ### **Response to Question 3**
>
> We apologize for the lack of clarity.
>
> "Estimated survival accuracy" is the classification accuracy of the survival estimate. For each partial trajectory $(d^{1:m}, c^{1:m})$, we assign a ground-truth survival label $y \in$ {$0,1$} using Monte Carlo lookahead with many rollouts of the base model: $y = 1$ if at least one rollout from this prefix reaches a correct final answer (the prefix is survivable), and $y = 0$ otherwise.
>
> Using the fast rollout model, for a given $K$, we sample $K$ continuations from the same prefix and compute the empirical success rate $ \hat g_K = \frac{1}{K}\sum_{k=1}^K \mathbf{1}{(\text{rollout } k \text{ is correct})}$. We then predict a survival label $ \hat y_K = 1 \text{ if } \hat g_K > 0 \text{ else } 0$.
>
> The prediction for that prefix is correct if $\hat y_K = y$. Estimated survival accuracy for a given $K$ is the fraction of prefixes for which $\hat y_K$ matches the ground-truth label $y$.
>
> This definition has been added to Appendix C.
>
> ### **Response to Question 4**
>
> Multiple sampling rounds per prefix can further reduce randomness, but there is a trade-off between prediction accuracy and efficiency. In practice, for each prefix we perform a single batch of $K = 4$ rollouts and introduce a margin threshold $\delta = 0.4$ when selecting preference pairs. This yields a good balance, maintaining reliable preferences without incurring excessive computational cost.
>
> We study the effect of varying the rollout count $K$.
> Utility estimates produced by the 1.5B rollout policy with $K$ rollouts are compared against reference utilities from the 8B model using 16 rollouts (Figure 15).
> As expected, increasing $K$ reduces estimator variance, though at the cost of higher computation.
> To mitigate this overhead, we introduce a margin threshold $\delta$ when selecting preference pairs.
> We further measure the agreement of the selected preference pairs as a function of $K$, defined as the fraction whose preference direction agrees with the base model using 16 rollouts (Figure 8).
> Finally, we report model performance across training stages under different values of $K$ (Figure 12).
> The results show that our chosen setting attains comparable final performance while substantially reducing computational cost.
> Implementation details are provided in Appendix D.
>
> ### **Response to Question 5**
>
> We have clarified this in Section 4 (Implementation details). In our experiments, we use Qwen-2.5-1.5B, fine-tuned on the low-level data, as the fast rollout policy. After SFT, this rollout model is frozen and shared across all base models.

---

> ### Author Response · Authors · 2025-12-03
> **Additional clarification for Reviewer 7Cso**
>
> As the discussion period concludes, we would like to briefly summarize how we have incorporated your comments into the revision.
>
> * **Clarifying the motivation.**
>    We reframe 'CoT derailment' as *long-horizon plan failure*, i.e., cases where the model’s implicit plan gradually drifts away from a valid solution strategy. We expanded the explanation in the main text (Sec. 2) and added a *long-horizon concatenated-problems evaluation* (Fig. 1), where MLR exhibits noticeably slower performance degradation than all baselines as the horizon increases.
>
> * **Comparison with baselines.**
>    We *construct shared SFT/RL splits* for all four benchmarks and train *SFT+DPO, SFT+Step-DPO, and SFT+GRPO* on *exactly the same prompt pool* as MLR. While preference optimization clearly strengthens the single-policy baseline over pure SFT, MLR still achieves higher accuracy across all benchmarks, with the largest gains on AIME24 and GPQA (Table 1).
>
> * **Summarization module.**
>    We clarify that summarization is handled by a *separate Qwen2.5-0.5B-Instruct model*, fully fine-tuned on summarization data and then frozen, and *shared across all base models* (Sec. 3.2, Appendix C). We report the empirical summary length distribution (Figure 7), highlighting how this module keeps the planner’s inputs concise and stable for long trajectories.
>
> * **Justification and ablation for rollout count in TSMC.**
>    We include a *parameter study on (K)* (Sec. 4). Utilities estimated by the 1.5B rollout policy with different $K$ are compared against 8B reference utilities (Figure 15), we report the *agreement of selected preference pairs* versus $K$ (Figure 8), and we show *training curves under different* $K$ (Figure 12). These results demonstrate that our selection offers a good balance between reliability of utilities and computational cost, and achieves final performance comparable to larger $K$.
>
> * **Implementation details.**
>    We add a detailed description of *data construction and prompts*: the use of public training splits from MATH500, AIME24, GPQA, and BoardGameQA; data augmentation via GPT-4o plus DeepSeek-R1 filtering; 60/40 SFT/RL splits; and the multi-trajectory, multi-level decomposition pipeline. We also:
>
>    * Precisely define *“estimated survival accuracy”* as the classification accuracy of survival labels derived from Monte Carlo lookahead.
>    * Specify that the *fast rollout policy* is *Qwen-2.5-1.5B*, fine-tuned on low-level SFT data, then frozen and shared across all base models.
>      All prompts and additional implementation details are provided in the Appendix C and D to support reproducibility.
>
> We hope these modifications address your concerns, and we thank you again for the helpful feedback.

---

### Official Review · Reviewer_yNnM · 2025-10-31

**Soundness:** 3
**Presentation:** 3
**Contribution:** 3
**Rating:** 6
**Confidence:** 2

**Summary:**

This paper proposes Multi-Level Reasoning (MLR), a framework that addresses limitations in single-policy long chain-of-thought (CoT) approaches for language model reasoning. MLR decomposes reasoning into alternating high-level step descriptors (abstract sub-goals) and low-level detailed content. The framework uses the base model as the low-level policy and a lightweight LoRA module for the high-level policy. To enable efficient training, the authors introduce online Step-DPO, which leverages Twisted Sequential Monte Carlo (TSMC) to provide process-level preference signals without requiring a separate reward model. Experiments on MATH500, AIME24, GPQA-diamond, and BoardGameQA-hard demonstrate improvements over both distillation baselines (DeepSeek-R1-Distill) and outcome-reward RL methods across three base models.

**Strengths:**

- Novel application of HRL concepts to reasoning tasks with a practical, efficient design (frozen base + LoRA)
- TSMC-based process preferences are creative and avoid the overhead of training separate process reward models
- Strong empirical results: MLR consistently outperforms DeepSeek-R1-Distill despite using only 10% SFT data and 5% preference pairs
- Addresses a timely problem as reasoning models gain prominence

**Weaknesses:**

- 各个组件的创新性有限 - 主要贡献在于其组合方式而非从根本上采用新的技术

**Questions:**

see weakness

---

> ### Author Response · Authors · 2025-11-24
> **Response to Reviewer yNnM**
>
> We thank the reviewer for the positive assessment and have revised the paper to further clarify our contributions.
>
> ### Response to Weakness 1
>
> We respectfully disagree that the novelty is limited. Our main contribution is to provide a *new formulation* and *training scheme* for *long-CoT reasoning* in a regime where existing approaches struggle: effective training of LLMs on *very long CoTs* without a separate *process reward model*. Concretely:
>
> - **Multi-level formulation for long CoTs.**
> We reformulate long-CoT generation as a two-level stochastic process (Eq. (2)), where interleaved step descriptors and detailed content are modeled by separate policies. This differs from existing *planner-based* methods and prior long-CoT methods that rely on a *single policy*. Ablations in Table 2 and 3 show that using only a low-level policy, even with Step-DPO, yields substantially lower performance than our multi-level design.
>
> - **TSMC-based process-level preferences.**
> We repurpose Twisted Sequential Monte Carlo (TSMC) for process preference modelling with long CoTs (Eqs. (5)–(10)). To our knowledge, TSMC importance weights have not previously been used to generate stepwise preference signals for LM reasoning. This design removes the need for a *separate process reward model* while still providing dense, effective supervision along long trajectories.
>
> - **Synergistic effect of multi-level modelling and Step-DPO.**
> The gains from our approach are not a simple sum of components. As shown in Table 3, (i) multi-level modeling alone (“DPO-only”) and (ii) step-level supervision alone ("Low-level policy + Step-DPO") provide modest improvements over SFT, but MLR yields a *substantially larger improvement*, indicating a synergistic effect.
>
> - **Practical efficiency and robustness.**
> Despite using only about 10% of the SFT data and 5% of the preference pairs compared to DeepSeek-R1-Distill, MLR consistently outperforms it across benchmarks and backbones (Table 1). This suggests that our approach is not only conceptually distinct, but also *data- and compute-efficient*.
> To further support our contribution, we add a new *long-horizon reasoning* experiment (Figure 1), where we concatenate multiple problems into a single prompt. As the horizon increases, MLR *degrades* significantly more *slowly* than all baselines, indicating that our formulation and training scheme help the model *avoid plan failures* and yield *more robust long-horizon reasoning*.

---

### Official Review · Reviewer_bbdw · 2025-11-02

**Soundness:** 2
**Presentation:** 2
**Contribution:** 2
**Rating:** 4
**Confidence:** 3

**Summary:**

The authors propose Multi-Level Reasoning to enhance LLM reasoning with structured multi-level modeling. Specifically, a high-level planner first generates step descriptors as the roadmap, and then the low-level executor generates the detailed content based on the high-level planner, with the help of the online step-DPO algorithm to provide step-wise supervision. The proposed model achieves good performance on both math and science reasoning tasks based on different backbone models.

**Strengths:**

-	Overall, the idea is sound, and the proposed method achieved good results.
-	The authors have conducted ablation tests to verify its essential components.

**Weaknesses:**

-	In Section 3.2, the authors first fine-tune based on the low-level content via full SFT, and then learn from the high-level content via LoRA. The order of training, as well as the usage of full SFT/LoRA on different high-/low- level data, should be discussed (better with detailed experimental results).
-	In the main experiment, although the authors have implemented their method on diverse backbone models, the compared baselines are not strong enough. Other popular preference optimization methods (e.g., RL methods such as GRPO) should be compared.
-	In the Related work, there are some step-wise preference optimization methods or those that also adopt certain forms of overall planners, which should be discussed and even compared. The authors are suggested to highlight the technical novelty of the proposed model compared to existing efforts.

**Questions:**

Refer to the weakness part.

---

> ### Author Response · Authors · 2025-11-24
> **Response to Reviewer bbdw (Part 1)**
>
> We thank the reviewer for the constructive feedback and have updated the paper accordingly. To clarify our training design, we now include an **ablation of high-level SFT strategies** (Table 2), which shows that our strategy achieves the best performance. To address the concern regarding baseline strength, we add **three additional baselines** (SFT+DPO, SFT+Step-DPO, and SFT+GRPO) trained on the same prompt pool as our method, and report that MLR consistently outperforms all of them across benchmarks (Table 1).
>
> We also introduce a new **long-horizon reasoning evaluation** using concatenated problems (Figure 1), demonstrating that MLR exhibits slower performance degradation than all baselines as the horizon increases. Finally, we expand the **related work** discussion on step-wise preference optimization and planner-based methods, and explicitly highlight MLR’s technical novelty in scalable step-wise supervision and adaptive planning. We additionally include **Plan-and-Solve as a baseline** and show that MLR achieves superior performance (Table 1).
>
> ### **Response to Weakness 1**
>
> Thank you for raising this point. We agree that the *training order* and the choice of *full SFT vs. LoRA* for different levels are important design decisions. In our framework, we first fully fine-tune the base LM on low-level trajectories and then freeze it, attaching a parameter-efficient LoRA adapter for high-level planning. This design is motivated by two observations: (1) *low-level trajectories are long, fine-grained, and cognitively demanding, so they benefit from full-parameter capacity*; (2) *high-level trajectories are short and abstract*, making them well suited to lightweight LoRA tuning, which avoids *interfering with the already-optimized executor*.
>
> Training the planner on top of the low-level SFT model also reduces *covariate shift*, because the planner is learned in the regime where it will be used at inference. From an optimization standpoint, LoRA benefits from a *stronger backbone* (after low-level SFT) and mitigates overfitting by only adapting a small number of parameters. At deployment time, this configuration is also *efficient*, requiring only a single base model plus a small LoRA adapter (< 2% additional parameters).
>
> To empirically validate this design, we keep the low-level SFT fixed and compare three strategies for training the high-level planner on Qwen-2.5-1.5B:
> - *Ours:* SFT on low-level trajectories, then LoRA on high-level trajectories.
> - *LoRA on the original base:* apply LoRA directly to the unfine-tuned base model using only high-level trajectories.
> - *Full SFT on high-level trajectories:* train a separate model with full-parameter SFT using only high-level data.
>
> All variants use the same optimizer family (AdamW with cosine schedule and warmup) and comparable training budgets. As shown in Table 3, our default configuration (SFT on low-level + LoRA on high-level) achieves the best performance on both MATH500 and AIME24, with particularly strong gains on AIME24, which requires deeper multi-step planning. For example, on AIME24 our method improves Pass@1 from 4.1 and 6.5 to 8.9, and Cons@32 from 9.2 and 11.0 to 13.3.
>
> These results support that (i) *full SFT* is necessary for modeling *the longer, more complex low-level reasoning*, and (ii) *learning the planner* as a lightweight *LoRA adapter on top of the SFT-enhanced base* is more *effective* and *stable* than either tuning LoRA on the original base or fully fine-tuning a separate high-level-only model.
>
> We have incorporated this clarification and the corresponding ablation table into the revised manuscript (see Table 2, Appendix C).
>
> ### **Response to Weakness 2**
>
> We appreciate the suggestion to compare against stronger preference-optimization baselines such as GRPO. As discussed in Section 2, *outcome-based RL* on *long trajectories* is *computationally demanding*: after the SFT phase, our models produce long chains-of-thought, so rollout-based RL must backpropagate through thousands of tokens per example. This makes large-scale GRPO training substantially more *expensive* than applying to base models with much shorter completions (e.g., <2K tokens). Nonetheless, to address this concern, we implement three additional baselines (SFT+DPO, SFT+Step-DPO, SFT+GRPO) and include them in the main results (Table 1).

---

> ### Author Response · Authors · 2025-11-24
> **Response to Reviewer bbdw (Part 2)**
>
> To compare MLR with a standard single-policy preference-optimization method, we train on the *same prompt pool* used for our method. We first construct a single-policy SFT checkpoint by fine-tuning the base model on the processed low-level trajectories, where all step descriptors are removed from both inputs and targets, using the same optimizer, schedule, and token budget as our low-level SFT.
>
> Starting from this checkpoint, we run full-parameter GRPO, keeping a frozen copy of the SFT model as the reference policy. We use Verl with vLLM as the rollout backend. For each prompt, we sample groups of $N=4$ candidate continuations with temperature 0.7 and top-p=0.9, cap the maximum response length at 4096 tokens to respect GPU memory limits, and assign a rule-based outcome reward of $1$ if the final answer is correct and $0$ otherwise. The actor is optimized with AdamW (learning rate $5\times10^{-7}$, weight decay $0.1$) under a KL-penalty objective with coefficient $\lambda_{\mathrm{KL}}=0.02$, using bf16, gradient checkpointing, and FlashAttention. We train for 4 epochs with a global batch size of 32 (PPO mini-batch size 16, micro-batch size 2) and evaluate every 100 steps on a held-out validation split, selecting the checkpoint with the best validation Pass@1.
> We report the evolution of validation accuracy and average response length during GRPO training in Figure 13 and Figure 14, and summarize the final results in Table 1.
> More *implementation details* can be found in Appendix C.
>
> Preference-optimization such as GRPO clearly strengthens the baseline over SFT, but MLR still achieves *substantially higher* performance across all benchmarks, especially on the more challenging AIME24 and GPQA. Moreover, we find that *outcome-reward RL* is *less efficient* for *long-horizon reasoning*: outcome rewards are (i) *sparse*, making credit assignment difficult for long chains-of-thought, and (ii) *computationally expensive*, since each update requires generating full rollouts with long responses. In practice, when starting from fine-tuned models with long CoTs, these costs limit the applicability of GRPO to larger base models (e.g., Qwen-2.5-MATH-7B SFT, Llama-3.1-8B SFT). In contrast, our online Step-DPO procedure is *simpler to implement*, *more stable*, and *significantly cheaper*, while delivering *stronger final performance*.
>
> We have incorporated the discussion and results into the revised manuscript (Table 1, Appendix C).
>
> ### **Response to Weakness 3**
>
> We appreciate the suggestion to better position our work relative to step-wise preference optimization and planner-based methods.
>
> Direct Preference Optimization (DPO) has been criticized for relying on purely *offline data* and *trajectory-level preferences*. Step-DPO [1] mitigates this by constructing curated step-wise preference data, while DPO with AI feedback [2-3] enables online updates. However, these approaches do *not scale* well to *long chains-of-thought*: they depend on *strong teacher models* (e.g., GPT-4) to provide step-level feedback, which is both *expensive* and often *unreliable on harder problems*.
>
> To address this, we propose a scalable *TSMC-based method* in which a *smaller* LM with *estimated utilities* supplies *step-wise preferences*. Unlike *naive tree-search* methods [4], which become prohibitively *expensive* on long trajectories, our approach remains *efficient and stable* for long-horizon supervision.
>
> On the other hand, MLR adopts a *multi-level* strategy, differing from prior planning methods [5-7] that *generate a full plan upfront* and implicitly assume that *all subtasks succeed as written*. Such fixed plans are brittle: *early mistakes propagate* through the entire reasoning process. In contrast, MLR learns a planner that *adapts its plans* based on *execution feedback*, allowing revisions and yielding more robust long-horizon reasoning.
>
> We further add comparison to the mentioned baselines.
> Because step-wise preference optimization with *AI feedback* is prohibitively *costly* for long-CoT reasoning and yields *noisy judgments* at the step level, we do not treat these methods as direct baselines in our setting.
> Instead, we compare against *Plan-and-Solve* [7], which first generates a concise global plan and then executes the solution conditioned on that plan. The example prompt is given (see Appendix D). For a fair comparison, we use the same backbone as our method and fine-tune two LoRA heads on top of it: a planner (Problem $\to$ Plan) and an executor (Problem + Plan $\to$ Solution). At inference time, we follow the standard two-pass Plan-and-Solve pipeline: Pass 1 generates the plan; Pass 2 solves the problem conditioned on that plan.

---

> ### Author Response · Authors · 2025-11-24
> **Response to Reviewer bbdw (Part 3)**
>
> Table 1 summarizes the performance. Because Plan-and-Solve here is trained only with SFT, we primarily compare it against MLR (SFT-only). Across all three backbones, MLR consistently outperforms Plan-and-Solve, with the largest margins on the *more challenging* benchmarks (AIME, GPQA). Empirically, we observe that Plan-and-Solve often assumes that *all subtasks succeed as initially planned*; *errors in early steps propagate*, and *the executor may partially deviate from the plan*. In contrast, MLR learns a planner that can *adapt its plans* based on execution signals, enabling revisions rather than committing to a fixed blueprint. This *tighter, adaptive coupling between planner and executor* yields more stable and robust long-horizon reasoning than prompting a static plan upfront and executing it verbatim.
>
> We have incorporated the discussion and results into the revised manuscript (see Table 1, Appendix C).
>
> In addition, to better support our claims on long-horizon reasoning, we have added a new experiment (Figure 1). This setup concatenates multiple problems per prompt, and MLR shows significantly slower degradation than all baselines as the horizon increases, further highlighting its advantage in long-horizon reasoning.
>
> ## Reference:
>
> [1] Lai, Xin, et al. "Step-dpo: Step-wise preference optimization for long-chain reasoning of llms." arXiv preprint arXiv:2406.18629 (2024).
>
> [2] Cui, Ganqu, et al. "Ultrafeedback: Boosting language models with high-quality feedback." (2023).
>
> [3] Guo, Shangmin, et al. "Direct language model alignment from online ai feedback." arXiv preprint arXiv:2402.04792 (2024).
>
> [4] Wang, Peiyi, et al. "Math-shepherd: Verify and reinforce llms step-by-step without human annotations." Proceedings of the 62nd Annual Meeting of the Association for Computational Linguistics (Volume 1: Long Papers). 2024.
>
> [5] Huang, Wenlong, et al. "Language models as zero-shot planners: Extracting actionable knowledge for embodied agents." International conference on machine learning. PMLR, 2022.
>
> [6] Xu, Binfeng, et al. "Rewoo: Decoupling reasoning from observations for efficient augmented language models." arXiv preprint arXiv:2305.18323 (2023).
>
> [7] Wang, Lei, et al. "Plan-and-solve prompting: Improving zero-shot chain-of-thought reasoning by large language models." arXiv preprint arXiv:2305.04091 (2023).

---

> ### Author Response · Authors · 2025-12-03
> **Additional clarification for Reviewer bbdw**
>
> As the discussion period concludes, we would like to briefly summarize how we have addressed your comments in the revised manuscript.
>
> * **Training order and SFT/LoRA design.**
>    We provided a detailed explanation of why we first apply full SFT on low-level trajectories and then train the high-level planner via LoRA on top of the SFT model (Sec. 3.2). Intuitively, low-level trajectories are long and fine-grained, so they benefit from full-parameter capacity, while high-level trajectories are short and abstract, making them suitable for a lightweight adapter without overwriting the executor. To support this, we added an ablation of high-level SFT strategies (Table 2, Appendix C), comparing: (i) our strategy, (ii) high-level LoRA on the original base model, and (iii) full SFT on high-level data. Our design achieves the best performance, especially on AIME24, supporting both the training order and the choice of SFT vs. LoRA.
>
> * **Stronger preference-optimization baselines.**
>    To strengthen the empirical comparison, we added three preference-optimization baselines trained on the *same prompt pool* as MLR: *SFT+DPO*, *SFT+Step-DPO*, and *SFT+GRPO* (Table 1, implementation details in Appendix C). These methods clearly improve over pure SFT, but MLR still attains higher accuracy on all benchmarks, particularly on challenging AIME24 and GPQA.
>
> * **Related work and baselines.**
>    We expanded the Related Work section to discuss step-wise preference optimization and planner-style methods, and clarified how MLR differs in its *scalable step-wise supervision* and *adaptive* multi-level planning. In addition, we implemented *Plan-and-Solve* as a baseline. As shown in Table 1, MLR (SFT-only) consistently outperforms Plan-and-Solve across all backbones. Empirically, we find that *fixed upfront plans are brittle*, whereas MLR’s planner can *adapt its plan based on execution feedback*, leading to more robust long-horizon reasoning. To further support this claim, we added a *long-horizon concatenated-problems evaluation* (Fig. 1), where MLR exhibits slower performance degradation with increasing horizon than all baselines.
>
> We hope this summary clarifies how your suggestions have been incorporated and how they strengthened both the method and its empirical evaluation.

---

### Meta-Review · Area_Chair_ptUq · 2025-12-30

**Summary:**

This paper introduces Multi-Level Reasoning (MLR), a framework designed to enhance the long-horizon reasoning capabilities of Large Language Models. The approach reformulates Chain-of-Thought generation as a two-level stochastic process: a high-level planner, implemented via a lightweight LoRA module, generates abstract step descriptors, while a low-level executor, the base model, produces detailed reasoning content conditioned on these descriptors in an alternating loop. To train this system effectively without a separate process reward model, the authors propose an online Step-DPO algorithm that utilizes Twisted Sequential Monte Carlo to estimate step-wise survivability utilities. The method is evaluated on math, science, and logical reasoning benchmarks, demonstrating improvements over distillation and standard reinforcement learning baselines.

**Reviewer Concerns:**

The initial reviews presented a mixed to negative outlook, primarily centering on the strength of baselines, the fairness of comparisons, and the justification for specific design choices. However, the authors provided an exceptionally thorough rebuttal that addressed the majority of these concerns.

1. First, a major concern raised by multiple reviewers (bbdw, 7Cso, fiGr) was the lack of strong reinforcement learning baselines (specifically GRPO) and the unfairness of comparing the proposed method against models trained on different data sources. The reviewers argued that without controlling for the training data and utilizing state-of-the-art RL baselines, the performance gains could not be attributed to the MLR framework. In response, the authors conducted extensive new experiments, training SFT+DPO, SFT+Step-DPO, and SFT+GRPO baselines on the exact same data splits as their method. The results demonstrated that MLR consistently outperformed these rigorous baselines, particularly on challenging datasets like AIME24. From a reviewer's perspective, this is a comprehensive resolution. The authors accepted the heavy computational burden to ensure a fair comparison, and the resulting data effectively invalidates the argument that the gains are merely due to data or baselines. While the inherent complexity of the system remains, the empirical superiority over GRPO on identical data is a strong defense.

2. Second, reviewers questioned the rationale behind specific architectural and training decisions (bbdw, 7Cso, fiGr). Specifically, they asked for justifications regarding the training order (SFT on low-level followed by LoRA on high-level), the choice of the rollout hyperparameter , the necessity of the hierarchical structure, and details regarding the summarization module. The authors responded with a suite of ablation studies. They empirically demonstrated that their specific training strategy outperformed alternative orders and showed that removing either the high-level or low-level components leads to performance degradation. They also provided a sensitivity analysis for the  value in TSMC and clarified that the summarizer is a separate 0.5B model. These responses satisfactorily address the technical skepticism. The provided evidence moves the design choices from "arbitrary" to "empirically justified," leaving little room for further objection on these grounds.

3. Third, reviewers (yNnM, 7Cso) expressed concerns regarding the motivation—specifically the concept of "CoT derailment"—and the overall novelty of the work, suggesting it was merely a combination of existing techniques (HRL + TSMC). The authors clarified the motivation by reframing it as "long-horizon plan failure" and introduced a new experiment involving concatenated problems to visually demonstrate that MLR degrades more slowly than baselines as the reasoning horizon expands. While the motivation is now clearly supported by the new "long-horizon" experiments, the concern regarding novelty remains the most significant lingering issue. The rebuttal proves the method works well, but it does not change the fact that the contribution is primarily an effective system engineering combination rather than a fundamental algorithmic breakthrough. However, given the strong empirical results, this is likely a trade-off reviewers are willing to accept.

**Reviewer Scores:**

* **Reviewer bbdw (4  to 6):** The reviewer's primary demand for a GRPO baseline and discussion on training strategies was fully met.
* **Reviewer yNnM (6 to 6):** This reviewer was already positive. While the novelty concern persists, the strengthened experiments reinforce the "Weak Accept" position.
* **Reviewer 7Cso (4 to 6):** The author addressed every specific technical question (data sources,  value, summarizer details) and the motivation issue. Maintaining a rejection would be difficult given the quality of the response.
* **Reviewer fiGr (4 to 4 or 6):** The fairness of comparison and the request for hierarchical ablations were addressed comprehensively.

---

### Decision · Program_Chairs · 2026-01-26

Accept (Poster)